# A new paradigm of islet adaptations in human pregnancy: insights from immunohistochemistry and proteomics

Faheem Seedat [1,2] ✉, Katie Holden [1], Simon Davis [3,4], Roman Fischer [3,4], James Bancroft[1], Edward Drydale [1], Neva Kandzija[2], John A. Todd[1], Manu Vatish [1,2] & M. Irina Stefana [1] ✉

Physiological changes during pregnancy support foetal growth, including adaptations in pancreatic islets to maintain glucose homeostasis. We investigate these adaptations using rare, high-quality pancreatic tissue from pregnant human donors and matched controls. We profile islets from pregnant donors using proteomics and assess α- and β-cell characteristics, as well as prolactin receptor and serotonin 2B receptor expression. Proteomic profiling of microdissected human islets identifies 7546 proteins but shows minimal differences in protein expression. In pregnancy, we show that islet area increases 1.9-fold, α- and β-cell areas increase 4.3- and 1.9-fold, driven by an increase in cell number rather than hypertrophy. Prolactin receptor expression is higher in α but not β cells, and serotonin 2B receptor is undetectable in β cells. Glucagon-like peptide-1 abundance increases 2.9-fold in α cells. These findings indicate that the molecular mechanisms driving pregnancy-induced islet adaptations in humans differ from those in mice, highlighting the need for human-based studies.

Pregnancy triggers physiological adaptations across multiple organ systems to support foetal growth and prepare the mother for delivery[1]. This includes the pancreatic islets of Langerhans, which house insulin-producing β cells and glucagon-producing α cells that regulate glucose metabolism[2,3]. Insulin facilitates cellular glucose uptake to lower blood glucose levels, while glucagon increases blood glucose levels by promoting glucose release from the liver[2,3]. As pregnancy progresses, insulin resistance gradually increases, requiring a compensatory rise in insulin secretion to maintain glucose homoeostasis[4,5]. Compared to pre-pregnancy, insulin secretion increases by 50% in early gestation and 100% by late gestation[6]. Pregnancy, thus, represents a state of enhanced islet-cell function. Understanding the mechanisms that enable islet plasticity during pregnancy could inform the development

of pharmacotherapies to enhance insulin secretion, offering potential novel strategies for diabetes management.

Failure to adequately increase insulin secretion in pregnancy leads to gestational diabetes mellitus (GDM), defined as the first occurrence of hyperglycaemia during pregnancy[1]. GDM is the most common pregnancy-related condition and is associated with materno-foetal complications, such as: large-for-gestational-age offspring, shoulder dystocia, and neonatal intensive care admission. Furthermore, GDM elevates the future risk of type 2 diabetes for the mother and increases the child's lifelong risk of type 2 diabetes, obesity, and cardiovascular disease[7,8]. Elucidating the molecular mechanisms underlying pregnancy-related islet adaptations is crucial for understanding GDM.

[1]Centre for Human Genetics, Nuffield Department of Medicine, University of Oxford, Oxford, UK. [2]Nuffield Department of Women's and Reproductive Health, University of Oxford, Oxford, UK. [3]Target Discovery Institute, Centre for Medicines Discovery, Nuffield Department of Medicine, University of Oxford, Oxford, UK. [4]Chinese Academy for Medical Sciences Oxford Institute, Nuffield Department of Medicine, University of Oxford, Roosevelt Drive, Oxford, UK. ✉e-mail: faheem.seedat@wrh.ox.ac.uk; seedat.faheem@gmail.com; maria.irina.stefana@gmail.com

Little is known about how human islets adapt to increase insulin secretion during pregnancy or why they fail to secrete enough insulin in GDM. Studies mapping transcriptomic profiles of pregnant mouse islets have thus far shaped our understanding of global islet changes during pregnancy[9-12]. However, changes in mRNA expression do not always correlate with changes in protein levels[13]. Only one in-depth proteomic study on islets from mouse pregnancy has been conducted, and, to our knowledge, no similar research has used pancreatic tissue from pregnant women[14]. Characterising the human islet proteome in pregnancy could uncover mechanisms driving pregnancy-related islet adaptations and functional plasticity.

Histological studies in humans show that β-cell area increases by 1.4- to 2.4-fold during pregnancy, but the mechanisms behind this expansion remain unclear[15,16]. Mouse models suggest this process is regulated by lactogenic hormones, such as prolactin and placental lactogen. Lactogenic hormones bind to the prolactin receptor (PRLR) on β cells, activating signalling pathways that promote β-cell proliferation and insulin secretion. Serotonin synthesis within β cells, stimulated by lactogenic hormones, further promotes β-cell proliferation during pregnancy, via the serotonin 2B ($5$-$HT_{2B}$) receptor, which is upregulated during mouse pregnancy and downregulated postpartum[1,4,5]. Additionally, α-cell mass increases in mouse pregnancy due to increases in cell size and replication[17]. While these mechanisms are well-documented in mice, their role in human islets remains unclear.

The limitations of using mouse models to study human pregnancy have been previously described[18,19]. Moreover, human and mouse islets have several biological differences. They differ in cellular composition, organisation, architecture, size, number, vascularisation, gene expression signatures, and functional responses to stimuli. Importantly, mice β cells exhibit robust proliferative capacity, while adult human β cells are largely quiescent, with replication rates below 0.1%[20-23]. These species differences highlight the importance of cautious interpretation when translating findings from mouse models of pregnancy to human biology. Furthermore, whether β-cell replication is crucial to the increased insulin secretion observed during human pregnancy remains controversial[5], underscoring the need for studies using human tissues.

Difficulty accessing well-preserved pancreatic tissue presents a challenge to research into islet biology. Pancreatic samples can only be obtained postmortem. However, autopsy-derived samples are often degraded due to rapid autodigestion by pancreatic digestive enzymes from the exocrine pancreas, leading to tissue necrosis, disruption of islet architecture, and destruction of islet proteins[24]. Additionally, studies on human pregnancy present practical and ethical challenges, further limiting access to pancreatic tissue from pregnant women. The Network for Pancreatic Organ Donors with Diabetes (nPOD) at the University of Florida is the largest biorepository of human pancreatic tissue worldwide. It collects pancreata from heart-beating donors, with a cold ischaemia time of less than 24 hours between organ collection and tissue processing. This rapid preservation process minimises postmortem tissue degradation, providing high-quality samples. Among the limited global sources, nPOD offers well-preserved pancreatic tissue from pregnant women, enabling the study of ex vivo islet architecture and protein expression in pregnancy[25,26]. Laser-capture microdissection (LCM) is a powerful technique for isolating, from complex tissues, minority cell populations that have been preserved in their native environment, allowing for their precise molecular analyses[27,28]. This method can be applied to pancreatic tissues to isolate islets for further study.

We utilise formalin-fixed paraffin-embedded (FFPE) pancreatic tissue sections from nPOD to isolate pancreatic islets by LCM and by liquid-chromatography mass spectrometry (LC-MS/MS) to characterise their protein expression profiles in pregnant women compared to non-pregnant controls. Additionally, we characterise pregnancy-associated changes in whole islet, as well as α- and β-cell metrics. High-resolution imaging of whole tissue sections was combined with powerful automatic image analysis and quantification, to ensure reproducibility and a lack of bias. We validate antibodies for and examine the expression of PRLRs and $5$-$HT_{2B}$ receptors in human α and β cells during pregnancy, while also measuring the abundance of glucagon-like peptide-1 (GLP-1) in α cells in pregnancy. By integrating deep proteomic profiling of LCM-isolated human pancreatic islets with high-resolution whole tissue imaging, we provide the most extensive characterisation, to date, of pregnancy-associated molecular changes that occur in human islets.

## Results

### Successful isolation and deep proteomic profiling of human islets and exocrine tissue using LCM and LC-MS/MS

Islets and exocrine tissue were successfully isolated from FFPE pancreas tissue sections from both pregnant and non-pregnant donors using LCM (Table 1 and Supp. Data S1). On average 10 islets per donor were analysed (Supp. Data S1). Isolated islets underwent unbiased proteomic characterisation by LC-MS/MS, followed by quantification of the detected proteins (Fig. 1a, 1b and 1c) (Supp. Data S3).

A similar number of proteins were detected in all samples (Fig. 1d). Per donor, the mean number of proteins identified in islets was 7388 (SD = 83) and 7124 proteins (SD = 120) in exocrine tissue.

The purity of islet isolation by LCM was demonstrated by the clear separation between the islet and exocrine tissue clusters observed in the principal component analysis (Fig. 1e). Furthermore, comparison of all islet and exocrine tissue samples revealed that several islet-specific proteins were detected at high levels in islet samples. (Fig. 1f). Insulin (INS), glucagon (GCG), and chromogranin A and B (CHGA and CHGB) were all highly expressed in isolated islets (Fig. 1f). Conversely, acinar cell-specific enzymes, amylase (AMY2A), lipase (PNLIPRP1), trypsin (PRSS1, PRSS2, and PRSS3), and chymotrypsin-like elastases (CELA2A, CELA2B, CELA3B, and CTRL), were detected in exocrine samples but not in islets (Fig. 1f).

Pathway analysis of proteins upregulated in islets confirmed enrichment of islet-specific pathways and pathways related to insulin processing and the regulation of insulin secretion (Fig. 1g).

**Table 1 | Comparison of the characteristics of pregnant women and non-pregnant controls who donated their pancreata**

| | Non-pregnant (n = 7) | Pregnant (n = 7) | P value |
|---|---|---|---|
| Age (years) | 25 (21–33) | 33 (28–26) | 0.15 |
| Ethnicity - no. (%) | | | |
| African American | 2 (28) | 3 (42) | >0.99 |
| East Asian | 1 (14) | 1 (14) | |
| White | 2 (28) | 1 (14) | |
| Latinx | 2 (28) | 2 (28) | |
| Body mass index (kg/m²) | 28.3 (24–31.3) | 34.4 (27.2–45) | 0.07 |
| Gestational age at delivery (weeks) | - | 33 (32–40) | N/A |
| Haemaglobin A1c (%) | 5.4 (5.2–5.9) | 5.4 (5–6.6) | 0.91 |
| C-Peptide (ng/ml) | 7 (1.9–8.1) | 6.4 (1.8–10.5) | >0.99 |
| Anti-GAD antibody positive - no. (%) | 0 (0) | 2 (29) | 0.46 |
| Insulinitis detected (histology) – no. (%) | 0 (0) | 2 (29) | 0.46 |

ª data are described using median/interquartile range (IQR). Continuous data were compared using the two-sided Mann-Whitney test and categorical data, presented as numbers (no.) and percentages (%), were compared using two-sided Fisher's exact tests.

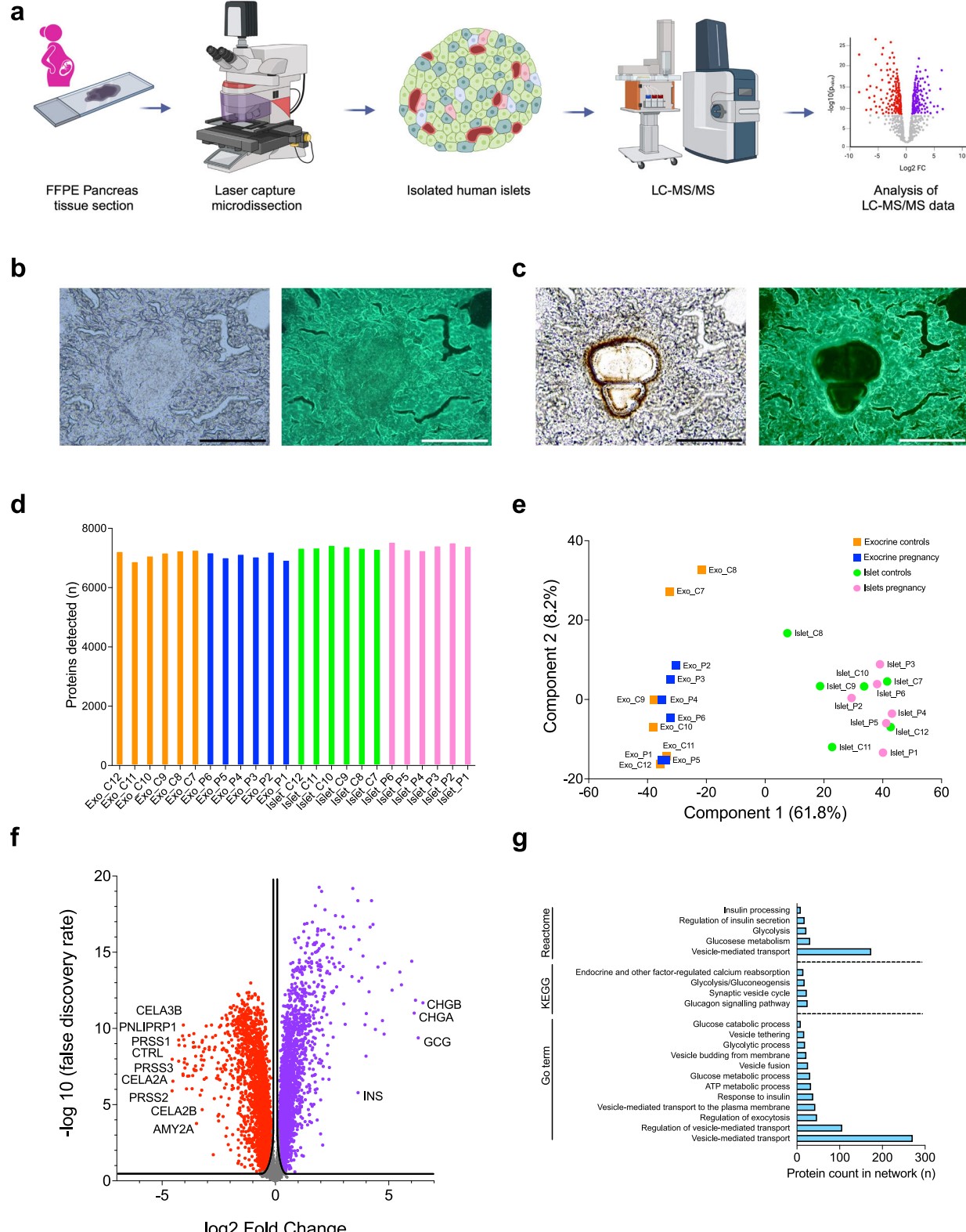

## Characterisation of the human islet proteome in pregnancy

Proteins expressed in islets from pregnant women were then compared to proteins expressed in pancreatic exocrine tissue from pregnant women. 2057 proteins were upregulated in islets, with 446 showing a log2 FC greater than 1 (Fig. 2a).

Over-representation analysis of these proteins enriched six pathways related to islet functional biology (Fig. 2b). Network clustering

identified 69 clusters. Four clusters contained three or more proteins associated with the enriched pathways (Fig. 2c). Cluster 1 comprised several subunits of the Vacuolar-type proton ATPase, which enriched the ion channel transport, synaptic vesicle, lysosome, and phagosome pathways. Cluster 2 consisted of several Guanine nucleotide-binding proteins (G proteins), contributing to the enrichment of pathways related to energy metabolism and neuronal system pathways. Cluster 3

**Fig. 1 | Identification, isolation and analysis of isolated islets and exocrine tissue. a** Schematic illustrating the experimental workflow: pancreatic islets were isolated from formalin-fixed paraffin embedded (FFPE) pancreatic tissue sections. Their proteome was characterised by liquid chromatography-mass spectrometry (LC-MS/MS) and between group comparisons analysed. Data analysis included between-group comparisons using permutation-based testing to correct for multiple comparisons using a false discovery rate (FDR) of < 0.05. No statistical data are presented in this figure. Created in BioRender. Stefana, I. (https://BioRender.com/3j35qyz). **b** Identification of islets by brightfield and fluorescence (488 nm channel) microscopy. Images of islets at 20x magnification before laser-capture microdissection (LCM) are shown. Scale bar = 200 μm. **c** Brightfield and fluorescence images of islets post-LCM. Scale bar = 200 μm. Images in (**b**, **c**) are representative of 12 biological replicate samples included in the analysis, with 9–12 islets dissected

per sample, and similar results observed across all islets within each biological replicate. **d** Number of proteins detected per sample type following LC-MS/MS. **e** Principal component analysis plot of the samples. Isolated islets are represented as circles (pregnant cases in pink, controls in green), while exocrine tissue samples are shown as squares (pregnant cases in blue, controls in orange). **f** Identification of islet proteins, volcano plot comparing protein expression between isolated islets (pregnant and non-pregnant controls) and exocrine tissue (pregnant and non-pregnant controls). Significantly upregulated proteins are shown in purple, and downregulated proteins in red. Selected proteins with a log2 fold change > 3 or < −3 are labelled. FDR < 0.05 = - log10 FDR > 1.3, S0 = 0.01. **g** Overrepresentation analysis of enriched pathways from the Reactome, KEGG, and GO Biological Processes databases. Pathways with FDR < 0.05 are shown.

included proteins involved in synaptic vesicle docking, fusion, and transport. Cluster 4 primarily comprised proteins belonging to the ATP-binding cassette, particularly sub-family member 9 or sulfonylurea receptor 2 (Fig. 2c).

## Analysis of differentially expressed proteins in pancreatic islets of pregnant versus non-pregnant donors

Islet proteins from pregnant women were compared to those from non-pregnant controls. Of the 7546 proteins detected in human islet samples, cathepsin Z (CTSZ) (log2 FC = 1.36, -log10 false discovery rate (FDR) = 4.98), β1,4-galactosyltransferase 4 (B4GALT4) (log2 FC = 0.59, -log10 FDR = 4.44), and cyclin-dependent kinase 5 (CDK5) (log2 FC = 0.32, -log10 FDR = 4.82) were more abundant, while laminin subunit alpha 4 (LAMA4) (log2 FC = −0.66, -log10 FDR = 3.98) was less abundant in islets of third trimester pregnant women compared to non-pregnant controls (Fig. 2d). No physical interactions between these proteins were detected (Fig. 2e).

No differences in protein abundance were found in exocrine tissue between pregnant and non-pregnant controls, or in islets between normal pregnancy and GDM donors.

An additional exploratory analysis using a more permissive statistical threshold (FDR < 0.15) was conducted to further investigate proteomic changes in islets from pregnant women, identifying additional differentially abundant proteins (Supp. Fig. S1).

## Quantitative image analysis reveals pregnancy-induced increase of whole islet, α-cell, and β-cell area

Using an automated image analysis pipeline, unbiased quantitative analysis of whole-pancreas tissue sections labelled for islet cell markers was performed (Fig. 3a, b–u). Islets, identified by α- and β-cell markers, were scattered throughout the pancreatic tissue sections (Fig. 4a).

Compared to non-pregnant controls, pregnant donors exhibited a 1.9-fold increase in the fractional area (i.e., as a % of the total tissue section area) of islets (3 ± 0.46% vs. 1.6 ± 0.16%, $p = 0.0145$), a 4.3-fold increase in α-cell area (0.44 ± 0.12% vs. 0.1 ± 0.02%, $p = 0.0206$), and a 1.9-fold increase in β-cell area (1.26 ± 0.23% vs. 0.65 ± 0.07%, $p = 0.0241$), and a 5.4-fold increase in bihormonal cell area (0.17 ± 0.05% vs. 0.03 ± 0.01%, $p = 0.0137$) (Fig. 4b–e).

Despite the increase in fractional area occupied by islets, islet density (i.e., number of islets per mm² of tissue section) remained unchanged between pregnant and non-pregnant donors (Fig. 4m). Suggesting that observed changes were driven by an increase in islet size rather than islet number. In line with this, absolute quantifications revealed a 1.4-fold increase in the average area of individual islets (i.e., total islet area divided by number of islets) (8538.61 ± 551.65 μm² vs. 6253.80 ± 288.86 μm², $p = 0.032$), along with a 1.2-fold increase in islet diameter (92.73 ± 1.84 μm² vs. 109.28 ± 4.59 μm², $p = 0.0058$) in pregnant donors compared to non-pregnant controls (Fig. 4f and n).

Further analysis showed that average α cell and β cell, and bihormonal cell areas per islet (i.e., measured area divided by number of islets) increased by 2.9-fold for α cells (1162.98 ± 180.99 μm² vs.

407.72 ± 87.05 μm², $p = 0.0027$) and 1.3-fold for β cells (3469.66 ± 253.16 μm² vs. 2657.40 ± 156.02 μm², $p = 0.0182$), and 3.1-fold for bihormonal cells (408.43 ± 83.20 μm² vs. 132.83 ± 28.59 μm², $p = 0.0087$) (Fig. 4g–i). These findings indicate that the increase in islet size during pregnancy is driven by an expansion in α cell and β cell, and bihormonal cell mass (using area as a proxy for mass).

Pregnancy also altered the relative composition of cells within islets (as a % of total islet area). The proportion of islet area occupied by α cells increased 2.3-fold (13.16 ± 2.60% vs. 5.83 ± 1.32%, $p = 0.0273$), while the area occupied by bihormonal cells increased by 2.6-fold (4.64 ± 1.21% vs. 1.76 ± 0.45%, $p = 0.0452$). The proportion of islet area occupied by β cells remained unchanged (Fig. 4j-l). These data suggest that α cells and bihormonal cells contribute more significantly to the increase in islet size during pregnancy than β cells, which increase proportionally with whole islet size. The proportion of islets containing α cells was also analysed and no difference was observed between pregnant and non-pregnant donors (71.71 ± 4.97% vs. 68.86 ± 5.01%, $p = 0.5946$) (Fig. 4o).

No differences in the absolute size of individual α or β cells were detected between pregnant and non-pregnant donors (Fig. 5a, b), indicating the size of individual α and β cells remains constant, and suggesting that the increase in cell area observed during pregnancy is driven by increased cell numbers. In support of this, the α-cell count tripled (3.3-fold increase, 1.01 ± 0.28% vs. 0.3 ± 0.06%, $p = 0.0323$), and the β-cell count nearly doubled (1.7-fold increase, 2.11 ± 0.37% vs. 1.22 ± 0.12%, $p = 0.0384$) in pregnant donors compared to non-pregnant controls (Fig. 5c, d).

As we noted an increase in cell number, we sought to investigate if evidence of islet cell proliferation could be observed during human pregnancy and used the proliferation marker Ki-67 to label pancreatic tissue sections. No Ki-67-positive cells were detected in the islets of pregnant women at any gestational age including in the sample obtained from a 1st trimester donor, nor in non-pregnant donors (Fig. 5e). Positive Ki-67 staining was observed in the concurrently labelled malignant tissues and adjacent exocrine cells, confirming the functionality of the antibody (Supp. Fig. S2).

For all analyses reported here, no significant differences were observed between donors with GDM and normal pregnancies (Supp. Fig. S3, S4 and S5).

## Quantitative image analysis confirms pregnancy-induced upregulation of CTSZ in islets, validating LC-MS/MS findings

To validate the LC-MS/MS findings, immunofluorescence was performed on FFPE human pancreatic tissue sections (IHC-IF) using an anti-CTSZ antibody (Supp. Fig. S6a). Quantitative analysis of the CTSZ-positive area (normalised to whole islet area) and signal intensity (Supp. Fig. S6b and c) in islets from pregnant donors was compared to non-pregnant controls. A significant increase in CTSZ-positive area (0.3865 ± 0.0094% vs. 0.3305 ± 0.0215%, p = 0.038) and signal intensity (2643 ± 106.1 AU/mm² vs. 2169 ± 154.1 AU/mm²; p = 0.0295) in pregnant donors was observed, thereby validating the LC-MS/MS-

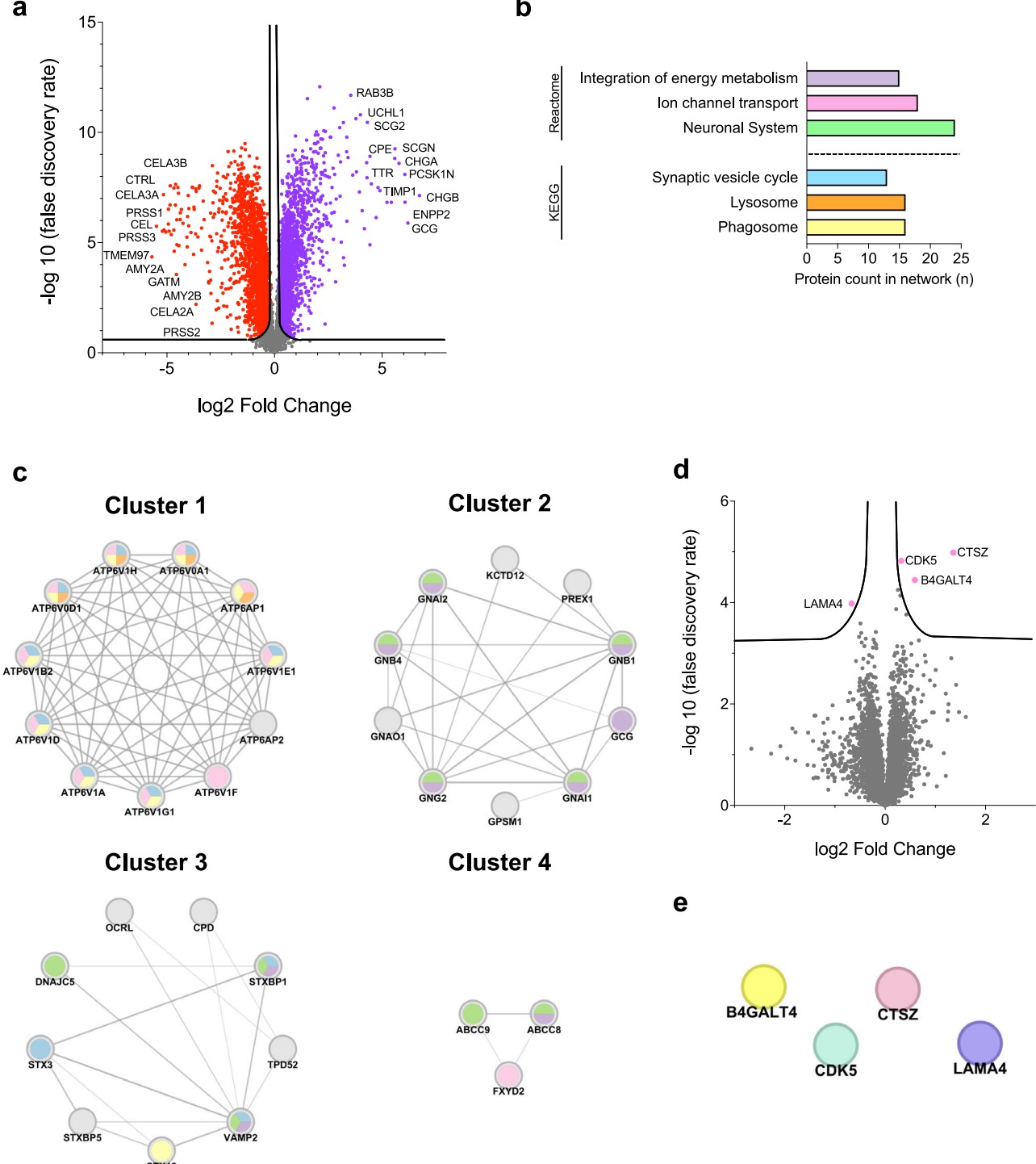

**Fig. 2 | Protein expression and pathway analysis of islets from pregnant women. a** Proteins in islets from pregnant women, volcano plot of proteins expressed in islets from pregnant women compared to exocrine tissue from pregnant women. Significantly upregulated proteins are shown in purple, and downregulated proteins in red. Selected proteins with a log2 fold change > 4 or < −4 are labelled. False discovery rate (FDR) < 0.05 = - log10 FDR > 1.3, S0 = 0.01. **b** Over-representation analysis of enriched pathways from the KEGG and Reactome data-bases. Upregulated proteins expressed in islets from pregnant women with a log2 fold change > 1 were analysed using the STRING database for physical protein interactions. FDR < 0.05. **c** Protein networks clustered by physical interactions between nodes. Networks with three or more proteins present in enriched path-ways are shown. Protein labels are displayed, and node colours correspond to the enriched pathways in (**b**). **d** Volcano plot illustrating differential protein expression in islets from pregnant women compared to islets in non-pregnant controls. Sig-nificantly expressed proteins are shown in pink, and are labelled. FDR < 0.05 = - log10 FDR > 1.3, S0 = 0.01. **e** Analysis of physical protein network interactions of differentially expressed proteins in islets from pregnant women analysed using the STRING database showing no interactions. FDR < 0.05.

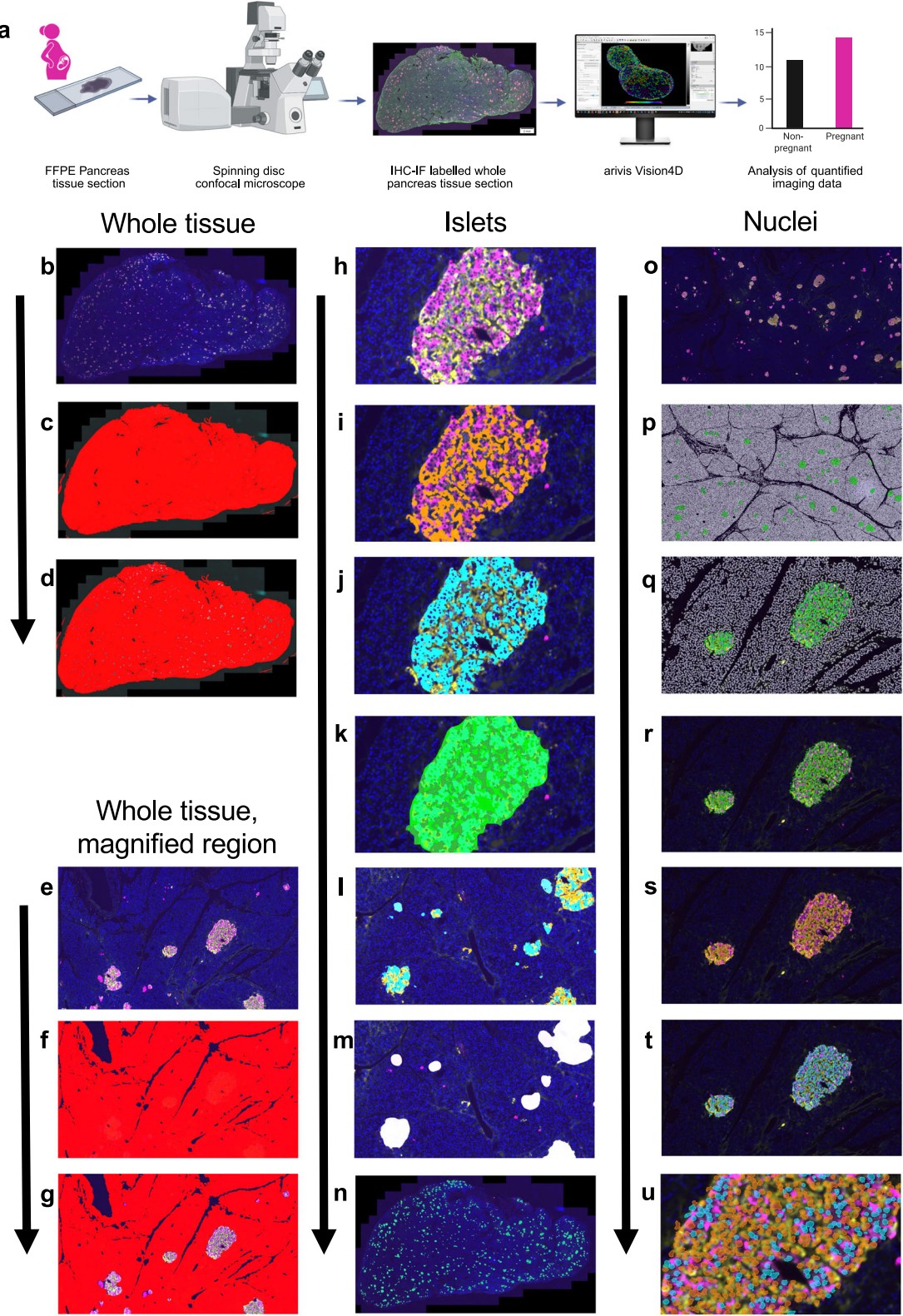

based finding of CTSZ upregulation in pancreatic islets during pregnancy.

### Validation of anti-PRLR and anti-5HT$_{2B}$ receptor antibodies

Mouse studies have suggested that PRLR and 5-HT$_{2B}$ receptor signalling play a role in β-cell adaptation during pregnancy. Neither the PRLR nor the 5-HT$_{2B}$ receptor was detected at the protein level by LC-MS/MS, likely due to their expression levels being below the detection threshold. Investigating their expression patterns in pancreatic β cells during pregnancy and quantifying protein levels within individual islet cell types required methodologies capable of discriminating with single cell resolution. As we only had access to FFPE pancreatic tissue, we decided to investigate the expression of PRLR and the 5-HT$_{2B}$ receptor during human pregnancy through combining high-resolution microscopy with affinity reagents that label the proteins of interest by IHC-IF.

**Fig. 3 | Experimental workflow and image analysis pipeline for human pancreatic sections labelled by immunofluorescence (IHC-IF). a** Schematic illustrating the experimental workflow: formalin-fixed paraffin-embedded (FFPE) pancreatic tissue sections were labelled by IHC-IF. Images were acquired and analysed to produce quantitative data using Arivis Vision4D image analysis software, data was compared between the pregnant and non-pregnant groups. Data analysis included between-group comparisons using the unpaired Student's t-test for normally distributed data and the Mann–Whitney test for non-parametric data, with statistical significance set at $P < 0.05$. Created in BioRender. Stefana, I. (2025) https://BioRender.com/frep1dj. **b**–**u** Images illustrating the steps of the analysis pipeline developed in Arivis Vision4D for examining IHC-IF labelled pancreatic tissue sections. **b** Whole tissue section, and a **e** magnified region, are labelled with DAPI (nuclei in blue), anti-glucagon (α cells in yellow), and anti-insulin (β cells in magenta) antibodies. **c, f** Thresholding was applied to the whole tissue section (red) and **d, g** exocrine tissue was identified by excluding whole islets from the thresholded regions (red). **h** Islets were also analysed and were labelled with anti-glucagon (yellow), and anti-insulin (magenta) antibodies, to mark α and β cells, respectively. **i** α cells (orange) and **j** β cells (cyan) were thresholded. **k** α and β cell compartments were merged to define whole islets (green). **l** α (orange) and β (cyan) cell compartments were marked across the entire tissue section. **m** Size filtering was applied to exclude islets smaller than 1000 μm² from further analysis. **n** Whole islets were defined across the whole tissue section (green). **o** Nuclei were also analysed within islet tissue and IHC-IF of labelled nuclei, α and β cells is shown. **p, q** Nuclei were thresholded in exocrine tissue (white) and whole islets (green). **r** Nuclei were thresholded within whole islets (green), **s** α cells (orange), and **t** β cells (cyan). **u** A magnified image of thresholded nuclei in α cells (orange), and nuclei in β cells (cyan) is shown.

First, we identified and validated commercially available antibodies to reliably detect the PRLR and 5-HT2B receptors in FFPE human pancreas sections. Of the antibodies tested, the anti-PRLR (Cell Signalling Technology cat.no. 13552) and anti-5-HT$_{2B}$ receptor (Santa Cruz Biotechnology cat. no. sc-376878) antibodies detected the respective target proteins with high specificity and sensitivity and were used for subsequent analysis.

The anti-PRLR antibody detected the PRLR receptor in HEK293T PRLR overexpression and parental WT HAP1 cell lysates by immunoblotting and no band was observed in HAP1 *PRLR* KO cells, (Supp. Fig. S7a). The PRLR was reliably detected in T47D FFPE cell pellets and FFPE placental tissue, known to express high levels of *PRLR* (Supp. Fig. S7b)[29]. The PRLR was also detected in HAP1 WT FFPE pellets but not in HAP1 *PRLR* KO FFPE cell pellets (Supp. Fig. S7c). Moreover, the PRLR was detected in FFPE human tissues known to express high *PRLR* levels, with no signal detected in tissues with low *PRLR* expression levels (Supp. Fig. S7d)[29]. By IHC-IF, the PRLR was detected in T47D FFPE cell pellets and in FFPE placental tissue which is known to express *PRLR* (Supp. Fig. S7e and S7f)[29]. Non-specific labelling of *PRLR* KO HAP1 cell pellets by anti-PRLR antibodies not used in this study are shown in Supp. Fig. S8.

The anti-5-HT$_{2B}$ receptor antibody successfully detected the target protein in cell lysates known to express the 5-HT$_{2B}$ receptor, as well as in the 5-HT$_{2B}$ receptor recombinant protein lysate by immunoblotting (Supp. Fig. S9a). No band was observed in cell lysates with low to absent levels of the 5-HT$_{2B}$ receptor (Supp. Fig. S9a). By IHC-IF, the anti-5-HT$_{2B}$ receptor antibody detected the receptor in WT HAP1 FFPE cell pellets and in FFPE cancerous tissues known to express high levels of the 5-HT$_{2B}$ receptor (Supp. Fig. S9b and c)[30].

### Increase abundance of PRLRs and 5-HT$_{2B}$ receptors in α cells during human pregnancy

Immunolabeling of the PRLR in both islets and exocrine tissue of pregnant and non-pregnant donors confirmed that the PRLR is expressed at detectable levels in adult human pancreatic islets (Fig. 6a, Supp. Fig. S10 and S11a [secondary only control]). Quantitative analysis of the PRLR signal intensity (as a corollary of PRLR abundance) revealed an increase in PRLR expression in α cells of pregnant donors compared to non-pregnant controls ($302.3 \pm 10.12$ AU/mm²  vs.  $267.7 \pm 4.582$ AU/mm²;  p = 0.0398), whereas no difference in PRLR expression was observed in β cells of pregnant donors ($277.4 \pm 3.002$ AU/mm² vs. $270.3 \pm 1.704$ AU/mm²; p = 0.0610) (Fig. 6b and c). No significant differences in PRLR expression were observed between donors with GDM and normal pregnancies (Supp. Fig. S11b and c).

5-HT$_{2B}$ receptor expression was detected in α cells in both pregnant and non-pregnant donor samples. Outside islets, 5-HT$_{2B}$ receptor expression was detected in peri-ductal regions (Fig. 7a and Supp. Fig. S12a [secondary only control]). Signal intensity quantifications

demonstrate an increase in the expression of 5-HT$_{2B}$ receptors in α cells of pregnant donors compared to non-pregnant controls ($387.4 \pm 10.01$ Au/mm² vs. $286.5 \pm 13.46$ Au/mm²; $p = 0.0359$) (Fig. 7b). No significant differences in 5-HT$_{2B}$ receptor expression (Supp. Fig. S12b) in α cells were observed between donors with GDM and normal pregnancies.

### Absence of 5-HT$_{2B}$ receptors in human β cells during pregnancy

In contrast to findings from mouse models, we did not detect 5-HT$_{2B}$ receptor expression in human β-cells in pregnant or non-pregnant donors (Fig. 8a). Colocalisation analysis confirmed absent colocalisation between 5-HT$_{2B}$ receptors and insulin across whole tissue sections (Fig. 8b and c). Importantly, pancreatic ducts, known to express the 5-HT$_{2B}$ receptor[31], showed clear signal, serving as an internal positive control within each pancreatic tissue section to validate the antibody's successful performance by IHC-IF (Fig. 8a).

To further validate the absence of 5-HT$_{2B}$ receptor expression in human β-cells from both pregnant and non-pregnant donors, we utilised tyramide signal amplification to enhance detection sensitivity in human pancreatic sections. The enhanced efficacy of the tyramide signal amplification method was verified through IHC-IF analysis of human placental sections (Supplementary Fig. S12c). Absent 5-HT$_{2B}$ receptor expression in human β cells was consistently demonstrated, regardless of donor pregnancy status. Colocalisation analysis of whole FFPE sections further validated the lack of overlap between the 5-HT$_{2B}$ receptor and insulin across entire tissue sections even with the use of tyramide signal amplification of the 5-HT$_{2B}$ receptor signal (Supp. Fig. S13).

### Increase abundance of GLP-1 in human α cells during pregnancy

In addition to glucagon, GLP-1 is also produced by α cells. GLP-1 enhances insulin secretion from human β cells and promotes β-cell mass expansion in mice and humans[32,33]. GLP-1 produced by α cells is biologically active and, alongside glucagon, enhances insulin secretion through intra-islet (paracrine) signalling[34]. Given the significant increase in α-cell area that we observed in islets from pregnant women, we sought to determine whether GLP-1 was more abundant in α cells of pregnant women. We used an anti-GLP-1 antibody (8G9) (Abcam, cat. no. ab26278), which had previously been shown to be sensitive and specific for GLP-1, without cross-reactivity to glucagon[35].

GLP-1 was detected within α cells of both pregnant and non-pregnant women (Fig. 9a). Using the computational pipeline, we successfully segmented the GLP-1 signal to quantify the area and intensity of GLP-1 signal within α cells (Fig. 9b and c).

Quantitative measures in pregnant women were compared to non-pregnant controls, the fractional area of GLP-1 positive α cells (as a proportion of total tissue area) was increased 2.9-fold in pregnant women ($0.6943 \pm 0.1417\%$ vs. $0.2386 \pm 0.08854\%$, $p = 0.0184$) (Fig. 9d). Similarly, the average GLP-1 positive area per islet was 3.1-fold greater

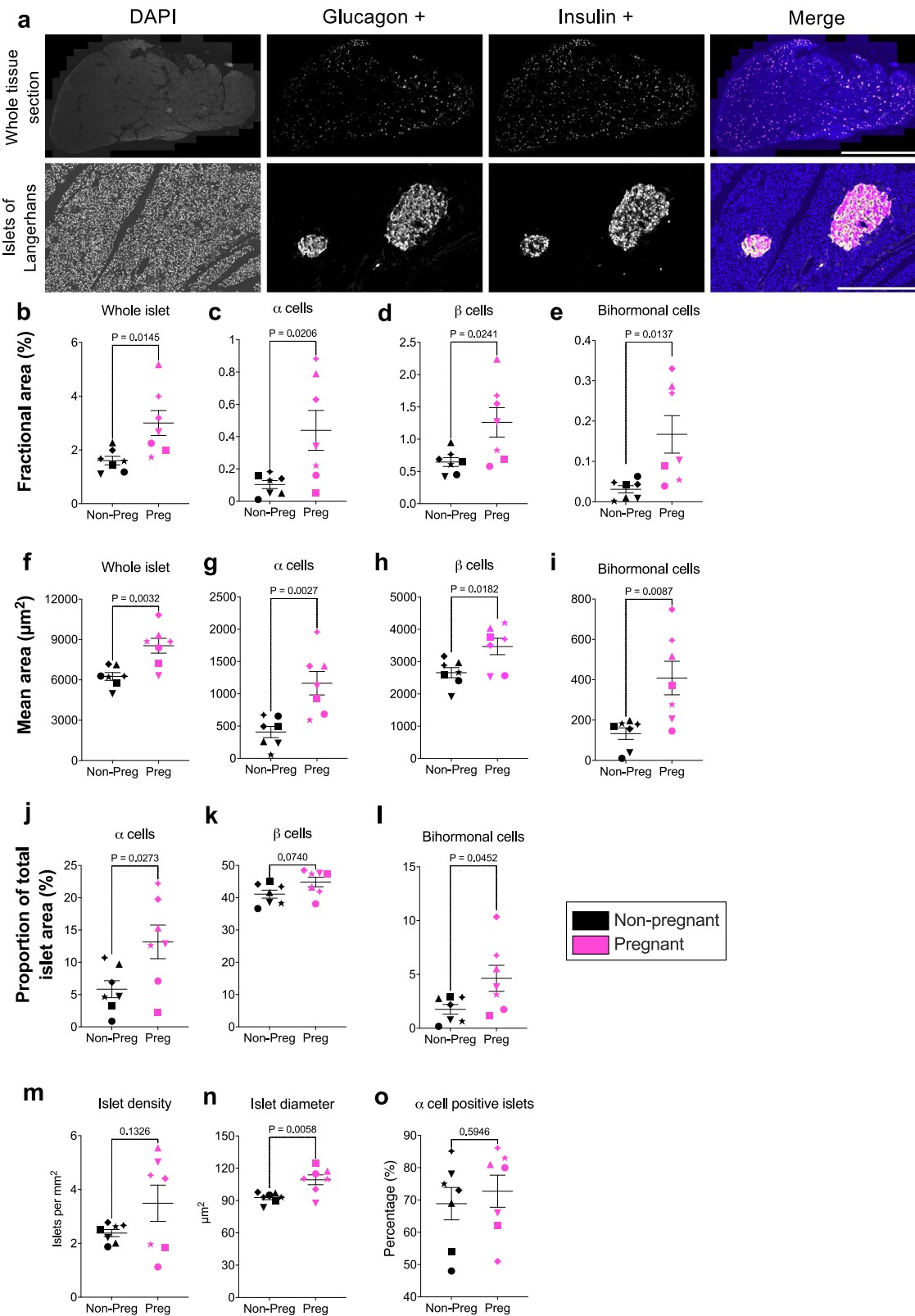

in pregnant women (1420 ± 244 µm² vs. 460.6 ± 147.8 µm², $p = 0.0056$) (Fig. 9e). GLP-1 positive area increased by 2.78-fold in proportion to the total islet area in pregnant women (15.55 ± 1.352% vs. 5.6 ± 1.484%, $p = 0.0003$) (Fig. 9f) and when normalised to α cell area, GLP-1 positive area increased 2.39-fold in pregnant women (52.69 ± 5.431% vs. 22.02 ± 2.546%, $p = 0.0003$) (Fig. 9g). The intensity of GLP-1 positive signal was 1.33-fold greater in pregnant women compared to non-

pregnant controls (488.8 ± 26.42 AU/mm² vs. 366.6 ± 23.88 AU/mm²; $p = 0.005$) (Fig. 9h). These data suggest that GLP-1 abundance increases in α cells during pregnancy, at a rate proportionally greater than the observed increases in islet and α cell size.

No significant differences in GLP-1 abundance or signal intensity were observed between donors with GDM and normal pregnancies (Supp. Fig. S14).

**Fig. 4 | Immunofluorescence labelling of human pancreas tissue (IHC-IF) and quantification of islet metrics, comparing pregnant and non-pregnant groups.** **a** Formalin-fixed paraffin-embedded human pancreatic tissue labelled by IHC-IF. Grayscale images of the individual channels are shown. The merged composite image shows labelling of the nuclei (DAPI, blue), glucagon-positive α cells (yellow), and insulin-positive β cells (magenta). The images depict both the entire tissue section and an individual islet. Scale bars = 10 mm for the whole tissue section and 500 μm for the individual islet. A representative image from one donor is shown; similar results were observed across the 14 biological replicates, each representing an independent human donor. **b–o** Quantitative comparisons between pregnant and non-pregnant women for several quantified islet measures. The data includes comparisons for (**b–e**) fractional area (measured area as a percentage of total tissue area) and (**f–i**) mean area (measured area divided by number of islets per tissue section) of whole islets, α cells, and β cells, and bihormonal cells. Additionally, comparisons of the (**j–l**) proportions of α and β cells relative to whole islets (measured area as a percentage of total whole islet area) are shown. Comparisons of **m**) islet density, **n** whole islet diameter, and **o** the proportion of α cell positive islets observed in non-pregnant and pregnant women are also shown. Symbols in each figure correspond to individual donors as indicated in Supp. Data S1. The pregnant group (n = 7 biological replicates) was compared to the non-pregnant control group (n = 7 biological replicates). Each biological replicate represents an independent human donor. Data are presented as mean ± SEM. Normally distributed data were analysed using a two-sided unpaired Student's t-test; non-parametric data were analysed using a two-sided Mann–Whitney test. Exact P values for each comparison are shown in the figure. Statistical significance was defined as P < 0.05.

## Discussion

In this study, we isolated islets from FFPE human pancreatic tissue sections using LCM and characterised the proteomic profile of pancreatic islets in human pregnancy by LC-MS/MS. Additionally, we employed IHC-IF of FFPE pancreatic tissue sections from pregnant women to conduct a comprehensive non-biased analysis of whole islet, α-cell, and β-cell metrics, as well as the abundance in islets of the PRLRs, 5-HT$_{2B}$ receptors, and GLP-1 during pregnancy.

A major strength of this study is the depth of proteomic analysis achieved, marking the first time FFPE pancreatic tissue has been used to successfully isolate and analyse islet proteins using LC-MS/MS. Techniques for identifying proteins in FFPE tissue by LC-MS/MS have advanced, even in the context of LCM, where only a small sample quantity may be available for analysis[27,28]. Previous studies using fresh-frozen pancreas tissue for LC-MS/MS have detected 1000–2000 proteins in islets from human donors with type 1 diabetes mellitus[36,37] whilst the Human Protein Atlas has identified 3708 proteins within the pancreas[38]. A spatial proteomics analysis of fresh-frozen human pancreas tissue identified over 6000 proteins from a single healthy donor[39]. A recent study by *Kolic* et al., analysing fresh-frozen islets from 90 donors by LC-MS/MS, both with and without type 2 diabetes mellitus, identified an average of 7877 proteins[40]. Even with the use of FFPE pancreatic tissue, we successfully detected close to 8000 unique proteins per sample, emphasising the capability of FFPE tissues for deep proteomic studies and identifying the highest number of proteins identified from FFPE human pancreatic islets to date.

A potential concern of using FFPE islets for proteomic analysis is that protein detectability may be impacted. To assess the reliability of our dataset, we compared our dataset with that obtained by *Kolic* et al., who analysed fresh islets by LC-MS/MS. We found that 88% of the proteins detected in our dataset were also identified in *Kolic* et al.'s study, demonstrating strong consistency between FFPE-derived and fresh islet proteomes. Notably, *Kolic* et al. also did not detect the PRLR or the 5-HT$_{2B}$ receptor, suggesting that the absence of these proteins in our dataset may not be solely due to FFPE preservation but could also reflect inherent challenges in detecting these proteins using mass spectrometry or their low abundance in human islets. Future studies employing targeted proteomics approaches may help to further elucidate the regulation and functional relevance of these proteins in human islets during pregnancy.

Among the four differentially expressed proteins identified in the islets of pregnant women, each is associated with cellular functions that may influence pregnancy associated islet adaptations. CTSZ, is involved in immune processes and promotes pancreatic neuroendocrine tumour (PNET) tumorigenesis through heterotypic cell signalling[41]. CDK5 enhances β-cell survival and proliferation, regulates insulin gene transcription, promotes insulin secretion, and protects β cells[42–46]. Laminins, such as LAMA4, are associated with β-cell proliferation and insulin secretion[47]. Further investigation is needed to clarify the mechanisms by which these proteins may contribute to human islet adaptations during pregnancy.

Compared to studies on mouse islets, we identified a limited number of differentially expressed proteins in islets of pregnant women. This difference could be attributed to the more relaxed statistical thresholds used in the mouse study, as *Horn* et al., identified 427 differentially expressed proteins (log2 FC > 0.2, FDR < 0.15) in pregnant mouse islets. In an exploratory analysis using a more permissive threshold (FDR < 0.15), the same FDR threshold used in the study by *Horn* et al., additional differentially expressed proteins were identified. However, none of these overlapped with those reported in the mouse study. This difference may be attributed to several factors, such as: differences in islet isolation methods, as *Horn* et al. isolated islets using Liberase TL/DNase I enzyme digestion, followed by filtration and snap freezing, rather than collection via LCM. Species-specific variations, as well as inherent differences between human and mouse pregnancy and islet biology, may also contribute to the observed discrepancy. While human islets exhibit significant functional adaptation, as demonstrated by their ability to double insulin secretion by late gestation[6], our findings demonstrate that human islets undergo few proteomic changes during pregnancy. However, we acknowledge that dynamic proteomic changes, which could contribute to islet adaptations during pregnancy, may remain undetected as our study does not include data on protein flux.

The increases in fractional and mean β-cell area observed during pregnancy are consistent with previous reports in human pregnancy[15,16]. In addition, we also report an increase in α-cell area during human pregnancy. Notably, this expansion in β- cell area is driven by increased cell numbers rather than hypertrophy. Three mechanisms for increasing β cell numbers are commonly proposed in the literature: increased proliferation, neogenesis, and transdifferentiation from one islet cell type to β cells[4,5,15].

In our study, we did not detect evidence of β cell proliferation. However, β cells are noted to proliferate in early- to mid-gestation[4,5], while our analysis was limited to samples from late gestation, with only a single donor from early pregnancy. Although β cell proliferation in humans remains a controversial topic due to the inherently low proliferative potential of β cells in human islets[20,48], further studies examining earlier gestational stages in human pancreas samples are necessary before ruling out proliferation as a mechanism for increased cell numbers.

*Butler* et al., analysed post-mortem pancreas samples from pregnant women, most of whom were in early to mid-gestation, and proposed that as β cell numbers increase due to an increase in islet density the rise in β cell mass occurs as a result of neogenesis[15]. However, unlike *Butler* et al., we did not observe a significant difference in islet density, which does not support the occurrence of islet neogenesis in our study. This discrepancy may stem from differences in methodology. *Butler* et al. defined a cluster of four or more insulin-positive cells as an islet and quantified islet density within randomly selected areas of the tissue, whereas our study involved analysis of the entire tissue section, employing stringent criteria to exclude smaller islets and minimise non-specific labelling[15].

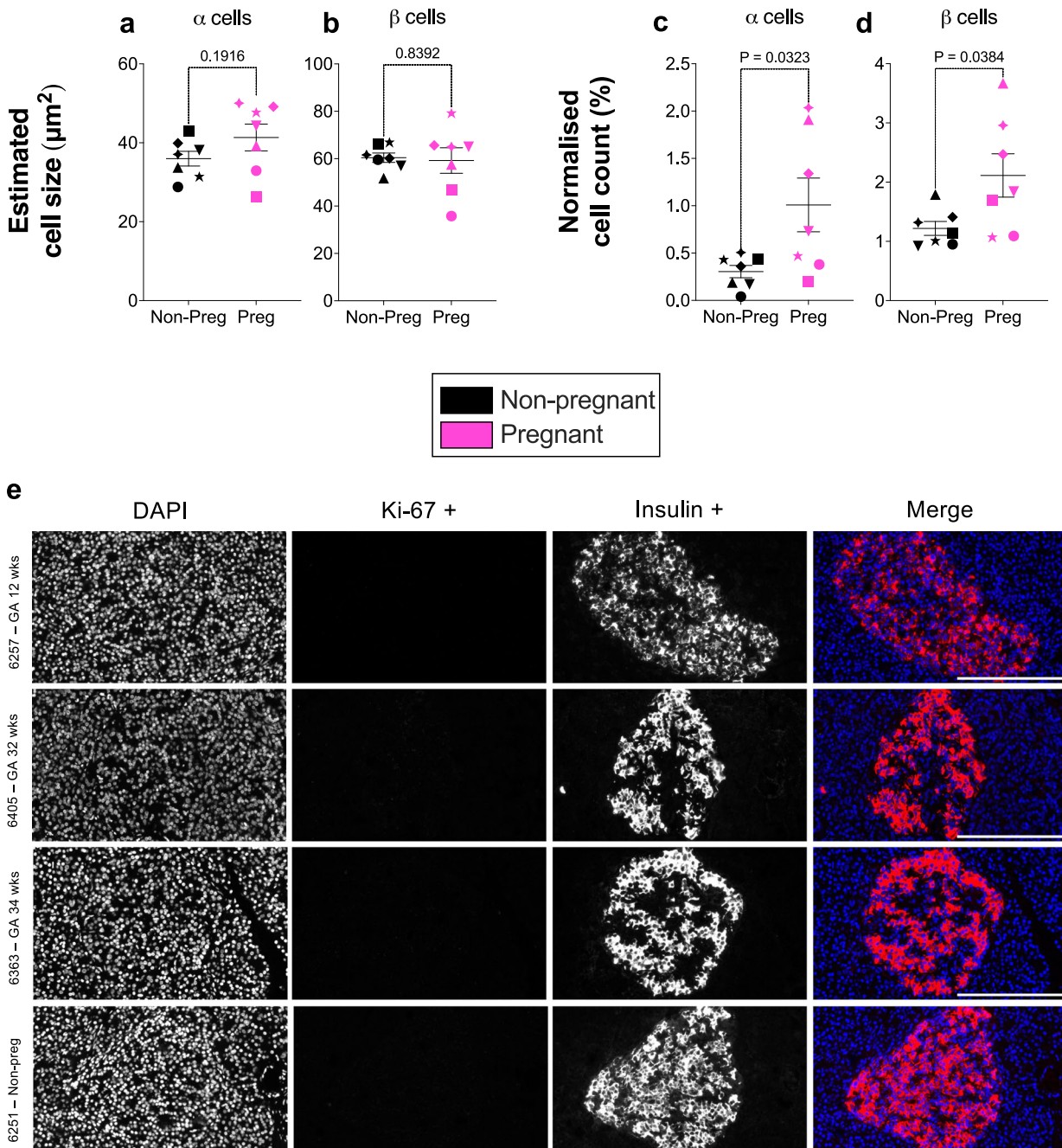

**Fig. 5 | Estimated cell size, normalised cell count and immunofluorescence (IHC-IF) for Ki-67. a**, **b** Measurements of estimated α-cell and β-cell sizes, as well as (**c**, **d**) normalised α-cell and β-cell counts compared between pregnant and non-pregnant women. Symbols in each figure correspond to individual donors as indicated in Supp. Data S1. The pregnant group ($n = 7$ biological replicates) was compared to the non-pregnant control group ($n = 7$ biological replicates). Each biological replicate represents an independent human donor. Data are presented as mean ± SEM. Normally distributed data were analysed using a two-sided unpaired Student's t-test; non-parametric data were analysed using a two-sided Mann–Whitney test. Exact $P$ values for each comparison are shown in the figure. Statistical significance was defined as $P < 0.05$. **e** Anti-Ki-67 labelling of human pancreatic sections from pregnant women at different gestational ages and a non-pregnant control sample. No Ki-67 signal is detected. The grayscale images represent the individual channels showing the nuclei (DAPI), Ki-67+, and insulin + channels with a composite merged colour image provided (nuclei in blue, Ki-67 in cyan, and insulin in red). Scale bar = 200 µm. Representative images from four donors are shown; similar results were observed across the 14 biological replicates, each representing an independent human donor. *GA* gestational age, *wks* weeks, *non-preg* non-pregnant.

The potential for transdifferentiation from α cells to β cells has also been described in human islets[49–51]. In our study, we observed bihormonal cells within islets and noted an increase in both their fractional and mean areas during pregnancy, suggesting that α-to-β cell transdifferentiation may contribute to islet plasticity. However, this interpretation is limited by the resolution of the imaging

techniques used. Without the application of super-resolution microscopy, it is challenging to distinguish individual bihormonal cells from closely juxtaposed α and β cells. Additionally, we did not observe a disproportionately greater increase in β cells compared to α cells, which would be expected if transdifferentiation were a significant contributing mechanism.

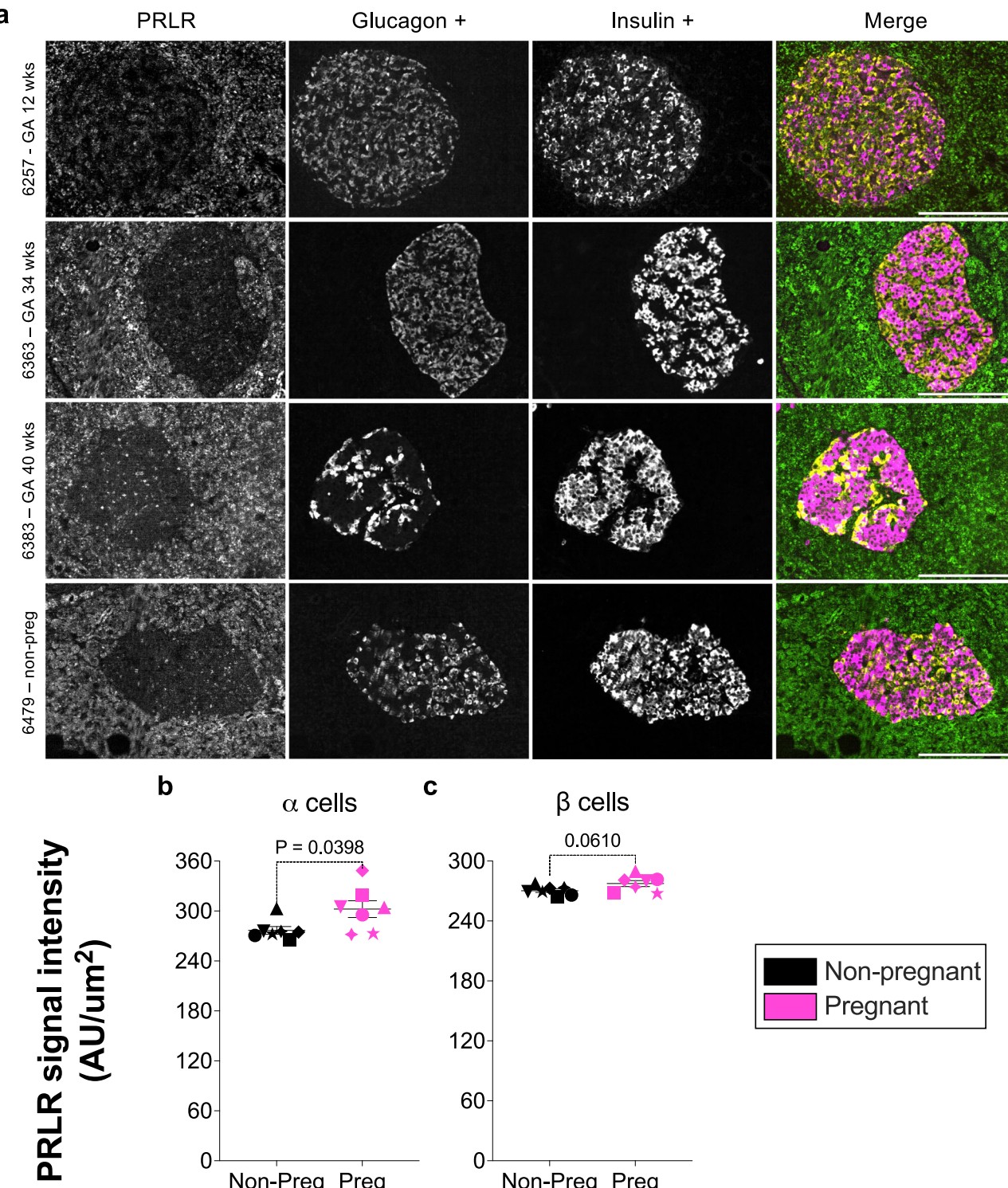

**Fig. 6 | Detection of the prolactin receptor (PRLR) in human pancreatic islets and quantification of PRLR expression in pancreatic α and β cells during pregnancy. a** Immunofluorescence analysis of human pancreatic sections (IHC-IF) from pregnant women at different gestational ages and a non-pregnant control sample. The PRLR signal is observed within α cells and β cells. The grayscale images represent the individual channels for PRLR, glucagon, and insulin. PRLR (green), glucagon (yellow), and insulin (magenta) are shown in the merged composite image. Scale bar = 200 μm. Representative images from four donors are shown; similar results were observed across the 14 biological replicates, each representing an independent human donor. **b, c** Signal intensity of the PRLR detected in α and β

cells of pregnant women compared to non-pregnant controls. Symbols in each figure correspond to individual donors as indicated in Supp. Data S1. The pregnant group (*n* = 7 biological replicates) was compared to the non-pregnant control group (*n* = 7 biological replicates). Each biological replicate represents an independent human donor. Data are presented as mean ± SEM. Normally distributed data were analysed using a two-sided unpaired Student's t-test; non-parametric data were analysed using a two-sided Mann–Whitney test. Exact *P* values for each comparison are shown in the figure. Statistical significance was defined as *P* < 0.05. *GA* gestational age, *wks* weeks, *non-preg* non-pregnant.

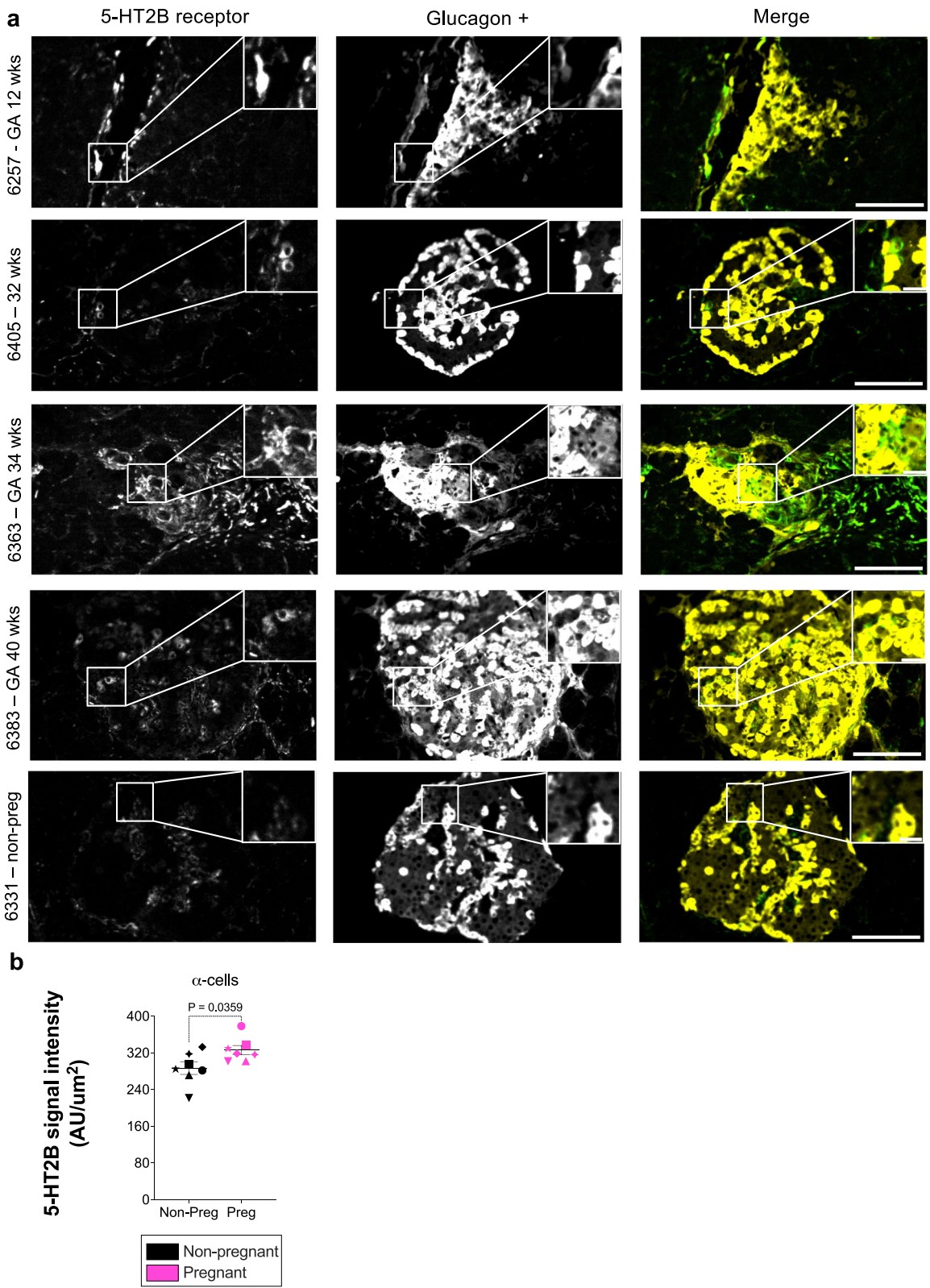

**b** α-cells

P = 0.0359

5-HT2B signal intensity (AU/um²)

Non-Preg    Preg

■ Non-pregnant
■ Pregnant

Taken together, our findings are consistent with previous studies showing that β-cell mass increases during pregnancy based on an increase in cell number. However, the underlying mechanisms remain incompletely understood. Further investigation using high-quality human pancreas samples from earlier gestational stages is critical to elucidate the contributions of proliferation, neogenesis, and transdifferentiation to islet plasticity during pregnancy.

We report an increase in α-cell area during human pregnancy. This finding aligns with studies by *Qiao* et al. and *Quesada-Candela* et al., which observed increases in α cell mass during mouse pregnancy[34,52]. Notably, we observed a more pronounced increase in α-cell area relative to β-cell area during pregnancy, contrasting with mouse models where a proportional increase in α and β cell area are reported[52]. In mouse models, the expansion of α-cell mass during pregnancy was attributed to cell proliferation rather than neogenesis

**Fig. 7 | Upregulation of the serotonin 2B (5-HT2B) receptor in pancreatic α cells during pregnancy. a** Immunofluorescence of human pancreatic sections (IHC-IF) from pregnant women at different gestational ages and a non-pregnant control sample. The 5-HT$_{2B}$ receptor signal is observed in α cells, overlapping with glucagon. Both α cells surrounding pancreatic ducts and those within islets are shown. The grayscale images represent the individual channels for the 5-HT$_{2B}$ receptor and glucagon. The composite merged image displays the co-localisation of the 5-HT$_{2B}$ receptor (green) and glucagon (yellow). Scale bar = 200 μm for the image overview and 20 μm for the magnified area in the inset. Representative images from five donors are shown; similar results were observed across the 14 biological replicates, each representing an independent human donor. **b** Signal intensity of the 5-HT$_{2B}$ receptor detected in α cells of pregnant women compared to non-pregnant controls. Symbols in each figure correspond to individual donors as indicated in Supp. Data S1. The pregnant group (n = 7 biological replicates) was compared to the non-pregnant control group (*n* = 7 biological replicates). Each biological replicate represents an independent human donor. Data are presented as mean ± SEM. Normally distributed data were analysed using a two-sided unpaired Student's t-test; non-parametric data were analysed using a two-sided Mann–Whitney test. Exact *P* values for each comparison are shown in the figure. Statistical significance was defined as *P* < 0.05. *GA* gestational age, *wks* weeks, *non-preg* non-pregnant.

or transdifferentiation[34,52]. However, we did not observe evidence of α-cell proliferation during pregnancy in our study. This may be due to our focus on pancreas tissue from late-gestation, whereas the proliferation of α cells in mouse studies was predominantly observed during early to mid-gestation.

*Qiao* et al., demonstrate that GLP-1 from α cells plays an essential role in glucose-homoeostasis during mouse pregnancy[34]. They showed that total pancreatic GLP-1 abundance increases significantly during pregnancy in mice. Furthermore, α cell ablation in pregnant mice impaired glucose-stimulated insulin secretion (GSIS), resulting in hypoinsulinemia and consequent disruptions in maternal glucose metabolism. Remarkably, treatment with GLP-1 restored GSIS in α-cell ablated pregnant mice. When GLP-1 receptor and glucagon receptor antagonist were added to islets from pregnant mice, GSIS was attenuated only in the presence of a GLP-1 receptor antagonist. The addition of a glucagon receptor antagonist, in contrast, had no effect[34]. These findings underscore the critical role of GLP-1, rather than glucagon, in supporting β-cell function and glucose regulation during pregnancy. It has been shown that, similar to glucagon, human α cells also produce and secrete biologically active GLP-1, which acts locally on adjacent β cells to potentiate GSIS[3,35,53].

During pregnancy, the increased demand for insulin to maintain glucose homoeostasis presents a significant metabolic challenge. Despite this, total islet insulin levels remain unchanged in our proteomic analysis, suggesting that β cells adapt by enhancing their production and secretion rates without accumulating excess insulin stores. Such an enhanced functional output likely relies on regulatory signals. Like *Qiao* et al., we too observe an increase in GLP-1 abundance in α cells during human pregnancy and propose that this increase in GLP-1 may play a role in supporting β-cell function.

Human islets are anatomically uniquely designed to facilitate paracrine communication, as the different hormone-secreting cells within islets are more intermixed than in mice[20–23]. For example, unlike the organised islet structure in mice where just 28% of β cells have heterotypic contacts (direct interactions/connections between different cell types), in human islets up to 80% of β cells are in direct contact with α cells and may even partially envelop them[22]. Additionally, human islets have a greater proportion of α cells, and these α cells secrete significantly more GLP-1 than mouse α cells[22,23]. A recent study using advanced 3D imaging demonstrated that approximately 50% of human insulin-expressing islets lack α cells, although these islets constitute only 16% of the total islet volume[54]. This indicates that the vast majority of β cells still maintained either direct or indirect contact with α cells, underscoring the relevance of intra-islet communication. In our study, we observed that 70% of islets contained α cells, all be it compared to the study by *Lehrstrand* et al. we studied sections only from the pancreas tail, a smaller region of interest, and performed a 2D analysis[54].

It is well described that GLP-1 enhances insulin biosynthesis, secretion, and β-cell responsiveness[55,56]. In doing so, GLP-1 may also optimise the utilisation of stored insulin and stimulate the translational machinery necessary to support the increased insulin requirements of pregnancy. We propose that GLP-1 facilitates these adaptations through paracrine signalling within the islet microenvironment, enabling β cells to meet the heightened metabolic demands of pregnancy while maintaining steady insulin content. This underscores the potential role of GLP-1 as a key mediator of β-cell adaptation observed during human pregnancy.

The increased α-cell mass we observe during human pregnancy may enhance the production and secretion of GLP-1 to support β-cell function through paracrine pathways. Importantly, increased islet glucagon production and secretion is unlikely to increase circulating glucagon levels in human pregnancy, as plasma glucagon levels, unlike insulin, rise only intermittently during pregnancy[34,57]. Even mouse studies note low circulating and intra-islet glucagon in late gestation and no change in glucagon content or gene expression[34,52]. These observations suggest that during pregnancy glucagon secretion from α cells may not play a key role in islet adaptations, however, the contribution of glucagon on intra-islet signalling in human pregnancy cannot be ruled out and requires further study[2,22,23].

The physiological adaptations of pregnancy impose significant metabolic challenges, necessitating coordinated islet function. The observed α-cell expansion during pregnancy likely reflects a broader role played by α cells in maintaining maternal glucose homoeostasis and adapting to metabolic alterations in pregnancy. Our findings align with evidence from the literature in mice, providing reassurance that the observed adaptations in α cells in mice models are conserved in human pregnancy and may contribute to pregnancy-associated metabolic changes. This concordance supports the need for mechanistic studies to investigate the effects of enhanced α-cell mass and mediators of paracrine communication, such as GLP-1, on islets during human pregnancy. These studies would help clarify the role of α cells in maintaining metabolic balance during this unique physiological state.

In contrast to findings in mouse models, we did not observe upregulation of the PRLR in β cells during pregnancy[4,58–60]. This may indicate that the PRLR may not be a critical factor in human β-cell adaptation. However, it can be argued that in the context of higher circulating levels of lactogenic hormones receptor upregulation may not be necessary to confer biological effects. Furthermore, it would be expected that higher circulating lactogen levels would lead to receptor down regulation and hence unchanged β cell PRLR abundance reflects a relative increase. We observed an increased abundance of the PRLR in α cells, interestingly in α-TC1.9 cells lactogens stimulate cell proliferation and inhibit glucagon secretion at low glucose levels[52]. Thus, it remains plausible that lactogenic hormones exert biologically significant effects via PRLR activation in both α and β cells during human pregnancy, warranting further investigation into their role.

Mouse studies have demonstrated that 5-HT$_{2B}$ receptor signalling is crucial for β-cell proliferation and to augment insulin secretion[11,61,62]. However, in contrast, a study using tryptophan hydroxylase 1—the enzyme responsible for peripheral serotonin synthesis—and 5-HT$_{2B}$ receptor knockout mice found that neither serotonin nor the 5-HT$_{2B}$ receptor is required for β-cell proliferation during pregnancy in mice[63]. We report that 5-HT$_{2B}$ receptor expression is absent in human β cells in early and late pregnancy as well as in non-pregnant women.

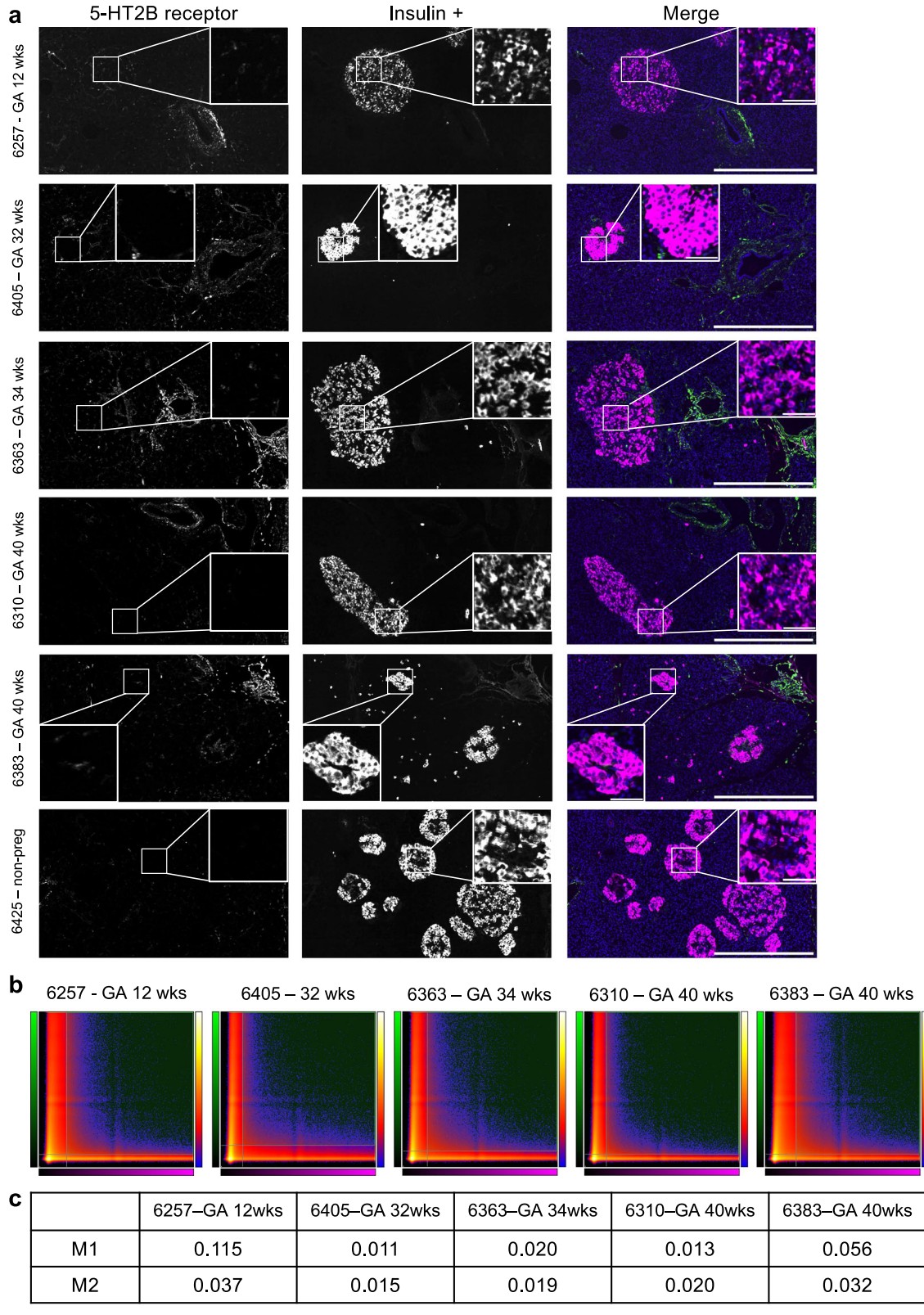

**c**

|      | 6257–GA 12wks | 6405–GA 32wks | 6363–GA 34wks | 6310–GA 40wks | 6383–GA 40wks |
|------|---------------|---------------|---------------|---------------|---------------|
| M1   | 0.115         | 0.011         | 0.020         | 0.013         | 0.056         |
| M2   | 0.037         | 0.015         | 0.019         | 0.020         | 0.032         |

Importantly, the detection of 5-HT$_{2B}$ receptor signal in adjacent pancreatic ducts, which express the 5-HT$_{2B}$ receptor[31], validates the functionality of the antibody we used under our experimental conditions. This data suggests that the 5-HT$_{2B}$ receptor does not play a role in human pregnancy associated β-cell alterations. 5-HT$_{2B}$ receptor expression in human islets has been previously demonstrated using a polyclonal antibody[62]. Given that polyclonal antibodies are raised against multiple epitopes, their limitations in antigen specificity are

well known[64]. To overcome this, we used a monoclonal antibody clone, which offers greater specificity due to its recognition of a single epitope. Additionally, we performed thorough antibody validation, including immunoblotting of recombinant 5-HT$_{2B}$ receptor protein and cell lysates known to express or lack 5-HT$_{2B}$ receptor expression. Validation also included IHC-IF analysis of HAP1 cell pellets and cancerous tissues known to express the 5-HT$_{2B}$ receptor, confirming that the antibody reliably detects the target protein. Differences in

**Fig. 8 | Absence of serotonin 2B (5-HT$_{2B}$) receptor in insulin-producing pancreatic β cells during pregnancy. a** Immunofluorescence of human pancreatic sections (IHC-IF) from pregnant women at different gestational ages and a non-pregnant control sample. No staining of the 5-HT$_{2B}$ receptor is observed within β cells; however, 5-HT$_{2B}$ receptor signal is evident within pancreatic ducts. The grayscale images represent the individual channels for the 5-HT$_{2B}$ receptor and insulin. The composite merged image displays the 5-HT$_{2B}$ receptor (green) and insulin (magenta) channels. Scale bar = 500 μm for the image overview and 50 μm for the magnified area in the inset. Representative images from five donors are shown; similar results were observed across the 14 biological replicates, each representing an independent human donor. **b** Colocalisation analysis demonstrating no colocalisation between the 5-HT$_{2B}$ receptor and insulin signals. Scatter plots showing pixel intensity from the 5-HT$_{2B}$ receptor and insulin channels plotted against each other are presented. Pixel intensity for each channel is plotted on the respective axes: the 5-HT$_{2B}$ receptor channel is represented in green on the Y-axis, and the insulin channel is represented in magenta on the X-axis. Whole tissue sections from pregnant women at different gestational ages were analysed. Scatter plots from five donors are shown; similar results were observed across the 14 biological replicates, each representing an independent human donor. **c** Mander's overlap coefficients are shown in the table for each whole tissue section, where M1 represents the overlap of the insulin signal over the 5-HT$_{2B}$ receptor signal, and M2 represents the 5-HT$_{2B}$ receptor signal overlapping the insulin signal. Mander's overlap coefficients from five donors are shown; similar results were observed across the 14 biological replicates, each representing an independent human donor. * *GA* gestational age, *wks* weeks, *non-preg* non-pregnant.

antibody specificity and validation methods between studies may account for the observed discrepancy. Furthermore, variations in tissue processing, staining protocols, or experimental conditions could also contribute to differences in reported 5-HT$_{2B}$ receptor expression. Our approach aimed to minimise such variability, providing robust evidence for our findings.

We observed greater 5-HT$_{2B}$ receptor expression in α cells in pregnant women and *5-HTR2B* RNA is detected in human α cells[65]. In human islets from non-pregnant donors, serotonin, that is produced by β cells in response to hyperglycaemia, acts in a paracrine manner on α-cell serotonin 1 F receptors (5-HT$_{1F}$) receptors to reduce glucagon secretion[65]. 5-HT$_{1F}$ receptors are coupled to the inhibitory G-protein (Gi) to inhibit adenyl cyclase and cAMP production, while the 5-HT$_{2B}$ receptors are coupled to the stimulatory G-protein (Gq) to activate phospholipase C to increase intracellular calcium signalling and cell replication via the mitogen-activated protein kinase pathway[66]. Although we did not detect markers of cell proliferation in α cells, it is noteworthy that calcium plays a key role in glucagon secretion[3,67]. We postulate that the paracrine signalling of the 5-HT$_{2B}$ receptor on α cells by serotonin may stimulate glucagon release. Additionally, we also note an increase in PRLR expression in α cells. In mouse α cells, PRLR activation decreases glucagon release under low glucose conditions[52]. As both 5-HT$_{2B}$ receptor and PRLR signalling can influence α cell function, further mechanistic studies are needed to clarify their roles in human α cells during pregnancy.

We did not detect differences in islet metrics, PRLR, 5-HT$_{2B}$ receptor expression, GLP-1 abundance, and overall protein abundance between islets from normal and GDM pregnancies. Clinical data show no significant differences in first-phase insulin response or insulin secretion throughout pregnancy between women with GDM and those without GDM who have no risk factors for hyperglycaemia. However, when comparing pregnant women with GDM to pregnant women without diabetes mellitus but with risk factors for hyperglycaemia, GDM was associated with reduced insulin secretion[6,68]. Recently it has been recognised that the pathophysiology of GDM is heterogeneous, comprising insulin-deficient and insulin-resistant forms[8]. This suggests that GDM may have variable effects on islets, with subtle perturbations between subtypes that are not easily detectable. These could involve genetic changes, post-translational modifications, or external factors like oxidative stress. All of these factors may contribute to islet cell dysfunction in GDM, potentially explaining the lack of detectable differences in the islet proteome[17]. Further work is needed in this area to clarify the molecular mechanisms that contribute to islet dysfunction in GDM.

Together, our findings suggest that pregnancy related adaptations in human β cells differ to those described in mice. While β-cell area increases in pregnancy, it remains uncertain whether this alone can account for the significant rise in insulin secretion seen in pregnancy, although, it is well-established that β cells store substantial amounts of insulin and secrete only a small fraction in response to hyperglycaemia[69,70]. We propose that, during human pregnancy, the insulin secretion capacity of β cells may be enhanced by mechanisms mediated by α cells. This could explain the relatively larger increase in α-cell compared to β-cell mass observed during pregnancy, as a greater α-cell population size is required to influence β cells. This may occur through paracrine signalling, whereby glucagon or GLP-1 from neighbouring α cells stimulate β cells to enhance insulin release. Our observations reflect the plasticity of α and β cells to enhance insulin output in human pregnancy through subtle intra-islet environmental changes, rather than extensive remodelling as is observed in murine pregnancy. Further functional islet studies are required to clarify the role of α cells during pregnancy, particularly their role in paracrine signalling through GLP-1.

The limited availability of high-quality pancreatic tissue from pregnant women significantly constrained the sample size in this study and introduced heterogeneity within the cohort. Variability in ethnicity, GDM status, anti-GAD antibody status, and insulinitis among donors may have contributed to the few observed differences in proteomic profiles and may have masked more subtle biological patterns. For pregnant donors, only the pregnancy BMI was available. Due to the circumstances under which donor tissues are obtained, it is not possible to acquire pre-pregnancy BMI measurements, and this limitation complicates the interpretation of BMI-related effects on islet adaptations during pregnancy. Furthermore, when comparing GDM to normal pregnancy, it is essential to interpret the data with caution due to the limited sample size. While the findings provide valuable preliminary insights, larger cohorts are needed to validate these observations and draw more definitive conclusions regarding the impact of GDM on islet adaptations during pregnancy. The heterogeneity of the cohort highlights a broader challenge in the field of pregnancy-related islet adaptions. Greater collaboration between researchers, to increase the availability of pancreatic tissue from pregnant women, in order to study more homogenous cohorts is critical to advancing our understanding of islet adaptations during pregnancy and improving outcomes in GDM research. Future research should focus on studying pancreatic tissue from different stages of gestation; however, high-quality first- and second-trimester pancreas tissue were not available for this study. Measures of islet diameter are limited by the irregular shapes of islets, making islet area a more reliable metric for measurement. Prolactin can also bind to the pancreatic growth hormone receptor (GHR); however, restricted availability of islet tissue prevented assessment of GHR expression, representing a limitation of this study. We acknowledge the limitation of not exploring additional serotonin receptor isoforms due to challenges with antibody specificity. Despite efforts to validate commercial antibodies for receptors such as the serotonin 1D, 5-HT$_{1F}$, and serotonin 3 receptors, high sequence similarity among subtypes hindered reliable analysis. Future studies employing alternative approaches may help address this gap. The use of immunofluorescence with islet-specific markers, such as insulin and glucagon, would improve islet isolation by LCM rather than relying on the use of brightfield and islet autofluorescence. While we attempted these methods heat-mediated antigen retrieval disrupted

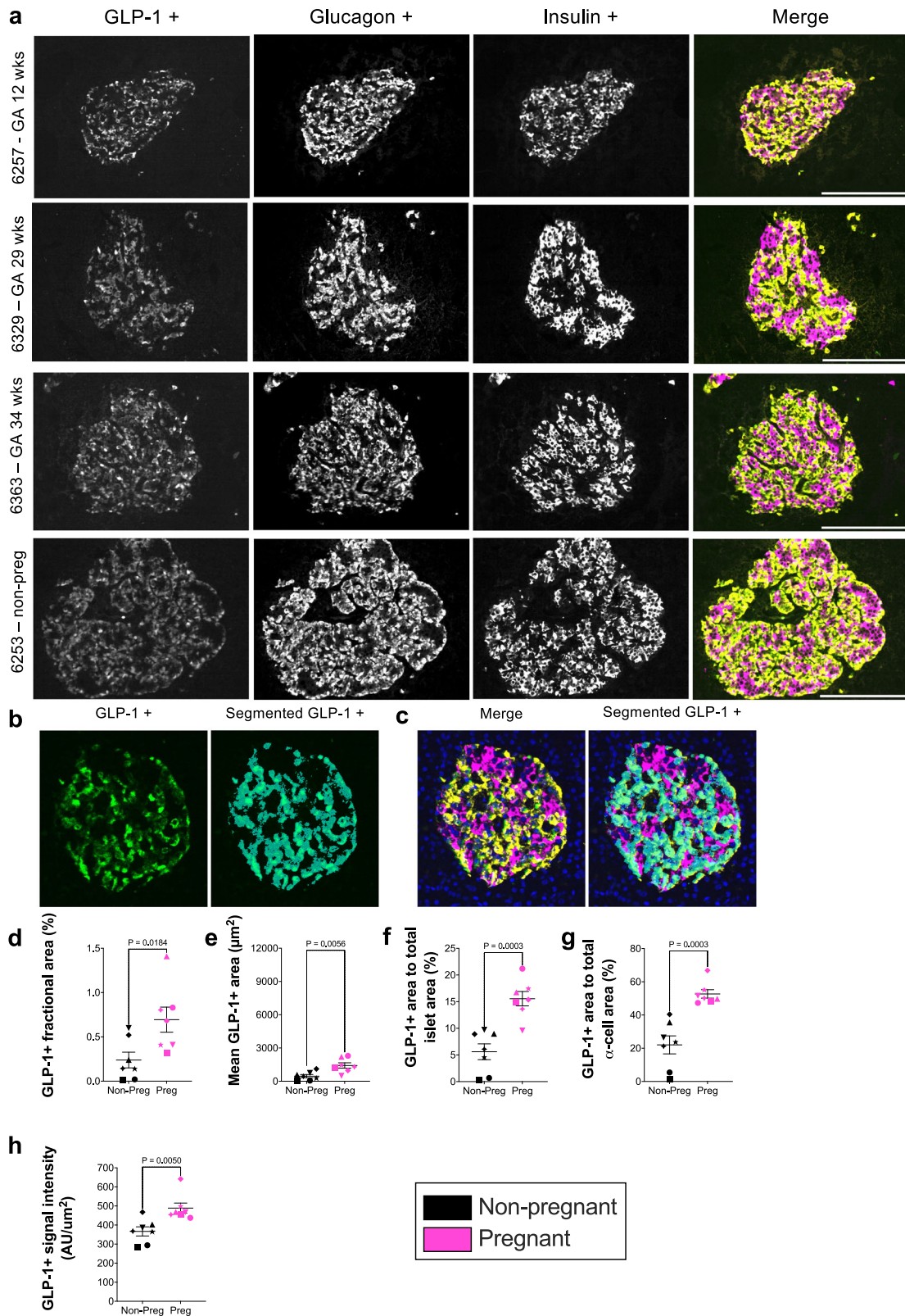

the microscope slide membrane-coating which hindered tissue collection. Efforts to minimise membrane disruption after antigen retrieval were unsuccessful[71], and further optimisation of these methods is needed for future studies.

In conclusion, we successfully characterised the proteome of human pancreatic islets during pregnancy, creating the largest dataset of pancreatic islet and exocrine proteins derived from pancreata from pregnant women to date. Despite the highly dynamic state of pregnancy, we observed few changes in islet protein abundance during pregnancy. Using a non-biased computational pipeline, we quantitatively measured α- and β-cell areas demonstrating increases in fractional and mean areas due to an increase in cell number. PRLR, 5-HT$_{2B}$ receptor expression, and GLP-1 abundance increased in α cells during pregnancy; however, in β cells, PRLR

**Fig. 9 | Upregulation of glucagon-like peptide-1 (GLP-1) expression in pancreatic α cells during pregnancy. a** Immunofluorescence of human pancreatic sections (IHC-IF) from pregnant women at different gestational ages and a non-pregnant control sample. The GLP-1 signal is observed in α cells, overlapping with glucagon. Both α cells surrounding pancreatic ducts and those within islets are shown. The grayscale images represent the individual channels for the GLP-1, glucagon and insulin. GLP-1 (green), glucagon (yellow), and insulin (magenta) are shown in the composite merged image. Scale bar = 200 μm. Representative images from four donors are shown; similar results were observed across the 14 biological replicates, each representing an independent human donor. **b**, **c** Images demonstrating the use of the imaging pipeline to quantify GLP-1 area and signal intensity. The labelling of GLP-1 labelled α cells by IHC-IF was followed by thresholding of the GLP-1 positive signal. An example of the thresholded region (cyan) identifying GLP-1 positive signal is shown for the (**b**) GLP-1 channel (green) and (**c**) composite merged image (GLP-1 [green], glucagon [yellow] and insulin [magenta]). **d**–**h** Quantitative comparisons of GLP-1 between pregnant and non-pregnant women for several quantified islet measures. The data includes comparisons for **d**) fractional area (measured area as a percentage of total tissue area) and (**e**) mean area (measured area divided by number of islets per tissue section). Additionally, comparisons of the proportions of GLP-1 positive area relative to (**f**) whole islets (measured area as a percentage of total whole islet area) and **h**) (**g**) relative to α cell area (measured area as a percentage of α cell area) are shown. **h** Signal intensity of GLP-1 detected in α cells of pregnant women compared to non-pregnant controls. Symbols in each figure correspond to individual donors as indicated in Supp. Data S1. The pregnant group ($n = 7$ biological replicates) was compared to the non-pregnant control group ($n = 7$ biological replicates). Each biological replicate represents an independent human donor. Data are presented as mean ± SEM. Normally distributed data were analysed using a two-sided unpaired Student's t-test; non-parametric data were analysed using a two-sided Mann–Whitney test. Exact P values for each comparison are shown in the figure. Statistical significance was defined as $P < 0.05$. * *GA* gestational age, *wks* weeks.

expression remained unchanged, and, in contrast to mouse models, the 5-HT$_{2B}$ receptor was absent. Our findings suggest that human islet adaptations during pregnancy differ to those observed in mice, however the role of lactogenic hormones and serotonin signalling cannot be excluded. The marked increases in α cell area and GLP-1 abundance suggest that β cells may have a greater reliance on intra-islet paracrine signalling during human pregnancy. This study provides data on pancreatic islet alterations during human pregnancy and underscores the importance of cautious extrapolation from small animal studies and reinforces the value of using human-based model systems or tissues for pancreatic islet research.

## Methods

### Donor tissues
The South Central – Oxford A Research Ethics Committee (REFS 18/SC/0559) approved this project. nPOD provided the FFPE pancreatic tissue sections, and corresponding donor data. Donors were de-identified according to Health Insurance Portability and Accountability Act (HIPAA) regulations, USA. FFPE pancreatic tissue sections from eight pregnant donors were available from the nPOD biorepository, seven third trimester and one first trimester donor. Written informed consent for research use of tissues was obtained from the next of kin of all donors. To ensure consistency amongst the pregnant donors, for the validity of later comparisons, the first trimester donor was excluded from between group comparisons. Pregnant donors were classified by GDM status based on clinical details provided by the nPOD biorepository. Donors were identified as GDM if they had a documented history of GDM diagnosis during the current pregnancy and had received treatment for GDM during the current pregnancy. Non-pregnant control donors were matched based on age, race, and body mass index (BMI) as closely as possible, dependent on sample availability. For pregnant donors the pregnancy BMI is provided. (Table 1 and Supp. Data S1).

### Laser-capture microdissection
Membrane-covered microscope slides (Carl Zeiss, cat. no. 15350731) and the Leica LMD7 were used for LCM.

Membrane-covered microscope slides (Carl Zeiss, cat. no. 15350731), as per manufacturer recommendations, were UV-irradiated (254 nm) for 30 minutes to improve tissue adherence. FFPE pancreatic tissue sections (10 μm thick) were mounted onto the membrane-covered slides and dried overnight at 56 °C to further enhance tissue adherence. FFPE tissue sections were de-paraffinised in two washes of xylene (2 minutes each) (Sigma, cat. no. 108298), followed by washes in a graded ethanol series (Thermo Fischer Scientific, cat. no. 397690025) and double distilled water (ddH$_2$O) (100%, 90%, and 70% ethanol for 1 minute each; ddH$_2$O for 5 minutes). Slides were dried and stored at -80 °C until LCM.

The LCM parameters used were: power 45, aperture 9, speed 5, middle pulse count 2, final pulse 15, head current 100% and pulse frequency 3773 Hz. Tissue was collected into 40 μL of acetonitrile in a 96-well PCR plate (Eppendorf, cat. no. 0030129580). To ensure the tissue was located at the bottom of the wells, 110 μL of acetonitrile was added to each well, the plate was sealed, centrifuged at 2000 g for 5 minutes, dried in a vacuum centrifuge at 40 °C, and then stored at −20 °C until further processing.

Islets were identified using brightfield and fluorescence imaging (laser excitation wavelength 495 nm) at 20x magnification. For each donor, 235,000 μm$^2$ of islet tissue and 235,000 of μm$^2$ exocrine tissue, consisting primarily of acinar cells, were microdissected. The exocrine tissue served as an internal control to verify the purity of islet tissue isolation. A single tissue section was utilised for each donor, islets were selected at random, and care was taken to ensure that the dissected islets were separated by a minimum distance of at least > 200 μm, to ensure that each dissected region represented a distinct islet.

### Liquid chromatography mass spectrometry and data analysis
To minimise sample losses, a single step digestion was performed[72]. Dried samples were reconstituted in 20 μL of lysis buffer (0.013% n-Dodecyl-β-D-Maltoside (DDM, Thermo Fisher Scientific, cat. No. 89902), 50 mM triethylammonium bicarbonate (TEAB, Sigma, cat. No. 18597) in LC-MS/MS-grade water), the plate was sealed and incubated at 95 °C for 90 minutes. Samples were cooled to 20 °C and normalised to 20 μL with LC-MS/MS-grade water. Five μL of 60% acetonitrile in 50 mM TEAB was added to each sample and incubated at 75 °C for 30 minutes, cooled to 20 °C, and centrifuged at 2000 g for 1 minute. A further 4 μL of 50 mM TEAB was added to dilute the acetonitrile to 10%. One μL of 8 ng/μL Trypsin/LysC enzyme mix (Promega, cat. No. V5073) in 50 mM TEAB was added to the samples. The plate was sealed and incubated at 37 °C overnight in a thermomixer with a heated lid (Eppendorf, cat. No. 5382000031). The digestion was quenched by adding formic acid (Fischer scientific, cat. No. 10596814) to 1%.

Samples (2 μL) were loaded on Evotip Pure C18 tips (Evosep, EV2011) as per the manufacturer's protocol. Briefly, tips were rinsed with 20 μL Solvent B by centrifugation, conditioned by soaking in 1-propanol, equilibrated with 20 μL Solvent A by centrifugation, loaded with 2 μL of sample and 18 μL of solvent A, then centrifuged, washed with 20 μL solvent A by centrifugation, and wetted with 100 μL solvent A and a brief centrifugation step for 10 seconds. All centrifugation steps were 60 seconds at 800 g unless stated otherwise.

Peptides were analysed on an Evosep One LC system (EvoSep) coupled to a timsTOF Ultra mass spectrometer (Bruker) using the Whisper 40 samples-per-day method and a 75 μm × 150 mm C18 column with 1.7 μm particles and an integrated Captive Spray Emitter (IonOpticks). Buffer A was 0.1% formic acid in water and Buffer B was 0.1% formic acid in acetonitrile. Data were collected using diaPASEF[73]

with 1 MS frame and 9 diaPASEF frames per cycle with an accumulation and ramp time of 100 ms, for a total cycle time of 1.07 seconds. The diaPASEF frames were separated into 3 ion mobility windows, in total covering the 400–1000 m/z mass range with 25 m/z-wide windows between an ion mobility range of 0.64–1.4 Vs/cm$^2$. The collision energy was ramped linearly over the ion mobility range, with 20 eV applied at 0.6 Vs/cm$^2$ to 59 eV at 1.6 Vs/cm$^2$.

Mass spectrometry raw files were analysed in DIA-NN version 1.8.1[74] using an in-silico spectral library generated by DIA-NN with default settings (1 missed cleavage, N-terminal methionine excision was allowed, carbamidomethylation of cysteine was not allowed) using a Uniprot human FASTA file containing 20,383 reviewed sequences with common contaminants added. MS1 and MS2 accuracies were set to 15 ppm, all other settings were left as default. DIA-NN incorporates retention time-dependent normalisation as part of its default parameters, along with the commonly used MaxLFQ quantitation algorithm, which also includes a data normalisation step. These normalisation processes are integral to the default settings of the software and ensure consistency in the analysis[75]. The LC-MS/MS proteomics data have been deposited to the ProteomeXchange Consortium via the PRIDE[76] partner repository with the dataset identifier PXD055852.

LC-MS/MS data were analysed using Perseus (v2.0.11), a data analysis and visualisation tool for proteomics data with a comprehensive suite of functionalities for pre-processing data, conducting quality control, and performing statistical tests for differential expression[77]. Data were log2 transformed and categorised into four groups: pregnant islet, pregnant exocrine, control islet, and control exocrine. After filtering out incomplete rows per group, downstream comparisons in both the islet and exocrine groups, respectively, were performed. To illustrate the purity of islet isolation, islet groups (pregnant and control islet) were compared to exocrine tissue groups (pregnant exocrine and control exocrine). A single donor and matched control (nPOD ID 6226 and 6559) were excluded from the analysis due to evidence of exocrine contamination within islets from nPOD ID 6226. In total, 12 donors were included for analysis (Table 1 and Supp. Data S1). To identify islet-specific proteins expressed in pregnant women, we compared the pregnant islet to the pregnant exocrine group. To infer whether protein expression was altered in either islets or exocrine tissue during pregnancy, we compared the pregnant islet group to control islets and the pregnant exocrine tissue group to exocrine controls. Within the pregnant islet group, GDM was compared to normal pregnancy. Between-group comparisons were performed using permutation-based testing to correct for multiple comparisons using an FDR of < 0.05 and an S0 value of 0.01[78]. (The FDR represents the adjusted $p$-value following correction for multiple comparisons). Principal component analysis was performed using the built-in principal component analysis function. Volcano plots were generated by plotting the negative log10 of the FDR against the log2 fold change of the normalised mean MS intensities. A negative log10 FDR of >1.3 is equivalent to a FDR < 0.05.

Protein network visualisation and over-representation analysis were performed using Cytoscape (v3.10.2) in conjunction with the STRING database, which contains both known and predicted protein-protein interactions (STRING, version 2023)[79]. Physical interactions between differentially expressed proteins were used to construct interaction networks. An over-representation analysis was performed on proteins with a log2 fold change (FC) >1 and FDR < 0.05, representing islet-specific proteins. The total number of proteins detected in the experiment was used as the background proteome for enrichment analysis, and redundancy filtering applied to prioritise the most robust and biologically meaningful enriched pathways, avoiding an extensive and potentially overwhelming list.

Pathways from GO biological processes, Reactome, and KEGG with an FDR < 0.05 were deemed significant. To identify clusters of proteins with dense interconnectedness within the protein networks, protein network clusters were identified using Markov Cluster algorithm (MCL) clustering, with the clusterMaker2 app and a granularity parameter (inflation value) set to four[80]. Clustered protein networks were visualised, and networks in which three or more proteins contributed to enriched pathways examined. Enriched pathways were overlaid onto protein networks and proteins observed in an enriched pathway highlighted to provide additional insights on the contribution of each to biological processes underlying islet function in third-trimester pregnant women.

To further explore proteomic differences between islets from pregnant and non-pregnant donors, an additional exploratory analysis was conducted using a more permissive threshold (FDR < 0.15), consistent with the approach used in the proteomic study analysing islets from pregnant mice conducted by _Horn_ et al. This analysis aimed to identify potential trends in protein expression that may not have been detected under the more stringent FDR cutoff and to allow for a direct comparison with the mouse study. However, the primary conclusions of the study are based solely on the FDR < 0.05 analysis.

### Cell culture

HAP1 wild-type (WT) parental control cells (cat. no. C631, RRID: CVCL_Y019) and _PRLR_ knockout (KO) cells with a 25 bp deletion (cat. no. HZGHC003937c0061, RRID: CVCL_TH11) were purchased from Horizon Discovery. The _PRLR_-edited cell line, generated using CRISPR/Cas9 technology with the guide RNA sequence AACCCTACCTGTGGATTAAA, resulted in a 25 bp deletion in exon 6 of the _PRLR_ gene, causing frame shifts and multiple stop codons downstream.

HeLa cells (cat. no. 93021013, RRID: CVCL_0030), SH-SY5Y cells (cat. no. 94030304, RRID: CVCL_0019), MCF7 cells (cat. no. 86012803, RRID: CVCL_0031), and U2OS cells (cat. no. 92022711, RRID: CVCL_0042) were all obtained from the European Collection of Authenticated Cell Cultures via Sigma-Merck. EndoC-βH3 cells were obtained from Human Cell Design, France.

Both HAP1 WT parental control and _PRLR_ KO cell lines were maintained in Iscove's Modified Dulbecco's Medium (Sigma, cat. no. I3390) supplemented with 10% heat-inactivated foetal bovine serum (hiFBS) (Sigma, cat. no. F9665) and 1% Pen/Strep (100 U/mL penicillin and 100 μg/mL streptomycin), as per the supplier's instructions. Cells were passaged every 2-3 days upon reaching 70–75% confluence and used for experiments at approximately 70% confluence.

HeLa cells were maintained in low glucose DMEM with Gluta-MAX and pyruvate (Gibco, cat. no. 21885-108), 10% hiFBS, and 1% Pen/Strep. They were passaged every 2-3 days at a 1:4 ratio upon reaching 75–80% confluence. SH-SY5Y cells were cultured in a 1:1 mixture of DMEM and F12 medium (Gibco, cat. no. 11330057) with 10% hiFBS and no antibiotics. They were passaged every five days at a 1:5 ratio upon reaching 90% confluence. MCF7 cells were maintained in low glucose DMEM with GlutaMAX and pyruvate, 10% hiFBS, and 1% Pen/Strep. They were split every 3-4 days at a 1:3 ratio upon reaching -80% confluence. U2OS cells were maintained in McCoy's 5 A (modified) Medium (Gibco, cat. no. 16600082) with 10% hiFBS and 1% Pen/Strep. They were passaged every 3-4 days at a 1:4 ratio upon reaching 75–80% confluence.

EndoC-βH3 cells were cultured in low glucose DMEM with Gluta-MAX and pyruvate, supplemented with 2% bovine serum albumin (BSA) (Roche, cat. no. 10775835001), 10 mM nicotinamide (Sigma, cat. no. N3376), 50 μM β-2 mercaptoethanol (Gibco, cat. no. 31350010), 5.5 μg/mL transferrin (Sigma, cat. no. T8158), 6.6 ng/mL sodium selenite (Sigma, cat. no. 214485), and 1% Pen/Strep. Cells were differentiated by adding 4-hydroxitamoxifen (1 μL/10 mL) to culture media for 21 days.

All cell lines were maintained at 37 °C in a humidified incubator with air and 5% $CO_2$ and, except HAP1 cell lines, collected for experiments upon reaching 85–90% confluence.

## Sequencing of the PCR-amplified cDNA from HAP1 cells

Parental WT and *PRLR* KO HAP1 cells were lysed in TRI Reagent® (Sigma, cat. no. 9424) and RNA extracted using the Direct-Zol™ RNA Purification Kit (Zymo Research, cat. no. R2070), as per the manufacturer's protocol. cDNA was synthesised from 1 µg of RNA using the LunaScript® RT SuperMix Kit (New England Biolabs, cat. no. E3010) and amplified by PCR using OneTaq® Hot Start DNA Polymerase (New England Biolabs, cat. no. M0484), according to the manufacturer's guidelines. PCR products were separated by agarose gel electrophoresis, and the bands of interest excised and purified using the QIAquick® Gel Extraction Kit (QIAGEN, cat. no. 28704). Purified PCR products underwent Sanger sequencing (Genewiz) using the same primers as those employed in the PCR amplification. The primer sequences were as follows: CTGCCTTCTGAATGGA-CAGTT (forward; exon 3) and CCACAGAGATCCACACGGTTG (reverse; exon 8).

## FFPE cell pellets

Eight T175 flasks each of *PRLR* KO cells and parental WT HAP1 cells were used to prepare FFPE cell pellets. *PRLR* KO cells and parental WT HAP1 cells were dislodged from T175 flasks using 10 mM EDTA (Sigma, cat. no. E9884) and pelleted by centrifugation at 930 *g* for 5 minutes. Cell pellets were washed with phosphate-buffered saline (PBS) (Gibco, cat. no. 10010023) then fixed overnight using neutral buffered formalin (Sigma, cat. no. HT501128). Following fixation, the pellets were processed, trimmed, embedded, sectioned (4 µm thick), and mounted onto slides by the Translational Histopathology Laboratory, University of Oxford, using previously described methods[81]. Prepared FFPE T47D cell pellets were purchased from Amsbio, UK (cat. no. 3010-0620).

## Protein extractions and immunoblotting

The cell culture medium was removed from the dishes, and the cells washed twice with cold PBS. Next, 1X Radio-Immunoprecipitation Assay (RIPA) lysis and extraction buffer (Thermo Fisher Scientific, cat. no. 89900) containing 1X protease inhibitor (Roche, cat. no. 11873580001) and 1X phosphate inhibitor (Roche, cat. no. 4906837001) was added to the dishes. The cells were incubated on ice for 15 minutes. Whole cell lysates were scraped from the culture dishes and centrifuged at 20,000 *g* for 20 minutes at 4 °C. The supernatant was collected, and protein concentrations measured using the Pierce™ BCA Protein Assay Kit (Thermo Fisher Scientific, cat. no. 23227).

Laemmli sample buffer (4x) (Bio-Rad, cat. no. 1610747), including β-2-mercaptoethanol (Sigma cat. no. M6250), was added at a 1:3 ratio to cell lysates (40ug), full length 5-HT$_{2B}$ recombinant protein (20 ng) (Abnova, H00003357-P01), and PRLR overexpression human embryonic kidney (HEK293T) cell lysate (5ug) (Origene cat. no. LY424444). Samples were then boiled at 90 °C for 10 minutes then separated using 4-15% Mini-PROTEAN TGX precast gel cassettes (Bio-Rad, cat. no. 4568084, 10-well gels, and 4561086, 15-well gels). Proteins were transferred onto a Immobilon-FL polyvinylidene difluoride membrane (Merck Millipore, cat. no. IPFL00010). Membranes were blocked with 5% ECL Prime blocking agent (Cytiva, cat. no. RPN418) for 1 hour and then incubated with primary antibodies (Supp. Data S2) overnight at 4 °C. After washing with 10% tris-buffered saline (TBS) (Thermo Fischer Scientific, cat. no. J60764.K2) with 0.1% Tween 20 (T) (Bio-Rad, cat. no. 1610781), the membranes were incubated with secondary antibodies (Supp. Data S2) for 1 hour at room temperature. Following another wash with TBS-T, the membranes were dried using 100% methanol (Sigma, cat. no. 34860).

Dried membranes were imaged with the Licor Odyssey CLx Infrared Imaging System (LI-COR Biosciences), and visualised using Image Studio software (V.5.5.4; LI-COR Biosciences).

## Immunofluorescence labelling of FFPE tissue sections and FFPE embedded cell pellets

FFPE tissue sections (4 µm thick), cell pellets, and the human multiple cancer tissue microarray (Insight Biotechnology, cat. no. MC482) were dewaxed and rehydrated by incubating in Histo-Clear (National Diagnostics, cat. no. HS-200) for 5 minutes twice followed by washes in a graded ethanol series and ddH$_2$O (100%, 90%, 70% ethanol, and 100% methanol for 1 minute; final wash in ddH$_2$O for 5 minutes). Antigens were unmasked by heat-induced epitope retrieval (HIER). Slides were incubated in 10 mM citrate buffer (pH 6) (Sigma, cat. no. C9999) or tris-EDTA buffer (pH 9) (Abcam, cat. no. ab93684) and heated to 100 °C for 20 minutes using a microwave. Tissue sections and cell pellets were blocked with 5% goat serum (Sigma, cat. no. G6767) followed incubation with primary antibodies overnight at 4 °C in a humidity chamber. (Supp. Data S2). Tissue sections and cell pellets were incubated in secondary antibodies (Supp. Data S2) at room temperature for 30 minutes. For the anti-PRLR antibody the VectaFluor™ Excel Amplified Anti-Rabbit IgG, DyLight™ 488 Antibody Kit (Vector Laboratories, cat. no. DK-1488) was used, as per manufacturer instructions, in lieu of 5% goat serum and the secondary antibodies to amplify the PRLR signal. DAPI (1 µg/mL; Thermo Fischer Scientific, cat. no. 62248) was used to label nuclei. Slides were washed in 10% TBS-T after each antibody incubation step. The 5-HT$_{2B}$ receptor signal was amplified by tyramide signal amplification using the Tyramide SuperBoost™ Kit (Thermo Fisher Scientific, cat. no. B40941), as per the manufacturer's instructions. Following initial labelling of the PRLR, the 5-HT$_{2B}$ receptor, cathepsin Z (CTSZ), Ki-67, and GLP-1, respectively, FFPE tissue sections were subsequently sequentially labelled for glucagon and/or insulin. For glucagon and insulin, primary antibodies were incubated at room temperature for 2 hours each followed by incubation with secondary antibodies as described above (Supp. Data S2). Autofluorescence in FFPE tissue sections was quenched using the Vector® TrueVIEW® autofluorescence quenching kit (Vector Laboratories, cat. no. SP-8500-15). Fluorescence mounting media (Agilent, cat. no. S302380-2) was used to mount coverslips before imaging.

## Chromogenic immunolabelling of FFPE tissue sections and FFPE embedded cell pellets

FFPE tissue sections, cell pellets, and human multi-tissue microarray (Bio-Techne cat. no. NBP2-30232) were dewaxed and rehydrated as above, with 50% ethanol substituted for 100% methanol. Antigen retrieval was performed as previously described. The VECTASTAIN® Elite® ABC Universal PLUS Kit, Peroxidase (Horse Anti-Mouse/Rabbit IgG) (Vector Laboratories, cat. no. PK-8200) was used for chromogenic immunolabelling of tissue sections and cell pellets. Endogenous peroxidases were quenched using BLOXALL Blocking Solution for 10 minutes followed by blocking in 2.5% horse serum. Primary antibodies were incubated overnight at 4 °C followed by incubation in prediluted biotinylated horse anti-mouse/rabbit IgG secondary antibody for 30 minutes at room temperature. Tissue sections and cell pellets were incubated in VECTASTAIN Elite ABC Reagent for 30 minutes and signal developed using ImmPACT 3,3′-Diaminobenzidine (DAB) EqV solution. Slides were washed 3 times in PBS after incubation in primary and secondary antibodies. Nuclei were counterstained with haematoxylin (Vector Laboratories, cat. no. H3502) followed by dehydration in a graded ethanol series (50%, 70%, 90%, and 100% for a minute each), clearing in Histo-Clear for 10 minutes twice and mounted in DPX mounting media (Sigma, cat. no. 06522) before imaging.

## Image acquisition

IHC-IF imaging was carried out on an Olympus SpinSR SoRa spinning disc confocal microscope, utilising a 10 × 0.4 numeric aperture (NA) air objective and the 50 µm pinhole disc of a Yokogawa CSU-W1 spinning

disc unit. The system was equipped with a Hammamatsu ORCA Fusion BT camera and excitation lasers at 405 nm, 488 nm, 561 nm and 640 nm. Images following IHC-DAB labelling were acquired using the Olympus VS200 research slide scanner equipped with a 20 × 0.8 NA air objective and controlled by the VS2000 ASW software system.

## Image analysis pipeline for quantitative data analysis

A comprehensive analysis pipeline was developed and applied to all acquired pancreas whole tissue IHC-IF images. Images were processed and analysed using Arivis Vision4D (v4.1.2) software through a sequence of operations. Initially, a closing operation was performed on the glucagon and insulin channels to eliminate small gaps and link separate objects, followed by a mean filter to reduce noise and enhance image clarity. The 488 channels (anti-PRLR, anti-5-$HT_{2B}$ receptor, and anti-GLP-1) were also subjected to a mean filter to improve signal quality. Thresholding was then performed on the insulin and glucagon channels to identify the respective signals. These signals were combined to form a complete islet signal (whole islet). Size filtering of whole islets < 1000 $\mu m^2$, was employed to exclude non-specific staining, ensuring that the analysis focused on genuine islet structures. A closing operation was subsequently applied to refine islet shapes, followed by intersections with glucagon and insulin channels to isolate regions expressing both glucagon and insulin signal. The islets were then compartmentalised to differentiate whole islets, α cells (glucagon positive cells only), and β cells (insulin positive cells only) enabling detailed morphological and quantitative analysis of each compartment.

Object math was used to assess exocrine tissue and islet compartments to identify and quantify areas. Islet diameter was measured across the longest axis of the islet. To identify nuclei within islets, a watershed function was applied to the nuclei (DAPI) channel, followed by a size filter to exclude objects larger than 10 μm, ensuring artefacts or incorrectly detected objects were removed. This filtering process did not affect the identification or quantification of α cells or β cells. Nuclei counts and diameters within α cells, β cells, and whole islets were then quantified. Data for all images were exported to Microsoft Excel for processing.

To account for variations in total tissue area between different sections, multiple methods of normalising whole islet, α cell, β cell, bihormonal cell, and GLP-1 positive areas were applied before comparisons. To adjust for the variation in tissue size between sections the area of whole islets, α cells, β cells, bihormonal cells, and GLP-1 positive area was expressed as a ratio to the total exocrine area per section, termed fractional area. The mean area of islet, α-cell, β-cell, bihormonal cell, and GLP-1 positive area was calculated by dividing the measured area by the number of islets per tissue section. Additionally, the proportions of α cells, β cells, bihormonal cells, and GLP-1 positive regions relative to the total islet area were determined, as well as the proportion of GLP-1 positive area relative to the total α-cell area. Finally, the number of islets normalised to the tissue section area was represented as islet density.

To estimate the cell size of α and β cells, their total measured areas were divided by the number of nuclei per tissue section within the α and β cell labelled regions, respectively, and the nuclei counts within α and β cells were expressed as a percentage of the total islet nuclei count per tissue section. Data were plotted, and statistical comparisons between groups were performed in GraphPad Prism (v10.4.2, build 534).

## Colocalisation analysis

Colocalisation analysis of the 5-$HT_{2B}$ receptor and insulin channels for whole pancreatic tissue sections was performed using the in-built colocalisation tool of Arivis Vision4D image analysis software. In-built software determined auto thresholds for the 5-$HT_{2B}$ receptor and insulin channels were applied. Scatter plots of the 5-$HT_{2B}$ receptor channel versus the insulin channel were generated, and Manders' coefficients calculated. Specifically, Manders' coefficient M1 (representing the proportion of the insulin signal overlapping with the 5-$HT_{2B}$ receptor signal) and Manders' coefficient M2 (representing the proportion of the 5-$HT_{2B}$ receptor signal overlapping with the insulin signal) were determined.

## Statistics

Statistical analysis for the LC-MS/MS was performed as described above. For the image analysis data, normally distributed data were analysed using the unpaired Student's t-test, while non-parametric data were assessed with the Mann-Whitney test. Statistical significance was set at $p < 0.05$.

## Reporting summary

Further information on research design is available in the Nature Portfolio Reporting Summary linked to this article.

## Data availability

The LC-MS/MS proteomics data generated in this study have been deposited in the ProteomeXchange Consortium via the PRIDE[76] partner repository with the dataset identifier PXD055852, https://proteomecentral.proteomexchange.org/cgi/GetDataset?ID=PXD055852. The LC-MS/MS proteomics data, representing protein identifications and label-free quantification intensities prior to statistical analysis in this study are provided in the supplementary materials. The full microscopy datasets supporting the findings of this study are not publicly available due to image file sizes exceeding repository upload limits. Representative images are provided in the manuscript and supplementary materials, furthermore, source data are provided with this paper. All data supporting the findings described in this manuscript are also available from the corresponding author upon request. Source data are provided with this paper.

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

## Acknowledgements

We a grateful to the Network for Pancreatic Organ donors with Diabetes (nPOD), University of Florida, USA, a collaborative type 1 diabetes research project supported by Breakthrough T1D and The Leona M. & Harry B. Helmsley Charitable Trust (Grant#3-SRA-2023-1417-S-B) for the preparing, processing and providing tissues used in this study. The content and views expressed are the responsibility of the authors and do not necessarily reflect the official view of nPOD. Organ Procurement Organisations (OPO) partnering with nPOD to provide research resources are listed at (https://npod.org/for-partners/npod-partners/). We are also grateful to the Translational Histopathology Laboratory, University of Oxford for processing and mounting FFPE HAP1 cell pellets onto slides. This research was supported by the Wellcome Trust (grants 091157/Z/10/Z and 107212/Z/15/Z), the Juvenile Diabetes Research Foundation (JDRF; grants 9-2011-253, 5-SRA-2015-130-A-N, and 4-SRA-2017-473-A-N), and the 2021 EASD–Novo Nordisk Foundation Diabetes Prize for Excellence, awarded to the Diabetes and Inflammation Laboratory, University of Oxford.

## Author contributions

F.S. and M.I.S. conceptualised the study, designed experiments, and directed the research. F.S. performed the major experiments and draughted the initial manuscript. KH assisted with experiments. J.B. and E.D. assisted with image acquisition and image analysis. S.D. and R.F. performed the LCM, LC-MS/MS experiment, and assisted with LC-MS/MS data analysis. J.T., N.K. and M.V. contributed intellectually to overall study design and manuscript editing. All authors read and reviewed the final manuscript.

## Competing interests

Authors declare that they have no competing interests.
