## [Transparent Peer Review file · Nature Communications]

A new paradigm of islet adaptations in human pregnancy: insights from immunohistochemistry and proteomics.

Corresponding Author: Dr Faheem Seedat

Version 0:

Reviewer comments:

Reviewer #1

(Remarks to the Author)

This manuscript reports about the proteomic profile of human pancreas exocrine and endocrine tissue in pregnant women compared to non-pregnant women. These analyses are complemented by image analyses pertaining islet size distribution, alpha, beta and bi-hormonal cell composition as well as the expression of PRLR and 5-HT2A receptors, which have been implicated in the signalling cascade promoting proliferation of beta cells in pregnant mice.

The large proteomic data sets are novel and intriguing, being from a rare, unique cohort of pregnant women and the most extensive collected so far from human islets in situ. The lack of pronounced proteomic changes between islets of pregnant women compared to non-pregnant women is especially remarkable. The observation that pregnancy is associated with a greater expansion of the alpha cell compartment relative to the one of beta cells is also a valuable contribution.

Despite these merits, the study has significant conceptual and methodological shortcomings, with some apparent internal inconsistencies.

Specific comments

- The comparison between the proteomic profiles islets of islets from pregnant and non-pregnant women is limited by the heterogeneity of the donors. Among the 7 women in the third semester of pregnancy, 4 were not known to suffer from gestational diabetes, although in one case (6310) HbA1c levels were unknown, 2 had gestational diabetes but were negative for islet autoantibodies, while one (6405) had type 1 diabetes, being positive for >2 anti-islet autoantibodies (GADA, IA2A, ZnT8A). The occurrence of beta cell autoimmunity is likely to have an impact on the outcome of the analysis. Moreover, one of the non-pregnant donors (6401) had an HbA1c of 5.8, meaning she had impaired glucose tolerance, while in another case (6253), who had the highest BMI, HbA1c levels were unknown, i.e., the occurrence of glucose intolerance cannot be ruled out. Further differences in ethnicity and especially in the BMI among pregnant and not pregnant women are additional confounders which likely account for the meagre number of islet proteins differentially expressed in pregnant versus non-pregnant women. These considerations, together the very limited power of the analysis between normal pregnancy and GDM donors, greatly temper the conclusions which can be drawn from the data (e.g., lines 589-590; 649-650; 703-704 and so on).

- To assess the representative value of the analysis, it would be important to clarify whether the LCM area of 235,000 μm^2 tissue/donor was the collective surface of both endocrine and exocrine tissue, and in this case how much was the islet fraction, or of each part of the pancreas. Moreover, it is not clear, which part of the exocrine tissue was collected (acinar cells, ductal cells, both?). Along the same line, it would be important to specify the average number of islets analysed for each donor and whether these were collected from serial sections or from sections distanced >200 μm from each other, i.e. most likely representing distinct islets.

- Knowledge about the number of individual islets analyzed for each donor is also important to assess how strong the evidence for the lack of Ki67 detection in the islets (Fig. 5) is, as well as for the other markers (PRPL, Fig. 6; 5-HT2B, Fig. 7 and related figures). Moreover, islet diameter is not a good approach to estimate the islet size, since islets are irregular in shape. The measurement of the whole islet area is a better way to estimate changes in islet size.

- Supplement Fig. 4k shows that the fraction of beta cells is not changed, and it accounts approx. 45% of the total islet cell mass. What is the remaining 55%?

- The methods section does not include information on T47D cells. Also, it is unclear whether human EndoC-bH3 cells were treated with tamoxifen to remove the cassette with the SV40 large T-antigen. If not, such cells keep proliferating and thus their comparison to adult human beta cells is less stringent.

- As only four islet proteins were found to change in their expression the authors conclude that human islets, unlike mouse islets, undergo few proteomic changes during pregnancy. However, the interpretation of the proteomic data is hindered by the provision of the raw data only without normalization. Moreover, inter-individual differences are large, conceivably due to the limited size of the cohort, and likely accounting for the meagre number of islet proteins found to be differentially expressed in pregnancy. The conclusion regarding the lack of islet proteomic changes during pregnancy should be further tempered, since for obvious reasons protein flux could not be quantified. For instance, in pregnancy the islet levels of insulin were unchanged but given its 2-fold increased secretion, its translational rate is likely to be incremented. Such an adaptive mechanism, which is not considered by the authors, seems more plausible than the neogenesis of alpha cells, which is not adequately documented (see comments below), and their transdifferentiation into beta cells, which remains speculative.

- Caution should be applied in the interpretation of the image quantification of bihormonal cells. The possibility that the signals of glucagon and insulin overlap accidentally due to superimposed alpha and beta cells is not negligible, even in 4 μ m thick sections due to interdigitations of their profiles. The additional labelling of cells for membrane markers and the employment of superresolution microscopy techniques, such as structured illumination microscopy rather than spinning disk confocal microscopy, which relative to conventional confocal microscopy reduces the bleaching of the fluorescence overtime, but does not improve resolution, would have been beneficial.

- To maximize sensitivity and detect low-abundance targets such as the 5-HT_{2B} receptor the choice of Tyramide Signal Amplification (TSA) would have been a more powerful option than the used Vecta Fluor Excel Amplified IgG.

- Again regarding the bihormonal cells, were they included in the plots for alpha and beta cells? Have they been considered when investigating the expression of PRLR and 5-HT_{2B} in islets?

- Placental lactogen, which is produced starting from 24 weeks of gestation, can also bind to the growth hormone receptor (GRH), although with lower affinity than to PRLR. Therefore it would have been appropriate to analyse its expression by immunomicroscopy,

- Supplementary Fig. 6. In the western blots shown in panel A, the size of the molecular weight markers is missing. In the case of the PRLR HEK293T cells, could the overexpressed receptor be detected also with an antibody directed against the DDK tag? In panel c, the difference in the PRLP signal between WT and PRLP KO cells is not readily visible.

- The absence of 5-HT_{2B} in pancreatic islet beta cells in situ could have been corroborated by single molecule RNA FISH, which benefits from greater potentials for signal amplification.

- Serotonin can bind to multiple receptors, including 5-HT₃, which was shown to affect insulin secretion of mouse beta cells (Kim et al, 2015) and be upregulated in pregnant mice (Ohara-Imaizumi et al, 2013). Was 5-HT₃ detected in the proteomic analysis? Along this line, the expression of other serotonin receptors, and especially 5-HT₃ would have been appropriate.

- The discussion cites evidence that ~80% of human beta cells are in contact with alpha cells (lines 832-833). In view of these data, the suggestion that the greater alpha cell population size in pregnancy is required to better influence beta cells is rather puzzling/naïve (lines 907-909). This notion, on the other hand, is challenged by recent evidence that ~50% of human insulin-expressing islets are virtually devoid of glucagon-producing α -cells (Lehrstrand et al, 2024).

- As mentioned above, the suggestion that alpha cells transdifferentiate into beta cells during pregnancy is merely speculative. In such a scenario, shouldn't the number of beta cells increase more and not less than alpha cells, especially in view of the limited evidence of islet cell proliferation by Ki67+ labelling? In this context, the increase of alpha cells in the periductal region of pregnant donors is mentioned (lines 812-814), also in relationship with the number of beta cells in the same region, but no evidence for this is provided (and no duct cell marker is shown). And if so, shouldn't the islet hormone positive clusters, i.e. the islet density being increased rather than remain equivalent in pregnant compared to non-pregnant donors (Suppl. Fig. S1a)?

Reviewer #2

(Remarks to the Author)

In this manuscript Seedat F et al exploit unique FFPE pancreatic tissue samples from pregnant human donors and matched non-pregnant controls to characterize the adaptive changes occurring in islets during pregnancy, through both proteomics (LC-MS/MS) of laser-captured micro-dissected exocrine and islet tissue samples and morphometric analysis (by immunofluorescence staining and high-resolution imaging) of the whole tissue sections. Proteomic profiling shows a few changes between pregnant donors and matched controls and no significant differences between pregnant donors affected by gestational diabetes mellitus (GDM) and donors with a normal pregnancy. Morphometric analysis of tissue sections identifies a 1.9-fold increase of islet area and a 4.3-, 1.9- and 3.1-fold increase of alpha, beta and bi-hormonal cell area, respectively, in pregnant donors compared to non-pregnant controls, possibly driven by an increase in islet size and cell number. The evaluation of prolactin receptor (PRLR) and serotonin 2B (5-HT_{2B}) receptor expression by immunofluorescence and high-resolution imaging, as possible mediators of the observed islet changes, results in an

increased expression of both PRLR and 5-HT2B receptor in alpha cells only. Finally, a 2.9-fold increase of glucagon-like peptide-1 (GLP-1) positive alpha cell area is also reported. Again, no significant differences are observed between GDM donors and donors with a normal pregnancy.

The work is of potential interest to the diabetes field. The main strengths are the rarity of pancreatic samples from pregnant human donors and the accurate experimental design and analysis. The main limits are the limited number of samples and the overall lack of biological validation. Thus, the manuscript could be implemented via additional context and additional in vitro experiments.

MAJOR

1. It is not clear whether the donors, classified as affected by GDM, have been diagnosed with GDM or if this status has been deduced from clinical details (lines 176-178). The authors should clarify, and eventually specify, which are the clinical details used for this classification and, if possible, the therapy followed by the donors. It is very unlikely that the comparison of 5 normal pregnant and 3 GDM pregnant donors only would give enough statistical power to draw a proper conclusion about the effect (or lack of it) of GDM in the performed evaluation but could be of interest anyway if supported by a proper donor characterization. The authors should add this limit (limited number of donors) also in the relative discussion section (line 886-899).

2. Proteomic analysis identifies four differentially expressed proteins (DEPs) when pregnant and non-pregnant donors are compared. This, of course, could be due to different reasons since human islets are heterogenic and pregnancy adds additional confounding factors with the variation of beta and alpha cell ratio (as suggested by the morphometric analysis). The additional use of a lower (but still acceptable) statistical threshold, at least for this comparison, might give some additional pieces of information about the specific proteomic changes of human islets in pregnancy, and better clarify the differences with what seen in mouse islets (as in ref 6, mentioned by the authors in the discussion section).

3. At least one of the identified DEPs should be validated by immunofluorescence on the pancreatic tissue sections and its role in alpha and/or beta cell physiology in pregnancy investigated. Is this protein expressed by beta cells, alpha cells or both? Does its expression change in pregnant donors and in which cell type? Are there any DEPs that could explain the decisive and unexpected increase of alpha cell area, especially of that positive for GLP-1? Some additional mechanistic studies are needed.

4. Interestingly, the authors find a greater increase of alpha cell area than beta cell area in pregnant donors compared to control. They suggest that this might sustain the increase of insulin secretion seen during pregnancy, possibly through GLP-1 secretion and/or alpha to beta cell transdifferentiation. It is, however, not clear the role of the observed increased expression of PRLR and 5-HT2B in alpha cells in the above-mentioned changes. Some additional mechanistic studies in a mice alpha cell line (such as aTC1.9) or primary human alpha cells would give some hints about the topic. In addition, previous works on mice models should be better discussed and compared to the present results, such as those reported in ref 43 and 56.

MINOR

1. Line 145: This might be better defined as "ex vivo", rather than "in vivo". Please correct.

2. Line 475-477: According to this phrase it seems that objects smaller than 10 μm have been excluded during nuclei identification and count. Since the beta cell has an average diameter of 10 μm (doi.org/10.1016/j.semcd.2020.04.005), It is not clear what the authors intended.

3. The number of islets isolated and subjected to proteomic analysis for each donor should be indicated in the methods, as well as the number of islets analysed per tissue section in the morphometric part.

4. Fig. 2b: It looks like an Overrepresentation Analysis rather than a Gene Set Enrichment Analysis as reported in figure caption and in the methods section. Please clarify.

5. Figure S1 is of interest for the reader and might be added to the main figure panels (maybe Fig 4?).

6. Antibody validation (paragraph 3.6) is technically important, but it might better fit the methodology section instead of the results.

7. Paragraphs 3.7 and 3.8 might be better presented as one paragraph.

8. Figures 7.a and 8.a should also show the staining of at least one non-pregnant donor sample. In addition, the ref to Fig 8.c should be added at line 722.

9. Line 890-891: The phrase is not clear; may the authors clarify?

10. Please be aware that Kolic J et al ([doi:10.1016/j.cmet.2024.06.001](https://doi.org/10.1016/j.cmet.2024.06.001)) has recently published a manuscript in which almost 8000 islet proteins are identified. The authors may want to correct sentences such as those at line 777-778 and 932-933, accordingly.

Reviewer #3

(Remarks to the Author)

This study utilises precious human donor pancreas formalin-fixed paraffin embedded (FFPE) tissue blocks from nPOD from pregnant (n=7) and moderately well matched non-pregnant (n=7) women with the aim of determining the mechanisms of human islet adaptation to pregnancy. From laser captured exocrine and islet tissue, proteomics was performed which was analysed using various bioinformatics processes. Pancreas sections were also immunostained for insulin, glucagon, prolactin receptor (PRLR), serotonin receptor 2B (HTR2B) and GLP-1, and images were systematically analysed using Arivis Vision4D software.

The strengths of the work are the high quality nPOP pancreas FFPE blocks, high quality immunohistochemistry analyses of pancreas with extensive validation of the quality of the antibodies used and the methodology used for analysis using the Arivis software.

The data from the IHC studies showing increased alpha-cell area and increased expression of GLP-1 in alpha cells in pancreases of pregnant women is of particular interest.

A major achievement was successful laser microcapture of endocrine and exocrine tissue and proteomic analyses of the

tissue obtained in which endocrine tissue was easily distinguished from exocrine tissue. This methodology, however, showed minimal differences between the pregnant and non-pregnant islet proteomes. This seems surprising considering the marked increases in both alpha and beta cells within the islets, suggesting that the methodology using FFPE tissue material may be limiting (easy to differentiate endocrine from exocrine tissue, but challenging to detect differences within endocrine tissue).

Weaknesses include exclusion of small islets from the analyses, a failure of the proteomics to detect the PRLR and HTR2B despite it being detected in exocrine and endocrine IHC analyses. Further, only one protein within serotonin signalling has been focussed on. Another limitation that is unavoidable with the availability of samples is that there was only one first trimester pancreas and no second trimester pancreases available for analysis (islet adaptations which occur earlier in pregnancy may have been missed).

Major comments:

- 1) The matching of the subjects showed BMI to be higher in the pregnant women (almost significant despite low sample size). Was this a pre-pregnancy or pregnancy BMI?
- 2) Table 1 suggests that 2 of the non-pregnant subjects were GAD Ab positive, whereas Suppl Table 1 suggests the GAD Ab subjects were in the pregnant group. Please check the data and correct.
- 3) Did the GAD Ab positive pancreases show any evidence of insulinitis, and could this have affected the results? This should be reported in the manuscript.
- 4) I have concerns about the proteomics findings of pregnant vs non-pregnant islets. Validation of the proteomics of human islets obtained from formalin fixed paraffin embedded sections against fresh human islets (do not need to be pregnant subjects) would be helpful. Are some of the proteins of interest (e.g. PRL receptor, serotonin receptors, tryptophan hydroxylases) better detected in fresh islets, as opposed to FFPE islet tissue?
- 5) Why has so much focus been placed on HTR2B? It was the tryptophan hydroxylases that had much increased expression in pregnant compared to non-pregnant islets in mice. Also could other serotonin receptor isoforms be more important for pregnancy adaptation in human islets?
- 6) HTR2B was shown to be present in human islet beta-cells by IHC previously (*Diabetologia* 2016 59:744–754). Why the discrepancy between this study and that previous study? Please discuss.
- 7) Butler et al (*Diabetologia* 53, 2167-2176 (2010)), showed that there was a marked increase in small islets (including islet cell clusters) in human pregnancy, and that this could be from neogenesis from pancreatic ductal tissue. Why were small islets excluded from this study, when previously it was considered the small islets were important in islet adaptation to human pregnancy?
- 8) The methodology section is very detailed which is helpful, but some of the detail could be moved to the supplemental data section.
- 9) The discussion needs to cover off on the limitations of the study better.

Overall, it is worthwhile making the point of the importance not to extrapolate too much from small animal studies to humans in determining the mechanisms of islet adaptation to pregnancy. However, the evidence from this study, as it stands, is not strong enough to exclude important roles for lactogenic hormones and serotonin signalling in the adaptation of human islets to pregnancy.

Version 1:

Reviewer comments:

Reviewer #1

(Remarks to the Author)

To the Editor and the Authors

The authors have largely addressed my concerns by acknowledging in the text the limitations related to the number of available samples, reagents, and methodological approaches, which ultimately temper the conclusions that can be drawn from this study. The result is a substantially improved article.

However, regarding my comment about placental lactogen and the expression of GHR, it would be appropriate for the authors to explicitly state among the study's limitations that PRL may also bind to this receptor, but that its expression could not be investigated due to the restricted availability of tissue samples.

That said, I still maintain that the rarity of samples from pregnant women—contrary to the authors' implication in their introductory remarks—does not underscore their scientific value, but rather highlights the complexity of drawing sound conclusions about the scientific questions raised. If the limited number and/or significant heterogeneity of samples hinder such conclusions, it would be preferable to refrain from making them, as they risk being merely speculative.

Evidence of similar samples being included in prior studies published in high-profile journals does not justify lowering the threshold for rigorous analysis; rather, it raises important questions about our standards for acceptable scientific practice. Along the same lines, being the largest study on islets from pregnant women does not automatically imply that the cohort is adequately powered to yield robust insights. As the authors themselves acknowledge in response to my initial comments, the results remain preliminary. Therefore, in my view, their readiness for publication is still uncertain.

Reviewer #2

(Remarks to the Author)

I thank the authors for their detailed responses and revision work. Overall, the manuscript has been improved through a more in-depth discussion of the results in comparison with the literature, as well as additional analyses that provide a more comprehensive perspective.

Below, I provide my feedback on the proposed modifications.

MAJOR COMMENTS

1. Regarding the first point, the authors have adequately addressed my concern.
2. Regarding the second point, the authors have provided additional context for the proteomic analysis, which is now properly discussed in the Discussion section.
3. Regarding the third point, the authors have correctly performed a technical validation of one of the differentially expressed proteins using immunofluorescence and have also provided an appropriate discussion of the results.
4. The authors have correctly added a discussion of previous studies using mouse models, which highlight a similar increase in alpha cell mass.

Points 3-4: However, it is worth noting that the authors did not perform biological validation or mechanistic studies, stating that these aspects are beyond the scope of the present study. While this is understandable, studies published in prestigious journals, such as Nature Communications, are generally expected to include some form of biological validation to support their findings and speculations.

Additionally, I do not agree with the statement that 'the rarity of these samples underscores their scientific value,' especially since it is followed by references to other journals, which does not seem entirely appropriate. The value and rarity of the samples—while certainly recognized and acknowledged throughout the review process—should not be the primary or sole merit of a study. That said, I understand that proper validation may take time and could, indeed, be the focus of a future study. Furthermore, the methodological and technical accuracy demonstrated throughout this work somewhat compensates for the lack of biological validation and ensures the study's overall value.

MINOR COMMENTS

1-10. Almost all the minor observations have been addressed, and the Results section is now more accessible to readers.

Final Recommendation

In conclusion, the manuscript has been improved, and most of my concerns have been addressed. I only suggest that the authors correct a typo at line 938 (where there is an extra square bracket). After this minor correction, in my view, the manuscript is ready for publication.

Reviewer #3

(Remarks to the Author)

My comments have been adequately addressed. The limitations of the small sample size and heterogeneity of the pregnancy group, as well as the methodology, is now much better outlined in the manuscript.

Reviewer #1 (Remarks to the Author):

Dear Reviewers,

Thank you for reviewing the manuscript and for your insightful comments.

Please find our responses to your feedback below. The original comments are displayed in bold and responses in normal text. Changes made to the manuscript are in italics and underlined. The relevant line numbers are also shown.

In addition to the untracked manuscript, a manuscript showing the tracked changes and line numbers is also provided.

This study was aimed to interrogate human pancreatic tissue and in doing so we sought the largest biorepository of globally available of pancreatic samples from pregnant women, the Network for Pancreatic Organ Donors with Diabetes (nPOD). This repository is unique in that it takes samples from patients perimortem thus it is important to stipulate that the patients analysed were not selected but instead all the available samples from pregnant women were used for the study. Entry into the biobank is not based on prespecified criteria. Entry into the biobank is based on consent and being close to death. The rarity of these samples underscores their scientific value, and they have been utilised in high-impact publications, including *Cell Metabolism*, *Nature Cell Biology*, *Diabetes*, and *Diabetologia*.

While we acknowledge that a more stringent selection of patient samples would have been ideal, the eight pancreata from pregnant women included in our study constitute the largest study of islets from pregnant women to date. Given how uncommon such specimens are, our study makes full use of all available samples, providing a unique and valuable opportunity to investigate human pancreatic adaptations in pregnancy.

This manuscript reports about the proteomic profile of human pancreas exocrine and endocrine tissue in pregnant women compared to non-pregnant women. These analyses are complemented by image analyses pertaining islet size distribution, alpha, beta and bi-hormonal cell composition as well as the expression of PRLR and 5-HT2A receptors, which have been implicated in the signalling cascade promoting proliferation of beta cells in pregnant mice.

The large proteomic data sets are novel and intriguing, being from a rare, unique cohort of pregnant women and the most extensive collected so far from human islets in situ. The lack of pronounced proteomic changes between islets of

pregnant women compared to non-pregnant women is especially remarkable. The observation that pregnancy is associated with a greater expansion of the alpha cell compartment relative to the one of beta cells is also a valuable contribution.

Despite these merits, the study has significant conceptual and methodological shortcomings, with some apparent internal inconsistencies.

Specific comments

- The comparison between the proteomic profiles islets of islets from pregnant and non-pregnant women is limited by the heterogeneity of the donors. Among the 7 women in the third semester of pregnancy, 4 were not known to suffer from gestational diabetes, although in one case (6310) HbA1c levels were unknown, 2 had gestational diabetes but were negative for islet autoantibodies, while one (6405) had type 1 diabetes, being positive for >2 anti-islet autoantibodies (GADA, IA2A, ZnT8A). The occurrence of beta cell autoimmunity is likely to have an impact on the outcome of the analysis. Moreover, one of the non-pregnant donors (6401) had an HbA1c of 5.8, meaning she had impaired glucose tolerance, while in another case (6253), who had the highest BMI, HbA1c levels were unknown, i.e., the occurrence of glucose intolerance cannot be ruled out. Further differences in ethnicity and especially in the BMI among pregnant and not pregnant women are additional confounders which likely account for the meagre number of islet proteins differentially expressed in pregnant versus non-pregnant women. These considerations, together the very limited power of the analysis between normal pregnancy and GDM donors, greatly temper the conclusions which can be drawn from the data (e.g., lines 589-590; 649-650; 703-704 and so on).

Thank you for this insightful comment from the reviewer. We agree with the reviewer and acknowledge that the heterogeneity among donors, including differences in GDM status, β -cell autoimmunity, BMI, and HbA1c levels, is a limitation of this study. These factors may have contributed to variability in the proteomic profiles and limited the ability to identify significant differences between pregnant and non-pregnant donors.

Due to the rarity of obtaining such unique human islet samples, the sample size is inherently limited, which we have now explicitly acknowledged in the discussion. Based on the reviewers comments we have tempered the conclusions accordingly to reflect the constraints of the study and highlighted the need for larger, more homogeneous cohorts in future investigations. Despite these limitations, the study provides valuable preliminary insights into islet adaptations during pregnancy.

Line 1016 – 1033: “The limited availability of high-quality pancreatic tissue from pregnant women significantly constrained the sample size in this study and introduced heterogeneity within the cohort. Variability in ethnicity, GDM status, anti-GAD antibody status, and insulinitis among donors may have contributed to the few observed differences in proteomic profiles and may have masked more subtle biological patterns. For pregnant donors, only the pregnancy BMI was available. Due to the circumstances under which donor tissues are obtained, it is not possible to acquire pre-pregnancy BMI measurements, and this limitation complicates the interpretation of BMI-related effects on islet adaptations during pregnancy. The heterogeneity of the cohort highlights a broader challenge in the field of pregnancy-related islet adaptations. Greater collaboration between researchers to increase the availability of pancreatic tissue from pregnant women to study more homogenous cohorts is critical to advancing our understanding of islet adaptations during pregnancy and improving outcomes in GDM research.”

- To assess the representative value of the analysis, it would be important to clarify whether the LCM area of 235,000 μm^2 tissue/donor was the collective surface of both endocrine and exocrine tissue, and in this case how much was the islet fraction, or of each part of the pancreas. Moreover, it is not clear, which part of the exocrine tissue was collected (acinar cells, ductal cells, both?). Along the same line, it would be important to specify the average number of islets analysed for each donor and whether these were collected from serial sections or from sections distanced $>200 \mu\text{m}$ from each other, i.e. most likely representing distinct islets.

Thank you for this important point. This is now clarified in the methods.

Line 191 – 197: “For each donor, 235,000 μm^2 of islet tissue and 235,000 of μm^2 exocrine tissue, consisting primarily of acinar cells, were microdissected. The exocrine tissue served as an internal control to verify the purity of islet tissue isolation. A single tissue section was utilised for each donor, islets were selected at random, and care was taken to ensure that the dissected islets were separated by a minimum distance of at least $> 200 \mu\text{m}$, to ensure that each dissected region represented a distinct islet.”

Line 489 – 490: “On average 10 islets per donor were analysed (Supp. Table S1).”

Supplementary Table S1 has been updated to include the number of islets analysed per donor.

- Knowledge about the number of individual islets analyzed for each donor is also important to assess how strong the evidence for the lack of Ki67 detection in the islets (Fig. 5) is, as well as for the other markers (PRPL, Fig. 6; 5-HT2B, Fig. 7 and related figures). Moreover, islet diameter is not a good approach to estimate the islet size, since islets are irregular in shape. The measurement of the whole islet area is a better way to estimate changes in islet size.

Thank you for this valuable comment. The number of islets isolated and subjected to proteomic analysis for each donor, as well as the number of islets analysed per tissue section in the morphometric analysis and expression of Ki67 and PRLR, are now included in Supplementary Table S1.

Furthermore, the reviewer raises a valuable point. To address this, we have used whole islet area rather than islet diameter to measure islet size in tissue sections. While we have included data describing the longest axis of islet diameter, we have also noted that this metric is limited due to the irregular shapes of islets.

Line 1035 - 1036: "Measures of islet diameter are limited by the irregular shapes of islets, making islet area is a more reliable metric for measurement."

- Supplement Fig. 4k shows that the fraction of beta cells is not changed, and it accounts approx. 45% of the total islet cell mass. What is the remaining 55%?

We thank the reviewer for their thoughtful comment. Our analysis of islet cell composition was conducted using IHC-IF and an automated image analysis pipeline, with normalisation to whole islet area to ensure a systematic and unbiased quantification approach. In addition to the contribution from β cells to islet area, we observed an α -cell fraction of 5-12.5% and identified 2.5-5% bihormonal (insulin+ glucagon+) cells. The remaining 55% comprises constituents that were not explicitly examined, as they were beyond the focus of this study. These likely include contributions from other endocrine cell types, connective tissue, vasculature, and stromal components, which were not specifically measured. Several factors may contribute to the observed proportions. First, our normalisation to whole islet area may result in proportions differing from those based on absolute cell counts. Second, our automated image analysis pipeline, while rigorous, is inherently influenced by segmentation and classification parameters. Importantly, our approach provides a reproducible and quantitative assessment of islet composition, and we are confident in the robustness of our findings. Future work incorporating additional markers for δ -cells and refining classification parameters may help further dissect the contributions of different endocrine populations. We appreciate the reviewer's insight and believe our

findings add to the understanding of islet adaptations during pregnancy.

- The methods section does not include information on T47D cells. Also, it is unclear whether human EndoC-bH3 cells were treated with tamoxifen to remove the cassette with the SV40 large T-antigen. If not, such cells keep proliferating and thus their comparison to adult human beta cells is less stringent.

Thank you for highlighting this. These points have now been clarified in the Methods and Supplementary Methods section.

Line 331 - 332: "Prepared FFPE T47D cell pellets were purchased from Amsbio, UK (cat. no. 3010-0620)."

Supplementary Methods: "Cells were differentiated by adding 4-hydroxitamoxifen (1 μ L/10 mL) to culture media for 21 days."

- As only four islet proteins were found to change in their expression the authors conclude that human islets, unlike mouse islets, undergo few proteomic changes during pregnancy. However, the interpretation of the proteomic data is hindered by the provision of the raw data only without normalization. Moreover, inter-individual differences are large, conceivably due to the limited size of the cohort, and likely accounting for the meagre number of islet proteins found to be differentially expressed in pregnancy. The conclusion regarding the lack of islet proteomic changes during pregnancy should be further tempered, since for obvious reasons protein flux could not be quantified. For instance, in pregnancy the islet levels of insulin were unchanged but given its 2-fold increased secretion, its translational rate is likely to be incremented. Such an adaptive mechanism, which is not considered by the authors, seems more plausible than the neogenesis of alpha cells, which is not adequately documented (see comments below), and their transdifferentiation into beta cells, which remains speculative.

We thank the reviewer for this important comment.

As part of the default parameters, DIA-NN does apply a retention time-dependent normalisation alongside the commonly used MaxLFQ quantitation algorithm, which also includes a data normalisation step. We recognise that this may not have been clear in the original text. To address this, we have revised the text (Methods) to explicitly clarify that these steps are part of the default parameter settings and cited the following paper

by Cox *et al*: “Accurate Proteome-wide Label-free Quantification by Delayed Normalization and Maximal Peptide Ratio Extraction, Termed MaxLFQ”.

Line 224 – 228: “DIA-NN incorporates retention time-dependent normalisation as part of its default parameters, along with the commonly used MaxLFQ quantitation algorithm, which also includes a data normalisation step. These normalisation processes are integral to the default settings of the software and ensure consistency in the analysis.”

As outlined in our response to the reviewer’s comment above, we now clearly acknowledge the sample size and interindividual variability of the cohort as a limitation.

Line 1016 – 1020: “The limited availability of high-quality pancreatic tissue from pregnant women significantly constrained the sample size in this study and introduced heterogeneity within the cohort. Variability in ethnicity, GDM status, anti-GAD antibody status, and insulinitis among donors may have contributed to the few observed differences in proteomic profiles and may have masked more subtle biological patterns.”

In response, we have revised the Discussion.

We have tempered our conclusion regarding the limited number of differentially expressed proteins detected and acknowledge that we have not presented data on dynamic protein changes, which could provide further insights into islet adaptation during pregnancy.

Line 800 – 805: “While human islets exhibit significant functional adaptation, as demonstrated by their ability to double insulin secretion by late gestation, our findings demonstrate that human islets undergo few proteomic changes during pregnancy. However, we acknowledge that dynamic proteomic changes, which may play a role in islet adaptations during pregnancy, may be masked as we do not present data on protein flux.”

We thank the reviewer for the thoughtful comment regarding the adaptive mechanisms of β cells during pregnancy. We agree that the increased translational activity of β cells, rather than neogenesis of α cells, is a plausible explanation for meeting the heightened insulin demands of pregnancy.

As outlined in our revised text below, we discuss that as whole islet insulin levels were changed during pregnancy, the high insulin demands of pregnancy are met through alternate adaptive mechanisms. We propose that GLP-1 paracrine signaling may play a

key role in optimising the utilisation of stored insulin and stimulating the translational machinery necessary to support these increased requirements.

Line 876 – 906: “During pregnancy, the increased demand for insulin to maintain glucose homeostasis presents a significant metabolic challenge. Despite this, total islet insulin levels remain unchanged in proteomic analysis, suggesting that β cells adapt by enhancing their production and secretion rates without accumulating excess insulin stores. Such an enhanced functional output likely relies on regulatory signals. Like Qiao et al., we too observe an increase in GLP-1 abundance in α cells during human pregnancy and propose that this increase in GLP-1 may play a role in supporting β -cell function.

Human islets are anatomically uniquely designed to facilitate paracrine communication, as the different hormone-secreting cells within islets are more intermixed than in mice 20, 21, 22, 23. For example, unlike the organised islet structure in mice where just 28% of β cells have heterotypic contacts (direct interactions/connections between different cell types), in human islets up to 80% of β cells are in direct contact with α cells and may even partially envelop them 22. Additionally, human islets have a greater proportion of α cells, and these α cells secrete significantly more GLP-1 than mouse α cells 22, 23. A recent study using advanced 3D imaging demonstrated that approximately 50% of human insulin-expressing islets lack α cells, although these islets constitute only 16% of the total islet volume 64. This indicates that the vast majority of β cells still maintained either direct or indirect contact with α cells, underscoring the relevance of intra-islet communication. In our study, we observed that 70% of islets contained α cells, all be it compared to the study by Lehrstrand et al. we studied sections only from the pancreas tail, a smaller region of interest, and performed a 2D analysis 64.

It is well described that GLP-1 enhances insulin biosynthesis, secretion, and β -cell responsiveness. In doing so, GLP-1 may also optimise the utilisation of stored insulin and stimulate the translational machinery necessary to support the increased insulin requirements of pregnancy. We propose that GLP-1 facilitates these adaptations through paracrine signalling within the islet microenvironment, enabling β cells to meet the heightened metabolic demands of pregnancy while maintaining steady insulin content. This underscores the potential role of GLP-1 as a key mediator of β -cell adaptation observed during human pregnancy.”

We thank the reviewer for the comment regarding the discussion of α cell neogenesis and α -to- β cell transdifferentiation during pregnancy. Please see our response to the later comment from the reviewer below, which addresses both these concerns.

- Caution should be applied in the interpretation of the image quantification of bihormonal cells. The possibility that the signals of glucagon and insulin overlap accidentally due to superimposed alpha and beta cells is not negligible, even in 4 μm thick sections due to interdigitations of their profiles. The additional labelling of cells for membrane markers and the employment of superresolution microscopy techniques, such as structured illumination microscopy rather than spinning disk confocal microscopy, which relative to conventional confocal microscopy reduces the bleaching of the fluorescence overtime, but does not improve resolution, would have been beneficial.

Thank you for this valuable point. We agree that the interpretation of image quantification for bihormonal cells is limited by the potential overlap of glucagon and insulin signals due to the superimposition of α and β cells using spinning disk confocal microscopy. To address this point, we have included the following statement to acknowledge this limitation and ensure that the data is interpreted in this context.

Line 837 - 840: "However, this interpretation is limited by the resolution of the imaging techniques used. Without the application of super-resolution microscopy, it is challenging to distinguish individual bihormonal cells from closely juxtaposed α and β cells."

- To maximize sensitivity and detect low-abundance targets such as the 5-HT_{2B} receptor the choice of Tyramide Signal Amplification (TSA) would have been a more powerful option than the used Vecta Fluor Excel Amplified IgG.

Thank you for this insightful suggestion. In response to the reviewer's comment, we conducted additional experiments using tyramide signal amplification (TSA) with the Tyramide SuperBoost™ Kit (Thermo Fisher Scientific, cat. no. B40941), as per the manufacturer's instructions, to enhance the detection of the 5-HT_{2B} receptor in human pancreatic sections which we have included in the methods section.

Supplementary Methods: "The 5-HT_{2B} receptor signal was amplified by tyramide signal amplification using the Tyramide SuperBoost™ Kit (Thermo Fisher Scientific, cat. no. B40941), as per the manufacturer's instructions"

The efficacy of the Tyramide SuperBoost™ Kit was validated through IHC-IF analysis of human placental sections, as shown in Supplementary Figure S11c.

Importantly, no signal was detected in β cells despite tyramide signal amplification and is presented in Figure S12. This further validates the reliability of our initial results.

Line 698 – 706: “To further validate the absence of 5-HT2B receptor expression in human β -cells from both pregnant and non-pregnant donors, we utilised tyramide signal amplification to enhance detection sensitivity in human pancreatic sections. The enhanced efficacy of the tyramide signal amplification method was verified through IHC-IF analysis of human placental sections (Supplementary Fig. S12c). Absent 5-HT2B receptor expression in human β -cells was consistently demonstrated, regardless of donor pregnancy status. Colocalisation analysis of whole FFPE sections further validated the lack of overlap between the 5-HT2B receptor and insulin across entire tissue sections even with the use of tyramide signal amplification of the 5-HT2B receptor signal (Supp. Fig. S13).”

- Again regarding the bihormonal cells, were they included in the plots for alpha and beta cells? Have they been considered when investigating the expression of PRLR and 5-HT2B in islets?

Thank you for this valuable point. As noted in our response to a previous comment, considering the limitations mentioned by the reviewer in using spinning disk confocal microscopy to analyse bihormonal cells, bihormonal cells were not included in the plots for α and β cells, nor were they considered when investigating the expression of PRLR and 5-HT2B in islets. This ensures that our analysis reflects a clear distinction between α and β cell populations.

- Placental lactogen, which is produced starting from 24 weeks of gestation, can also bind to the growth hormone receptor (GRH), although with lower affinity than to PRLR. Therefore it would have been appropriate to analyse its expression by immunomicroscopy.

We appreciate the reviewer highlighting the importance of analysing placental lactogen expression by immunomicroscopy, particularly considering its potential role in signaling through both the growth hormone receptor (GHR) and prolactin receptor (PRLR), albeit with differing affinities. While we recognise the relevance of such an analysis, we regret that we could not conduct this experiment due to limited remaining sample availability.

The samples used in our study were rare and were allocated through the nPOD biorepository, which necessitated prioritisation of experiments. We elected to use the remaining pancreatic tissue samples to address the requests for additional experiments by this as well as the other reviewers that focused on validating and strengthening the robustness of the presented data. Given the significance of ensuring the reliability and clarity of the findings we presented; we believed this approach was the most prudent course of action. We acknowledge that a follow up study focusing on immunomicroscopy for placental lactogen could provide additional insights into its interaction dynamics with GHR and PRLR during pregnancy.

- Supplementary Fig. 6. In the western blots show in panel A, the size of the molecular weight markers is missing. In the case of the PRLR HEK293T cells, could the overexpressed receptor be detected also with an antibody directed against the DDK tag? In panel c, the difference in the PRLP signal between WT and PRLP KO cells is not readily visible.

Thank you for this observation. We have addressed these points as follows:

1. Molecular weight markers have been included in the western blots shown in Supplementary Figure 6, panel A.
2. The overexpressed PRLR in HEK293T cells has been additionally validated using a western blot with an antibody directed against the DDK tag, as shown in Supplementary Figure 6, panel A. The anti-DDK tag antibody used is included in Table S2.
3. In panel C, the PRLR signal difference between WT and PRLR KO cells is now more readily appreciable following improved optimisation of the figure.

- The absence of 5-HT2B in pancreatic islet beta cells in situ could have been corroborated single molecule RNA FISH, which benefits from greater potentials for signal amplification.

Thank you for this insightful suggestion. We agree that single-molecule RNA FISH could provide additional corroboration for the absence of 5-HT2B in pancreatic islet β cells. However, while RNA FISH is technically feasible in FFPE tissues, the fixation and embedding process often leads to RNA degradation, which can limit the sensitivity for detecting low-abundance transcripts such as HTR2B. Additionally, the limited availability of pancreatic tissue samples further constrained our ability to perform this analysis. As

our primary focus was on protein expression rather than RNA, we prioritised the use of these scarce samples for protein detection. Given these challenges, we opted to employ tyramide signal amplification, as suggested by the reviewer, which validated the original findings, and we believe robustly supports our conclusion. We appreciate your understanding and value this suggestion for consideration in follow up studies.

- Serotonin can bind to multiple receptors, including 5-HTR3, which was shown to affect insulin secretion of mouse beta cells (Kim et al, 2015) and be upregulated in pregnant mice (Ohara-Imaizumi et al, 2013). Was 5-HTR3 detected in the proteomic analysis? Along this line, the expression of other serotonin receptors, and especially 5-HTR3 would have been appropriate.

Thank you for this valuable observation. 5-HT3 was not detected in the proteomic analysis. We agree that investigating the expression of other serotonin receptors, including 5-HT1D and 5-HT1F, would provide additional insights given their potential roles in insulin secretion and islet adaptations during pregnancy. In our study, we performed validation studies for several antibodies targeting these receptors. However, the high similarity in the amino acid sequences of the immunogen used to raise the commercial antibodies for each respective serotonin receptor subtype resulted in insufficient specificity of the available antibodies, as confirmed by our validation experiments. Given the limited availability of high-quality pancreatic tissue samples, we decided against proceeding with these non-specific antibodies, as this would not yield reliable data. We believe that without reliable reagents, it is not feasible to accurately quantify these receptors at this stage. We appreciate the reviewer's understanding and value this suggestion, which we hope to address in follow up studies when more specific reagents become available.

We have included this as a limitation.

Line 1036 -1041: "We acknowledge the limitation of not exploring additional serotonin receptor isoforms due to challenges with antibody specificity. Despite efforts to validate commercial antibodies for receptors such as the serotonin 1D, serotonin 1F, and serotonin 3 receptors, high sequence similarity among subtypes hindered reliable analysis. Future studies employing alternative approaches may help address this gap."

- The discussion cites evidence that ~80% of human beta cells are in contact with alpha cells (lines 832-833). In view of these data, the suggestion that the greater

alpha cell population size in pregnancy is required to better influence beta cells is rather puzzling/naïve (lines 907-909). This notion, on the other hand, is challenged by recent evidence that ~50% of human insulin-expressing islets are virtually devoid of glucagon-producing α -cells (Lehrstrand et al, 2024).

We thank the reviewer for their insightful comments regarding the role of α -cell expansion in pregnancy and its influence on β cells.

Regarding the interpretation of α -cell expansion during pregnancy, we acknowledge that attributing this solely to "better influence β cells" may oversimplify its role. As noted, approximately 80% of human β cells are already in contact with α cells under non-pregnant conditions, suggesting that increased α -cell mass may not primarily serve to enhance direct interactions with β cells. We now explore how α -cell adaptations may contribute to broader islet functionality, such as the upregulation of GLP-1, which enhances insulin secretion and supports paracrine signaling mechanisms. These adaptations suggest that α cells may play a crucial role in maintaining metabolic homeostasis during the increased metabolic demands of pregnancy.

Lehrstrand et al.'s excellent study indeed highlights that approximately 50% of insulin-expressing islets lack glucagon-producing α cells; however, it is important to note that these islets in fact accounts for just 16% of the total islet volume. Thus, in human islets, the overwhelming majority of β cells (84% of total islet volume) maintain contact with α cells either directly or through intra-islet communication. We have cited this paper in our resubmission.

To address the reviewer's concerns and strengthen our discussion, we have now included additional data from our cohort. We analysed the proportion of islets containing α cells and compared the pregnant to the non-pregnant group. We observed that approximately 70% of islets in the pancreas tail region contained α cells. While our analysis was limited to a smaller region of interest and employed 2D imaging techniques, these findings underscore the role of α -cell interactions in human islets.

Line 587 – 590: "The proportion of islets containing α cells was also analysed and no difference was observed between pregnant and non-pregnant donors ($71.71 \pm 4.97\%$ vs. $68.86 \pm 5.01\%$, $p = 0.5946$) (Fig. 4o)."

Line 908 – 929: "The increased α -cell mass we observe during human pregnancy may enhance the production and secretion of GLP-1 to support β -cell function through paracrine pathways. Importantly, increased glucagon secretion is unlikely to increase

circulating glucagon levels in human pregnancy, as plasma glucagon levels, unlike insulin, rise only intermittently during pregnancy 44, 63 and even mouse studies note low circulating and intra-islet glucagon in late gestation and no change in glucagon content or gene expression. These observations suggest that during pregnancy glucagon secretion from α cells may not play a key role in islet adaptations, however, the contribution of glucagon on intra-islet signaling in human pregnancy cannot be ruled out and requires further study 2, 22, 23.

The physiological adaptations of pregnancy impose significant metabolic challenges, necessitating coordinated islet function. The observed α -cell expansion during pregnancy likely reflects a broader role played by α cells in maintaining maternal glucose homeostasis and adapting to metabolic alterations in pregnancy. Our findings align with evidence from the literature in mice, providing reassurance that the observed adaptations in α cells in mice models are conserved in human pregnancy and may contribute to pregnancy-associated metabolic changes. This concordance supports the need for mechanistic studies to investigate the effects of enhanced α -cell mass and mediators of paracrine communication, such as GLP-1, on islets during human pregnancy. These studies would help clarify the role of α cells in maintaining metabolic balance during this unique physiological state.”

Line 890 – 897: “A recent study using advanced 3D imaging demonstrated that approximately 50% of human insulin-expressing islets lack α cells, although these islets constitute only 16% of the total islet volume. This indicates that the vast majority of β cells still maintained either direct or indirect contact with α cells, underscoring the relevance of intra-islet communication. In our study, we observed that 70% of islets contained α cells, all be it compared to the study by Lehrstrand et al. we studied sections only from the pancreas tail, a smaller region of interest, and performed a 2D analysis.”

- As mentioned above, the suggestion that alpha cells transdifferentiate into beta cells during pregnancy is merely speculative. In such a scenario, shouldn't the number of beta cells increase more and not less than alpha cells, especially in view of the limited evidence of islet cell proliferation by Ki67+ labelling? In this context, the increase of alpha cells in the periductal region of pregnant donors is mentioned (lines 812-814), also in relationship with the number of beta cells in the same region, but no evidence for this is provided (and no duct cell marker is shown). And if so, shouldn't the islet hormone positive clusters, i.e. the islet density being increased rather than remain equivalent in pregnant compared to

non-pregnant donors (Suppl. Fig. S1a)?

We thank the reviewer for their thoughtful comment regarding the mechanisms contributing to β -cell expansion during pregnancy. We acknowledge the speculative nature of α -to- β cell transdifferentiation and agree that further evidence is required to substantiate this mechanism.

In response to the reviewer's query, we have revised the text to clarify our findings and address the limitations of our study. Specifically, we discuss the observed increases in fractional and mean β -cell area during pregnancy and their consistency with prior reports. We highlight the three commonly proposed mechanisms for increasing β -cell numbers—proliferation, neogenesis, and transdifferentiation—and provide a detailed discussion of our findings in this context.

Our analysis did not detect evidence of β -cell proliferation, but as noted in the revised text, this may be due to the gestational age of our samples, which were predominantly from late gestation. Previous studies suggest that proliferation may occur earlier in pregnancy. To explore this further, we propose that additional studies using human pancreas samples from earlier gestational stages are needed to evaluate this possibility.

With regards to islet cell neogenesis, we agree with the reviewer's point that the lack of an increase in islet density in pregnancy finding makes neogenesis an unlikely explanation. We note the discrepancy with *Butler et al.* regarding islet density, attributing this to differences in methodology. *Butler et al.* used a broader definition of islets and analysed randomly selected areas of tissue, whereas we analysed entire tissue sections with stricter criteria to minimise non-specific labelling. This methodological difference may explain why we did not observe a significant increase in islet density. Furthermore, as we do not propose neogenesis as a primary mechanism, we have not included analysis of ductal structures or utilised a ductal marker in our study.

Additionally, while our study observed an increase in bihormonal cells during pregnancy which may suggest transdifferentiation as a possible mechanism of β cell mass expansion. However, we acknowledge the limitations of our imaging techniques in precisely distinguishing individual cell from closely juxtaposed α and β cells and that we did not observe a disproportionately greater increase in β cells compared to α cells, which would be expected if transdifferentiation were a major mechanism.

We have refined the text to focus on the observed changes in islet composition during pregnancy without attributing them to speculative mechanisms. We acknowledge that the data we present does not indicate which specific mechanism is responsible for the

increase in β cell number we observe, and further work is needed to elucidate the mechanism. By doing so, we aim to present a well characterised and evidence-based interpretation of the data.

Line 807 – 849: “The increases in fractional and mean β -cell area observed during pregnancy are consistent with previous reports in human pregnancy 15, 16. Notably, this expansion in β -cell area is driven by increased cell numbers rather than hypertrophy. Three mechanisms for increasing β cell numbers are commonly proposed in the literature: increased proliferation, neogenesis, and transdifferentiation from one islet cell type to β cells.

In our study, we did not detect evidence of β cell proliferation. However, β cells are noted to proliferate in early- to mid-gestation, while our analysis was limited to late gestation, with only a single donor from early pregnancy. Although β cell proliferation remains a controversial topic due to the inherently low proliferative potential of β cells in islet, further studies examining earlier gestational stages in human pancreas samples are necessary before ruling out proliferation as a mechanism for increased cell numbers.

Butler et al., analysed post-mortem pancreas samples from pregnant women, most of whom were in early to mid-gestation, increases due to an increase in islet density the rise in β cell mass occurs as a result of neogenesis¹⁵. However, unlike Butler et al., we did not observe a significant difference in islet density, which does not support the occurrence of islet neogenesis in our study. This discrepancy may stem from differences in methodology. Butler et al. defined a cluster of four or more insulin-positive cells as an islet and quantified islet density within randomly selected areas of the tissue, whereas our study involved analysis of the entire tissue section, employing stringent criteria to exclude smaller islets and minimise non-specific labelling¹⁵.

The potential for transdifferentiation from α cells to β cells has also been described in human islets 58, 59, 60. In our study, we observed bihormonal cells within islets and noted an increase in both their fractional and mean areas during pregnancy, suggesting that α -to- β cell transdifferentiation may contribute to islet plasticity. However, this interpretation is limited by the resolution of the imaging techniques used. Without the application of super-resolution microscopy, it is challenging to distinguish individual bihormonal cells from closely juxtaposed α and β cells. Additionally, we did not observe a disproportionately greater increase in β cells compared to α cells, which would be expected if transdifferentiation were a significant contributing mechanism.

Taken together, our findings are consistent with previous studies showing that β -cell mass increases during pregnancy based on an increase in cell number. However, the underlying mechanisms remain incompletely understood. Further investigation using high-quality human pancreas samples from earlier gestational stages is critical to

elucidate the contributions of proliferation, neogenesis, and transdifferentiation to islet plasticity during pregnancy.”

We hope these revisions address the reviewer’s concerns and enhance the clarity and transparency of the manuscript.

Reviewer #2 (Remarks to the Author):

Dear Reviewers,

Thank you for reviewing the manuscript and for your insightful comments.

Please find our responses to your feedback below. The original comments are displayed in bold and responses in normal text. Changes made to the manuscript are in italics and underlined. The relevant line numbers are also shown.

In addition to the untracked manuscript, a manuscript showing the tracked changes and line numbers is also provided.

This study was aimed to interrogate human pancreatic tissue and in doing so we sought the largest biorepository of globally available of pancreatic samples from pregnant women, the Network for Pancreatic Organ Donors with Diabetes (nPOD). This repository is unique in that it takes samples from patients perimortem thus it is important to stipulate that the patients analysed were not selected but instead all the available samples from pregnant women were used for the study. Entry into the biobank is not based on prespecified criteria. Entry into the biobank is based on consent and being close to death. The rarity of these samples underscores their scientific value, and they have been utilised in high-impact publications, including *Cell Metabolism*, *Nature Cell Biology*, *Diabetes*, and *Diabetologia*.

While we acknowledge that a more stringent selection of patient samples would have been ideal, the eight pancreata from pregnant women included in our study constitute the largest study of islets from pregnant women to date. Given how uncommon such specimens are, our study makes full use of all available samples, providing a unique and valuable opportunity to investigate human pancreatic adaptations in pregnancy.

In this manuscript Seedat F et al exploit unique FFPE pancreatic tissue samples from pregnant human donors and matched non-pregnant controls to characterize the adaptive changes occurring in islets during pregnancy, through both proteomics (LC-MS/MS) of laser-captured micro-dissected exocrine and islet tissue samples and morphometric analysis (by immunofluorescence staining and high-resolution imaging) of the whole tissue sections. Proteomic profiling shows a few changes between pregnant donors and matched controls and no significant differences between pregnant donors affected by gestational diabetes mellitus (GDM) and donors with a normal pregnancy. Morphometric analysis of tissue

sections identifies a 1.9-fold increase of islet area and a 4.3-, 1.9- and 3.1-fold increase of alpha, beta and bi-hormonal cell area, respectively, in pregnant donors compared to non-pregnant controls, possibly driven by an increase in islet size and cell number. The evaluation of prolactin receptor (PRLR) and serotonin 2B (5-HT2B) receptor expression by immunofluorescence and high-resolution imaging, as possible mediators of the observed islet changes, results in an increased expression of both PRLR and 5-HT2B receptor in alpha cells only. Finally, a 2.9-fold increase of glucagon-like peptide-1 (GLP-1) positive alpha cell area is also reported. Again, no significant differences are observed between GDM donors and donors with a normal pregnancy.

The work is of potential interest to the diabetes field. The main strengths are the rarity of pancreatic samples from pregnant human donors and the accurate experimental design and analysis. The main limits are the limited number of samples and the overall lack of biological validation. Thus, the manuscript could be implemented via additional context and additional in vitro experiments.

MAJOR

1. It is not clear whether the donors, classified as affected by GDM, have been diagnosed with GDM or if this status has been deducted from clinical details (lines 176-178). The authors should clarify, and eventually specify, which are the clinical details used for this classification and, if possible, the therapy followed by the donors. It is very unlikely that the comparison of 5 normal pregnant and 3 GDM pregnant donors only would give enough statistical power to draw a proper conclusion about the effect (or lack of it) of GDM in the performed evaluation but could be of interest anyway if supported by a proper donor characterization. The authors should add this limit (limited number of donors) also in the relative discussion section (line 886-899).

We thank the reviewer for their comments regarding the classification of donors and their thoughtful observations about the sample size and its potential limitations.

The classification of donors as affected by GDM was based on clinical details provided the nPOD biorepository. Specifically, donors were classified as GDM if they had a reported history of GDM diagnosis during the current pregnancy, based on standard clinical criteria. The Methods section of the manuscript has been revised to explicitly state this. Supplementary Table S1 now included all available details on the therapies received by the GDM donors.

Line 175 – 178: “Pregnant donors were classified by GDM status based on clinical details provided by the nPOD biorepository. Donors were identified as GDM if they had a documented history of GDM diagnosis during the current pregnancy and had received treatment for GDM during the current pregnancy.”

We acknowledge the reviewer’s observation regarding the limited number of donors (5 normal pregnant and 3 GDM pregnant donors), which inherently reduces the statistical power to draw robust conclusions about the effects of GDM. While this limitation does not allow us to definitively establish the role of GDM in our evaluations, we believe the findings remain of interest and provide valuable insights into pregnancy-associated islet adaptations.

To address this, we have added a discussion of the limited sample size for this comparison as a potential limitation of the study in the Discussion and emphasised the need for larger cohorts in future studies.

Line 1024 - 1028: “Furthermore, when comparing GDM to normal pregnancy, it is essential to interpret the data with caution due to the limited sample size. While the findings provide valuable preliminary insights, larger cohorts are needed to validate these observations and draw more definitive conclusions regarding the impact of GDM on islet adaptations during pregnancy.”

2. Proteomic analysis identifies four differentially expressed proteins (DEPs) when pregnant and non-pregnant donors are compared. This, of course, could be due to different reasons since human islets are heterogenic and pregnancy adds additional confounding factors with the variation of beta and alpha cell ratio (as suggested by the morphometric analysis). The additional use of a lower (but still acceptable) statistical threshold, at least for this comparison, might give some additional pieces of information about the specific proteomic changes of human islets in pregnancy, and better clarify the differences with what seen in mouse islets (as in ref 6, mentioned by the authors in the discussion section).

We thank the reviewer for their insightful comment regarding the use of a lower (but still acceptable) statistical threshold on the proteomic analysis. To further investigate potential proteomic differences between islets from pregnant and non-pregnant donors, we conducted an additional exploratory analysis using a more permissive statistical threshold (FDR < 0.15), consistent with the approach used in the proteomic study by analysing islets from pregnant mice conducted by *Horn et al.* This analysis aimed to

identify potential differences in protein abundance between pregnant and non-pregnant islets that may not have been detected under the more stringent FDR cutoff and to facilitate a direct comparison with the mouse study by *Horn et al.*

As described in the Methods, this exploratory analysis identified additional differentially expressed proteins, which are reported in Supplementary Figure S1. However, as noted in the Results, these findings did not influence the primary conclusions of the study, which remain based on the original $FDR < 0.05$ analysis. Furthermore, as discussed in the Discussion, none of the additional differentially expressed proteins identified in the exploratory analysis overlapped with those reported in the mouse study, suggesting potential species-specific differences in islet adaptation to pregnancy.

We believe this additional analysis provides useful context while maintaining the rigor of our original statistical approach. We appreciate the reviewer's suggestion, as it has allowed us to further explore proteomic changes while ensuring that our main conclusions remain robust.

Line 278 - 285: "To further explore proteomic differences between islets from pregnant and non-pregnant donors, an additional exploratory analysis was conducted using a more permissive threshold ($FDR < 0.15$), consistent with the approach used in the proteomic study by analysing islets from pregnant mice conducted by Horn et al. This analysis aimed to identify potential trends in protein expression that may not have been detected under the more stringent FDR cutoff and to allow for a direct comparison with the mouse study. However, the primary conclusions of the study are based solely on the $FDR < 0.05$ analysis."

Line 543 - 545: "An additional exploratory analysis using a more permissive statistical threshold ($FDR < 0.15$) was conducted to further investigate proteomic changes in islets from pregnant women, identifying additional differentially abundant proteins (Supp. Fig. S1)."

Line 792 - 795: "In an exploratory analysis using a more permissive threshold ($FDR < 0.15$), the same FDR threshold used in the study by Horn et al., additional differentially expressed proteins were identified. However, none of these overlapped with those reported in the mouse study."

3. At least one of the identified DEPs should be validated by immunofluorescence on the pancreatic tissue sections and its role in alpha and/or beta cell physiology in pregnancy investigated. Is this protein expressed by beta cells, alpha cells or

both? Does its expression change in pregnant donors and in which cell type? Are there any DEPs that could explain the decisive and unexpected increase of alpha cell area, especially of that positive for GLP-1? Some additional mechanistic studies are needed.

We thank the reviewer for their insightful comments and suggestions, which have helped refine our study. We have addressed the key aspects raised as follows:

To validate the LC-MS/MS findings, we performed immunofluorescence analysis of the pancreatic tissue sections (IHC-IF) for a differentially expressed protein, Cathepsin Z (CTSZ). The results demonstrate that CTSZ expression was significantly upregulated in islets from pregnant donors compared to non-pregnant controls, as shown by an increase in CTSZ-positive area relative to the total islet area and signal intensity. These findings are included in the text and supplied as Supplementary Figure S5, validate the proteomics analysis and confirm the upregulation of CTSZ in pancreatic islets during pregnancy.

Line 611 - 621: "3.5 Quantitative image analysis confirms pregnancy-induced upregulation of CTSZ in islets, validating LC-MS/MS findings"

To validate the LC-MS/MS findings, IHC-IF was performed on human islet sections using an anti-CTSZ antibody (Supp. Fig. S6a). Quantitative analysis of the CTSZ-positive area (normalised to whole islet area) and signal intensity (Supp. Fig. S6b and c) in islets from pregnant donors was compared to non-pregnant controls. A significant increase in CTSZ-positive area (0.3865 ± 0.0094 % vs. 0.3305 ± 0.0215 %, $p = 0.038$) and signal intensity (2643 ± 106.1 AU/mm² vs. 2169 ± 154.1 AU/mm²; $p=0.0295$) in pregnant donors was observed, thereby validating the LC-MS/MS-detected upregulation of CTSZ in pancreatic islets during pregnancy."

Based on the imaging findings, it is evident that CTSZ is predominantly expressed in β rather than α cells. Its upregulation in β cells during pregnancy may reflect an adaptive mechanism to support the increased metabolic demands associated with gestation. The specific functional role of CTSZ in β cells during pregnancy is a compelling area for further study.

While CTSZ is predominantly expressed in β cells and does not directly explain the observed increase in α cell area or GLP-1-positive α cells, this intriguing finding underscores the complexity of islet adaptations during pregnancy. At this stage, none of the DEPs identified in our proteomics analysis directly account for this observation. However, the interplay between β and α cells via paracrine or systemic signaling could be a potential mechanism and remains an area for future exploration.

We recognise the importance of additional mechanistic studies to clarify the role of CTSZ and other DEPs in islet adaptations during pregnancy. However, these investigations require extensive functional assays, cell-specific manipulations, and in vivo models, which are beyond the scope of the current study which focuses on identifying and validating proteomics alterations in human islets during pregnancy. Instead, our work provides an essential foundation for these future studies, which we intend to pursue to further elucidate the mechanistic pathways involved.

We hope that this response adequately addresses the reviewer's comments and appreciate the opportunity to clarify and strengthen our study.

4. Interestingly, the authors find a greater increase of alpha cell area than beta cell area in pregnant donors compared to control. They suggest that this might sustain the increase of insulin secretion seen during pregnancy, possibly through GLP-1 secretion and/or alpha to beta cell transdifferentiation. It is, however, not clear the role of the observed increased expression of PRLR and 5-HT2B in alpha cells in the above-mentioned changes. Some additional mechanistic studies in a mice alpha cell line (such as aTC1.9) or primary human alpha cells would give some hints about the topic. In addition, previous works on mice models should be better discussed and compared to the present results, such as those reported in ref 43 and 56.

We thank the reviewer for this thoughtful comment and for highlighting the potential significance of PRLR and 5-HT2B expression in α cells, as well as the suggestion to conduct additional mechanistic studies. We agree that further investigation into the role of PRLR and 5-HT2B in α cell adaptation, potentially using α cell lines such as aTC1.9 or primary human α cells, could provide valuable insights into the mechanisms underlying the observed changes. However, these studies are beyond the scope of the current manuscript, which focuses on identifying and validating key findings related to human pregnancy-associated islet adaptations. We have acknowledged the mechanistic studies would provide valuable insights into the role of the observed increases in the PRLR and 5-HT2B receptor. We will carefully consider pursuing these mechanistic experiments in a follow up study.

Line 982 - 983: "As both the 5-HT2B receptor and PRLR signaling can influence α cell function, further mechanistic studies are needed to clarify their roles in human α cells during pregnancy."

Additionally, we appreciate the reviewer's suggestion to discuss prior studies using mouse models, including references 43 and 56, and have revised the manuscript to better compare and contrast these findings with our results.

Line 851 – 874: “For the first time, we report an increase in α -cell area during human pregnancy. This finding aligns with studies by Qiao et al. and Quesada-Candela et al., which observed increases in α cell mass during mouse pregnancy. Notably, we observed a more pronounced increase in α -cell area relative to β -cell area during pregnancy, contrasting with mouse models where a proportional increase in α and β cell area are reported⁶⁰. In mouse models, the expansion of α -cell mass during pregnancy was attributed to cell proliferation rather than neogenesis or transdifferentiation. However, we did not observe evidence of α -cell proliferation during pregnancy in our study. This may be due to our focus on pancreas tissue from late-gestation, whereas the proliferation of α cells in mouse studies was predominantly observed during early to mid-gestation.

Qiao et al., demonstrate that GLP-1 from α cells plays an essential role in glucose-homeostasis during mouse pregnancy. They showed that total pancreatic GLP-1 abundance increases significantly during pregnancy in mice. Furthermore, α cell ablation in pregnant mice impaired glucose-stimulated insulin secretion (GSIS), resulting in hypoinsulinaemia and consequent disruptions in maternal glucose metabolism. Remarkably, treatment with GLP-1 restored GSIS in α -cell ablated pregnant mice. When GLP-1 receptor and glucagon receptor antagonist were added to islets from pregnant mice, GSIS was attenuated only in the presence of a GLP-1 receptor antagonist. The addition of a glucagon receptor antagonist, in contrast, had no effect. These findings underscore the critical role of GLP-1, rather than glucagon, in supporting β -cell function and glucose regulation during pregnancy. It has been shown that, similar to glucagon, human α cells also produce and secrete biologically active GLP-1, which acts locally on adjacent β cells to potentiate GSIS 3, 44, 62.”

We hope that these revisions address the reviewer's concerns and enhance the clarity and transparency of the manuscript.

MINOR

1. Line 145: This might be better defined as “ex vivo”, rather than “in vivo”. Please correct.

Thank you for this point. This has been corrected to ex vivo.

Line 141: “ex vivo”

2. Line 475-477: According to this phrase it seems that objects smaller than 10 μm have been excluded during nuclei identification and count. Since the beta cell has an average diameter of 10 μm (doi.org/10.1016/j.semcdb.2020.04.005), It is not clear what the authors intended.

Thank you for highlighting this point. To clarify, only objects larger than 10 μm in the nuclei (DAPI) channel were excluded, ensuring that artefacts or falsely detected objects in the nuclei channel were removed. As islet nuclei typically measure 5–8 μm , we believe this cutoff is appropriate. Cells themselves were not excluded based on this criterion. We have revised the text to correct this error and provide clearer explanation of this in the methods section.

Line 422 – 426: “To identify nuclei within islets, a watershed function was applied to the nuclei (DAPI) channel, followed by a size filter to exclude objects larger than 10 μm , ensuring artefacts or incorrectly detected objects were removed. This filtering process did not affect the identification or quantification of α cells or β cells.”

3. The number of islets isolated and subjected to proteomic analysis for each donor should be indicated in the methods, as well as the number of islets analysed per tissue section in the morphometric part.

Thank you for this valuable comment. The number of islets isolated and subjected to proteomic analysis for each donor, as well as the number of islets analysed per tissue section in the morphometric analysis, are now included in Supplementary Table S1.

4. Fig. 2b: It looks like an Overrepresentation Analysis rather than a Gene Set Enrichment Analysis as reported in figure caption and in the methods section. Please clarify.

The reviewer is correct. An over-representation analysis rather than gene set enrichment analysis was conducted. This has been corrected in the methods and figure caption.

Line 260 – 261: “An over-representation analysis was performed on proteins with a \log_2 fold change (FC) > 1 and FDR < 0.05, representing islet-specific proteins.”

Figure Caption: “b Overrepresentation analysis of enriched pathways from the KEGG and Reactome databases.”

5. Figure S1 is of interest for the reader and might be added to the main figure panels (maybe Fig 4?).

Thank you for this suggestion. The graphs from Figure S1 are not included in Figure 4.

6. Antibody validation (paragraph 3.6) is technically important, but it might better fit the methodology section instead of the results.

Thank you for this thoughtful suggestion. While we agree in principle that antibody validation is a crucial methodological aspect, we have chosen to retain this content within the results section. This decision was made to ensure clarity and accessibility for the reader, as presenting validation data in the methods section might disrupt the logical flow and make it less apparent how the validation of the antibodies support the experimental results. Including antibody validation in the results section allows readers to immediately evaluate the reliability of the data presented and its relevance to the study's findings. We hope the reviewer will understand this approach as an effort to maintain clarity and cohesion in the presentation of our work.

7. Paragraphs 3.7 and 3.8 might be better presented as one paragraph.

These have been merged into a single paragraph.

Line 665 – 677: “3.7 Increase abundance of PRLR and 5-HT2B receptors in α cells during human pregnancy

Immunolabeling of PRLR in both islets and exocrine tissue of pregnant and non-pregnant donors confirmed that PRLR is expressed at detectable levels in adult human pancreatic islets (Fig. 6a, Supp. Fig. S10 and S11a [secondary only control]).

Quantitative analysis of the PRLR signal intensity (as a corollary of PRLR abundance) revealed an increase in PRLR expression in α cells of pregnant donors compared to non-pregnant controls (302.3 ± 10.12 AU/mm² vs. 267.7 ± 4.582 AU/mm²; $p=0.0398$), whereas no difference in PRLR expression was observed in β cells of pregnant donors (277.4 ± 3.002 AU/mm² vs. 270.3 ± 1.704 AU/mm²; $p = 0.0610$) (Fig. 6b and c). No significant differences in PRLR expression was observed between donors with GDM and normal pregnancies (Supp. Fig. S11b and c).

5-HT2B receptor expression was detected in α cells in both pregnant and non-pregnant donor samples. Outside islets, 5-HT2B receptor expression was detected in peri-ductal

regions (Fig. 7a and Supp. Fig. S12a [secondary only control]). Signal intensity quantifications demonstrate an increase in the expression of 5-HT2B receptors in α cells of pregnant donors compared to non-pregnant controls (387.4 ± 10.01 Au/mm² vs. 286.5 ± 13.46 Au/mm²; $p=0.0359$) (Fig. 7b). No significant differences in 5-HT2B receptor expression (Supp. Fig. S12b) was observed between donors with GDM and normal pregnancies.”

8. Figures 7.a and 8.a should also show the staining of at least one non-pregnant donor sample. In addition, the ref to Fig 8.c should be added at line 722.

Thank you for raising this. The staining of a non-pregnant donor sample is included for Figures 7a and 8a.

Figure 8c has been included at line 722.

Line 692 - 693: “Colocalisation analysis confirmed absent colocalisation between 5-HT2B receptor and insulin across whole tissue sections (Fig. 8b and 8c).”

9. Line 890-891: The phrase is not clear; may the authors clarify?

Thank you for highlighting this. This is now clarified.

Line 986 - 990: “Clinical data show no significant differences in first-phase insulin response or insulin secretion throughout pregnancy between women with GDM and those without GDM and who have no risk factors for hyperglycaemia. However, when comparing pregnant women with GDM to non-diabetic pregnant women with risk factors for hyperglycaemia, GDM was associated with reduced insulin secretion.”

10. Please be aware that Kolic J et al (doi:10.1016/j.cmet.2024.06.001) has recently published a manuscript in which almost 8000 islet proteins are identified. The authors may want to correct sentences such as those at line 777-778 and 932-933, accordingly.

Thank you for highlighting this paper. We have cited this important paper in the discussion and amended the sentences at line 777-778 and 932-933.

Line 760 - 765: “A recent study by Kolic et al. analysing fresh-frozen islets from 90 donors by LC-MS/MS, both with and without type 2 diabetes, identified an average of

8,000 proteins⁴⁹. Even with the use of FFPE pancreatic tissue, we successfully detected over 7,000 unique proteins per sample, emphasising the capability of FFPE tissues for deep proteomic studies and identifying the highest number of proteins identified from FFPE human pancreatic islets to date.

Line 1049 – 1051: “In conclusion, we successfully characterised the proteome of human pancreatic islets during pregnancy, creating the largest dataset of pancreatic islet and exocrine proteins derived from pancreata from pregnant women to date.”

Reviewer #3 (Remarks to the Author):

Dear Reviewers,

Thank you for reviewing the manuscript and for your insightful comments.

Please find our responses to your feedback below. The original comments are displayed in bold and responses in normal text. Changes made to the manuscript are in italics and underlined. The relevant line numbers are also shown.

In addition to the untracked manuscript, a manuscript showing the tracked changes and line numbers is also provided.

This study was aimed to interrogate human pancreatic tissue and in doing so we sought the largest biorepository of globally available of pancreatic samples from pregnant women, the Network for Pancreatic Organ Donors with Diabetes (nPOD). This repository is unique in that it takes samples from patients perimortem thus it is important to stipulate that the patients analysed were not selected but instead all the available samples from pregnant women were used for the study. Entry into the biobank is not based on prespecified criteria. Entry into the biobank is based on consent and being close to death. The rarity of these samples underscores their scientific value, and they have been utilised in high-impact publications, including *Cell Metabolism*, *Nature Cell Biology*, *Diabetes*, and *Diabetologia*.

While we acknowledge that a more stringent selection of patient samples would have been ideal, the eight pancreata from pregnant women included in our study constitute the largest study of islets from pregnant women to date. Given how uncommon such specimens are, our study makes full use of all available samples, providing a unique and valuable opportunity to investigate human pancreatic adaptations in pregnancy.

This study utilises precious human donor pancreas formalin-fixed paraffin embedded (FFPE) tissue blocks from nPOD from pregnant (n=7) and moderately well matched non-pregnant (n=7) women with the aim of determining the mechanisms of human islet adaptation to pregnancy. From laser captured exocrine and islet tissue, proteomics was performed which was analysed using various bioinformatics processes. Pancreas sections were also immunostained for insulin, glucagon, prolactin receptor (PRLR), serotonin receptor 2B (HTR2B) and GLP-1, and images were systematically analysed using Arivis Vision4D software. The strengths of the work are the high quality nPOP pancreas FFPE blocks, high quality immunohistochemistry analyses of pancreas with extensive validation of

the quality of the antibodies used and the methodology used for analysis using the Arivis software.

The data from the IHC studies showing increased alpha-cell area and increased expression of GLP-1 in alpha cells in pancreases of pregnant women is of particular interest.

A major achievement was successful laser microcapture of endocrine and exocrine tissue and proteomic analyses of the tissue obtained in which endocrine tissue was easily distinguished from exocrine tissue. This methodology, however, showed minimal differences between the pregnant and non-pregnant islet proteomes. This seems surprising considering the marked increases in both alpha and beta cells within the islets, suggesting that the methodology using FFPE tissue material may be limiting (easy to differentiate endocrine from exocrine tissue, but challenging to detect differences within endocrine tissue). Weaknesses include exclusion of small islets from the analyses, a failure of the proteomics to detect the PRLR and HTR2B despite it being detected in exocrine and endocrine IHC analyses. Further, only one protein within serotonin signalling has been focussed on. Another limitation that is unavoidable with the availability of samples is that there was only one first trimester pancreas and no second trimester pancreases available for analysis (islet adaptations which occur earlier in pregnancy may have been missed).

Major comments:

1) The matching of the subjects showed BMI to be higher in the pregnant women (almost significant despite low sample size). Was this a pre-pregnancy or pregnancy BMI?

Thank you for this question. Pre-pregnancy BMI measurements were not available, as the BMI provided is specific to pregnancy. The nPOD biorepository only supplies pregnancy BMI data to investigators, and due to the unique circumstances under which pancreatic tissue is obtained from donors, it is not possible to access pre-pregnancy BMI measurements. This clarification has been added to the Methods, and this has been cited as a limitation in the Discussion.

Line 181 – 182: “For pregnant donors, only the pregnancy BMI is provided.”

Line 1020 – 1024: “For pregnant donors, only the pregnancy BMI is available, as the nPOD biorepository does not provide pre-pregnancy BMI. This limitation further complicates the interpretation of BMI-related effects on islet adaptations during pregnancy.”

2) Table 1 suggests that 2 of the non-pregnant subjects were GAD Ab positive, whereas Suppl Table 1 suggests the GAD Ab subjects were in the pregnant group. Please check the data and correct.

Thank you for highlighting this error. This has been corrected in Table 1 to reflect that the GAD Ab positive donors belonged to the pregnant group.

3) Did the GAD Ab positive pancreases show any evidence of insulinitis, and could this have affected the results? This should be reported in the manuscript.

Thank you for this question. Low-grade insulinitis was detected in the Anti-GAD antibody-positive donors, as noted in the manuscript. These findings are now reported in the manuscript (Table 1) and have been included in the Discussion as limitations which add to the heterogeneity of the cohort.

Line 1018 – 1020: “Variability in ethnicity, GDM status, anti-GAD antibody status, and insulinitis among donors may have contributed to the few observed differences in proteomic profiles and may have masked more subtle biological patterns.”

4) I have concerns about the proteomics findings of pregnant vs non-pregnant islets. Validation of the proteomics of human islets obtained from formalin fixed paraffin embedded sections against fresh human islets (do not need to be pregnant subjects) would be helpful. Are some of the proteins of interest (e.g. PRL receptor, serotonin receptors, tryptophan hydroxylases) better detected in fresh islets, as opposed to FFPE islet tissue?

We appreciate the reviewer’s thoughtful suggestion regarding validation of our proteomic findings in fresh human islets. While fresh islet validation is beyond the scope of our current study due to resource constraints, we have assessed the overlap between our dataset and a recent study by *Kolic et al.*, which performed proteomics on fresh human islets from donors with and without type 2 diabetes patients. **Notably, 88% of the proteins detected in our FFPE-derived islets were also identified in *Kolic et***

al.'s fresh islet dataset, underscoring the robustness and biological relevance of our findings using FFPE islet tissue.

Interestingly, despite the use of fresh islets, *Kolic et al.* also did not detect the prolactin receptor (PRLR), serotonin 2B (5-HT2B) receptors, or tryptophan hydroxylases, suggesting that the absence of these proteins in our dataset may not be solely due to formalin fixation but could reflect intrinsic challenges in detecting these proteins using mass spectrometry or their low abundance in human islets. We have now incorporated this discussion into our manuscript to provide additional context. Future studies employing targeted proteomics approaches may help to further elucidate the expression and regulation of these proteins in human islets during pregnancy.

Line 766 – 776: “A potential concern of using FFPE islets for proteomic analysis is that protein detectability may be impacted. To assess the reliability of our dataset, we compared our dataset with that obtained by Kolic et al. who analysed fresh islets by LC-MS/MS. We found that 88% of the proteins detected in our dataset were also identified in Kolic et al.’s study, demonstrating strong consistency between FFPE-derived and fresh islet proteomes. Notably, Kolic et al. also did not detect the PRLR or 5-HT2B receptor, suggesting that the absence of these proteins in our dataset may not be solely due to FFPE preservation but could also reflect inherent challenges in detecting these proteins using mass spectrometry or their low abundance in human islets. Future studies employing targeted proteomics approaches may help to further elucidate the regulation and functional relevance of these proteins in human islets during pregnancy.”

5) Why has so much focus been placed on HTR2B? It was the tryptophan hydroxylases that had much increased expression in pregnant compared to non-pregnant islets in mice. Also could other serotonin receptor isoforms be more important for pregnancy adaptation in human islets?

Thank you for this insightful comment. As we observed an increase in β cell area associated with an increase in cell number, we chose to study the 5-HT2B receptor based on previous mouse studies, which have identified the 5-HT2B receptor as a key receptor associated with β cell proliferation. This informed our decision to prioritise the 5-HT2B receptor in our investigations.

We do, however, acknowledge the valuable point raised by the reviewer regarding the potential roles of other serotonin receptor isoforms in human islet adaptations during pregnancy. To address this, we performed antibody validation for several commercial targeting additional serotonin receptors, including 5-HT1D and 5-HT1F, which are implicated in insulin secretion and islet function. Unfortunately, despite our efforts, our

validation experiments failed to demonstrate sufficient specificity of the available commercial antibodies. This is likely due to the high similarity in amino acid sequences of the immunogens used to raise antibodies against the 5-HT1D, 5-HT1E, and 5-HT1F receptor subtypes.

Given the limited availability of high-quality pancreatic tissue samples, we decided not to proceed with these non-specific antibodies, as they would not produce reliable data. We agree that further investigation of these receptors would provide valuable insights and hope to address this question in future studies when more specific reagents become available. We appreciate the reviewer's understanding and thank them for this thoughtful suggestion.

We have included this as a limitation.

Line 1036 -1041: "We acknowledge the limitation of not exploring additional serotonin receptor isoforms due to challenges with antibody specificity. Despite efforts to validate commercial antibodies for receptors such as the serotonin 1D, serotonin 1F, and serotonin 3 receptors, high sequence similarity among subtypes hindered reliable analysis. Future studies employing alternative approaches may help address this gap."

6) HTR2B was shown to be present in human islet beta-cells by IHC previously (Diabetologia 2016 59:744–754). Why the discrepancy between this study and that previous study? Please discuss.

Thank you for this important comment. As we already noted in the discussion, 5-HT2B receptor expression in human islets was previously demonstrated using a polyclonal antibody (Diabetologia 2016 59:744–754). Polyclonal antibodies, while valuable, are raised against multiple epitopes and are prone to antigen cross-reactivity, which can limit specificity. In our study, we addressed this by employing a monoclonal antibody clone, which recognises a single epitope, thereby improving specificity. We also performed thorough antibody validation, to ensure reliable detection of the 5-HT2B receptor.

Furthermore, methodological differences such as tissue processing, staining protocols, or experimental conditions may have contributed to the discrepancy between our findings and those of the earlier study. By addressing these potential sources of variability and employing robust validation, we are confident in the reliability of our findings. We have expanded on the original explanation in the Discussion in the manuscript.

Line 954 –967: “5-HT2B receptor expression in human islets has been previously demonstrated using a polyclonal antibody 68. Given that polyclonal antibodies are raised against multiple epitopes, their limitations in antigen specificity are well known 69. To overcome this, we used a monoclonal antibody clone, which offers greater specificity due to its recognition of a single epitope. Additionally, we performed thorough antibody validation, including immunoblotting of recombinant 5-HT2B receptor protein and cell lysates known to express or lack 5-HT2B receptor expression. Validation also included IHC-IF analysis of HAP1 cell pellets and cancerous tissues known to express the 5-HT2B receptor, confirming that the antibody reliably detects the target protein. Differences in antibody specificity and validation methods between studies may account for the observed discrepancy. Furthermore, variations in tissue processing, staining protocols, or experimental conditions could also contribute to differences in reported 5-HT2B receptor expression. Our approach aimed to minimise such variability, providing robust evidence for our findings.”

7) Butler et al (Diabetologia 53, 2167-2176 (2010)), showed that there was a marked increase in small islets (including islet cell clusters) in human pregnancy, and that this could be from neogenesis from pancreatic ductal tissue. Why were small islets excluded from this study, when previously it was considered the small islets were important in islet adaptation to human pregnancy?

We appreciate the reviewer’s comment regarding the potential contribution of small islets and islet cell clusters to β -cell adaptation in pregnancy, as highlighted by *Butler et al.* (2010). In our study, we applied a size threshold of $\geq 1000 \mu\text{m}^2$ to exclude small structures, as this was necessary to ensure the specificity of the segmentation and the quantitative image analysis of well-defined islet structures. While small islets may contribute to β -cell mass expansion, distinguishing them from non-specific staining or isolated endocrine cells in our dataset posed a technical challenge. Thus, for scientific rigor and reproducibility, we maintained this threshold to ensure robust and interpretable data.

To further explore this point, we conducted an exploratory analysis using a lower size threshold ($\geq 50 \mu\text{m}^2$) and assessed islet density, as this metric is most informative for detecting islet neogenesis, particularly the formation of new small islets. This analysis did not reveal a significant difference in islet density between pregnant and non-pregnant donors (pregnant: 5.864 ± 1.145 islets per mm^2 vs. non-pregnant: 4.383 ± 0.6895 islets per mm^2 , $p = 0.2895$). Given this, and to ensure consistency in our approach, we maintain our original analysis while acknowledging that future studies

employing alternative imaging strategies may further clarify the role of small islets in human pregnancy.

8) The methodology section is very detailed which is helpful, but some of the detail could be moved to the supplemental data section.

Thank you for this comment. As suggested by the reviewer, we have moved some of the details to a Supplementary Methods section. In doing so, we ensured that all key points necessary for replicating the work remain in the main Methods section. The techniques moved to the Supplementary Methods are well-established, routine methods commonly used in the field, or additional details that, while not critical, provide further context for interested readers. Their inclusion in the supplementary section allows us to streamline the main text while maintaining clarity and reproducibility.

9) The discussion needs to cover off on the limitations of the study better.

We appreciate the reviewer's feedback regarding the need to expand the discussion of the study's limitations. In response, we have revised the discussion to comprehensively address the key constraints of our study, as outlined below:

Regarding the sample size and cohort heterogeneity, we now explicitly discuss how the limited availability of high-quality pancreatic tissue from pregnant women constrained our sample size and introduced heterogeneity into the cohort. Variability in factors such as ethnicity, GDM status, anti-GAD antibody status, and insulinitis among donors is acknowledged as a potential source of variability in the proteomic profiles, potentially masking more subtle biological patterns.

The limited sample size for the comparison of GDM and normal pregnancy is emphasised as a significant limitation. We caution that the findings, while providing preliminary insights, require validation in larger cohorts to draw more definitive conclusions about the impact of GDM on islet adaptations.

We highlight that only pregnancy BMI was available for this study due to the nature of tissue collection, which precludes the acquisition of pre-pregnancy BMI measurements. This limitation complicates the interpretation of BMI-related effects on islet adaptations during pregnancy and is an area requiring further investigation.

The limitations of islet diameter as a metric due to the irregular shapes of islets are addressed, and we explain why islet area is a more reliable measure in this context.

We also acknowledge that not exploring additional serotonin receptor isoforms is a limitation of this study.

These points have been incorporated into the discussion to provide a thorough and transparent overview of the study's limitations and to suggest directions for future research. We hope these additions adequately address the reviewer's concerns.

We hope that these revisions address the reviewer's concerns and enhance the clarity and transparency of the manuscript.

Line 1016 - 1041: "The limited availability of high-quality pancreatic tissue from pregnant women significantly constrained the sample size in this study and introduced heterogeneity within the cohort. Variability in ethnicity, GDM status, anti-GAD antibody status, and insulinitis among donors may have contributed to the few observed differences in proteomic profiles and may have masked more subtle biological patterns. For pregnant donors, only the pregnancy BMI was available. Due to the circumstances under which donor tissues are obtained, it is not possible to acquire pre-pregnancy BMI measurements, and this limitation complicates the interpretation of BMI-related effects on islet adaptations during pregnancy. Furthermore, when comparing GDM to normal pregnancy, it is essential to interpret the data with caution due to the limited sample size. While the findings provide valuable preliminary insights, larger cohorts are needed to validate these observations and draw more definitive conclusions regarding the impact of GDM on islet adaptations during pregnancy. The heterogeneity of the cohort highlights a broader challenge in the field of pregnancy-related islet adaptations. Greater collaboration between researchers, to increase the availability of pancreatic tissue from pregnant women, in order to study more homogenous cohorts is critical to advancing our understanding of islet adaptations during pregnancy and improving outcomes in GDM research. Future research should focus on studying pancreatic tissue from different stages of gestation; however, high-quality first- and second-trimester pancreas tissue were not available for this study. Measures of islet diameter are limited by the irregular shapes of islets, making islet area a more reliable metric for measurement. We acknowledge the limitation of not exploring additional serotonin receptor isoforms due to challenges with antibody specificity. Despite efforts to validate commercial antibodies for receptors such as the serotonin 1D, serotonin 1F, and serotonin 3 receptors, high sequence similarity among subtypes hindered reliable analysis. Future studies employing alternative approaches may help address this gap."

Overall, it is worthwhile making the point of the importance not to extrapolate too much from small animal studies to humans in determining the mechanisms of islet adaptation to pregnancy. However, the evidence from this study, as it

stands, is not strong enough to exclude important roles for lactogenic hormones and serotonin signaling in the adaptation of human islets to pregnancy.

Thanks for this comment. We agree with the reviewer's valuable point and have incorporated these points in the conclusion.

Line 1057 - 1065: "Our findings suggest that human islet adaptations during pregnancy differ to those observed in mice, however the role of lactogenic hormones and serotonin signaling cannot be excluded. The marked increases in α cell area and GLP-1 abundance suggest that β cells may have a greater reliance on intra-islet paracrine signaling during human pregnancy. This study provides novel data on pancreatic islet alterations during pregnancy and underscores the importance of cautious extrapolation from small animal studies and reinforces the value of using human-based model systems or tissues for pancreatic islet research."

Reviewer #1 (Remarks to the Author):

Dear Reviewers,

Thank you for reviewing the manuscript and for your insightful comments.

Please find our responses to your feedback below. The original comments are displayed in bold and responses in normal text. Changes made to the manuscript are in italics and underlined. The relevant line numbers are also shown.

In addition to the untracked manuscript, a manuscript showing the tracked changes and line numbers is also provided.

This study was aimed to interrogate human pancreatic tissue and in doing so we sought the largest biorepository of globally available of pancreatic samples from pregnant women, the Network for Pancreatic Organ Donors with Diabetes (nPOD). This repository is unique in that it takes samples from patients perimortem thus it is important to stipulate that the patients analysed were not selected but instead all the available samples from pregnant women were used for the study. Entry into the biobank is not based on prespecified criteria. Entry into the biobank is based on consent and being close to death. The rarity of these samples underscores their scientific value, and they have been utilised in high-impact publications, including *Cell Metabolism*, *Nature Cell Biology*, *Diabetes*, and *Diabetologia*.

While we acknowledge that a more stringent selection of patient samples would have been ideal, the eight pancreata from pregnant women included in our study constitute the largest study of islets from pregnant women to date. Given how uncommon such specimens are, our study makes full use of all available samples, providing a unique and valuable opportunity to investigate human pancreatic adaptations in pregnancy.

This manuscript reports about the proteomic profile of human pancreas exocrine and endocrine tissue in pregnant women compared to non-pregnant women. These analyses are complemented by image analyses pertaining islet size distribution, alpha, beta and bi-hormonal cell composition as well as the expression of PRLR and 5-HT2A receptors, which have been implicated in the signalling cascade promoting proliferation of beta cells in pregnant mice.

The large proteomic data sets are novel and intriguing, being from a rare, unique cohort of pregnant women and the most extensive collected so far from human islets in situ. The lack of pronounced proteomic changes between islets of

pregnant women compared to non-pregnant women is especially remarkable. The observation that pregnancy is associated with a greater expansion of the alpha cell compartment relative to the one of beta cells is also a valuable contribution.

Despite these merits, the study has significant conceptual and methodological shortcomings, with some apparent internal inconsistencies.

Specific comments

- The comparison between the proteomic profiles islets of islets from pregnant and non-pregnant women is limited by the heterogeneity of the donors. Among the 7 women in the third semester of pregnancy, 4 were not known to suffer from gestational diabetes, although in one case (6310) HbA1c levels were unknown, 2 had gestational diabetes but were negative for islet autoantibodies, while one (6405) had type 1 diabetes, being positive for >2 anti-islet autoantibodies (GADA, IA2A, ZnT8A). The occurrence of beta cell autoimmunity is likely to have an impact on the outcome of the analysis. Moreover, one of the non-pregnant donors (6401) had an HbA1c of 5.8, meaning she had impaired glucose tolerance, while in another case (6253), who had the highest BMI, HbA1c levels were unknown, i.e., the occurrence of glucose intolerance cannot be ruled out. Further differences in ethnicity and especially in the BMI among pregnant and not pregnant women are additional confounders which likely account for the meagre number of islet proteins differentially expressed in pregnant versus non-pregnant women. These considerations, together the very limited power of the analysis between normal pregnancy and GDM donors, greatly temper the conclusions which can be drawn from the data (e.g., lines 589-590; 649-650; 703-704 and so on).

Thank you for this insightful comment from the reviewer. We agree with the reviewer and acknowledge that the heterogeneity among donors, including differences in GDM status, β -cell autoimmunity, BMI, and HbA1c levels, is a limitation of this study. These factors may have contributed to variability in the proteomic profiles and limited the ability to identify significant differences between pregnant and non-pregnant donors.

Due to the rarity of obtaining such unique human islet samples, the sample size is inherently limited, which we have now explicitly acknowledged in the discussion. Based on the reviewers comments we have tempered the conclusions accordingly to reflect the constraints of the study and highlighted the need for larger, more homogeneous cohorts in future investigations. Despite these limitations, the study provides valuable preliminary insights into islet adaptations during pregnancy.

Line 1016 – 1033: “The limited availability of high-quality pancreatic tissue from pregnant women significantly constrained the sample size in this study and introduced heterogeneity within the cohort. Variability in ethnicity, GDM status, anti-GAD antibody status, and insulinitis among donors may have contributed to the few observed differences in proteomic profiles and may have masked more subtle biological patterns. For pregnant donors, only the pregnancy BMI was available. Due to the circumstances under which donor tissues are obtained, it is not possible to acquire pre-pregnancy BMI measurements, and this limitation complicates the interpretation of BMI-related effects on islet adaptations during pregnancy. The heterogeneity of the cohort highlights a broader challenge in the field of pregnancy-related islet adaptations. Greater collaboration between researchers to increase the availability of pancreatic tissue from pregnant women to study more homogenous cohorts is critical to advancing our understanding of islet adaptations during pregnancy and improving outcomes in GDM research.”

- To assess the representative value of the analysis, it would be important to clarify whether the LCM area of 235,000 μm^2 tissue/donor was the collective surface of both endocrine and exocrine tissue, and in this case how much was the islet fraction, or of each part of the pancreas. Moreover, it is not clear, which part of the exocrine tissue was collected (acinar cells, ductal cells, both?). Along the same line, it would be important to specify the average number of islets analysed for each donor and whether these were collected from serial sections or from sections distanced $>200 \mu\text{m}$ from each other, i.e. most likely representing distinct islets.

Thank you for this important point. This is now clarified in the methods.

Line 191 – 197: “For each donor, 235,000 μm^2 of islet tissue and 235,000 of μm^2 exocrine tissue, consisting primarily of acinar cells, were microdissected. The exocrine tissue served as an internal control to verify the purity of islet tissue isolation. A single tissue section was utilised for each donor, islets were selected at random, and care was taken to ensure that the dissected islets were separated by a minimum distance of at least $> 200 \mu\text{m}$, to ensure that each dissected region represented a distinct islet.”

Line 489 – 490: “On average 10 islets per donor were analysed (Supp. Table S1).”

Supplementary Table S1 has been updated to include the number of islets analysed per donor.

- Knowledge about the number of individual islets analyzed for each donor is also important to assess how strong the evidence for the lack of Ki67 detection in the islets (Fig. 5) is, as well as for the other markers (PRPL, Fig. 6; 5-HT2B, Fig. 7 and related figures). Moreover, islet diameter is not a good approach to estimate the islet size, since islets are irregular in shape. The measurement of the whole islet area is a better way to estimate changes in islet size.

Thank you for this valuable comment. The number of islets isolated and subjected to proteomic analysis for each donor, as well as the number of islets analysed per tissue section in the morphometric analysis and expression of Ki67 and PRLR, are now included in Supplementary Table S1.

Furthermore, the reviewer raises a valuable point. To address this, we have used whole islet area rather than islet diameter to measure islet size in tissue sections. While we have included data describing the longest axis of islet diameter, we have also noted that this metric is limited due to the irregular shapes of islets.

Line 1035 - 1036: "Measures of islet diameter are limited by the irregular shapes of islets, making islet area is a more reliable metric for measurement."

- Supplement Fig. 4k shows that the fraction of beta cells is not changed, and it accounts approx. 45% of the total islet cell mass. What is the remaining 55%?

We thank the reviewer for their thoughtful comment. Our analysis of islet cell composition was conducted using IHC-IF and an automated image analysis pipeline, with normalisation to whole islet area to ensure a systematic and unbiased quantification approach. In addition to the contribution from β cells to islet area, we observed an α -cell fraction of 5-12.5% and identified 2.5-5% bihormonal (insulin+ glucagon+) cells. The remaining 55% comprises constituents that were not explicitly examined, as they were beyond the focus of this study. These likely include contributions from other endocrine cell types, connective tissue, vasculature, and stromal components, which were not specifically measured. Several factors may contribute to the observed proportions. First, our normalisation to whole islet area may result in proportions differing from those based on absolute cell counts. Second, our automated image analysis pipeline, while rigorous, is inherently influenced by segmentation and classification parameters. Importantly, our approach provides a reproducible and quantitative assessment of islet composition, and we are confident in the robustness of our findings. Future work incorporating additional markers for δ -cells and refining classification parameters may help further dissect the contributions of different endocrine populations. We appreciate the reviewer's insight and believe our

findings add to the understanding of islet adaptations during pregnancy.

- The methods section does not include information on T47D cells. Also, it is unclear whether human EndoC-bH3 cells were treated with tamoxifen to remove the cassette with the SV40 large T-antigen. If not, such cells keep proliferating and thus their comparison to adult human beta cells is less stringent.

Thank you for highlighting this. These points have now been clarified in the Methods and Supplementary Methods section.

Line 331 - 332: "Prepared FFPE T47D cell pellets were purchased from Amsbio, UK (cat. no. 3010-0620)."

Supplementary Methods: "Cells were differentiated by adding 4-hydroxitamoxifen (1 μ L/10 mL) to culture media for 21 days."

- As only four islet proteins were found to change in their expression the authors conclude that human islets, unlike mouse islets, undergo few proteomic changes during pregnancy. However, the interpretation of the proteomic data is hindered by the provision of the raw data only without normalization. Moreover, inter-individual differences are large, conceivably due to the limited size of the cohort, and likely accounting for the meagre number of islet proteins found to be differentially expressed in pregnancy. The conclusion regarding the lack of islet proteomic changes during pregnancy should be further tempered, since for obvious reasons protein flux could not be quantified. For instance, in pregnancy the islet levels of insulin were unchanged but given its 2-fold increased secretion, its translational rate is likely to be incremented. Such an adaptive mechanism, which is not considered by the authors, seems more plausible than the neogenesis of alpha cells, which is not adequately documented (see comments below), and their transdifferentiation into beta cells, which remains speculative.

We thank the reviewer for this important comment.

As part of the default parameters, DIA-NN does apply a retention time-dependent normalisation alongside the commonly used MaxLFQ quantitation algorithm, which also includes a data normalisation step. We recognise that this may not have been clear in the original text. To address this, we have revised the text (Methods) to explicitly clarify that these steps are part of the default parameter settings and cited the following paper

by Cox *et al*: “Accurate Proteome-wide Label-free Quantification by Delayed Normalization and Maximal Peptide Ratio Extraction, Termed MaxLFQ”.

Line 224 – 228: “DIA-NN incorporates retention time-dependent normalisation as part of its default parameters, along with the commonly used MaxLFQ quantitation algorithm, which also includes a data normalisation step. These normalisation processes are integral to the default settings of the software and ensure consistency in the analysis.”

As outlined in our response to the reviewer’s comment above, we now clearly acknowledge the sample size and interindividual variability of the cohort as a limitation.

Line 1016 – 1020: “The limited availability of high-quality pancreatic tissue from pregnant women significantly constrained the sample size in this study and introduced heterogeneity within the cohort. Variability in ethnicity, GDM status, anti-GAD antibody status, and insulinitis among donors may have contributed to the few observed differences in proteomic profiles and may have masked more subtle biological patterns.”

In response, we have revised the Discussion.

We have tempered our conclusion regarding the limited number of differentially expressed proteins detected and acknowledge that we have not presented data on dynamic protein changes, which could provide further insights into islet adaptation during pregnancy.

Line 800 – 805: “While human islets exhibit significant functional adaptation, as demonstrated by their ability to double insulin secretion by late gestation, our findings demonstrate that human islets undergo few proteomic changes during pregnancy. However, we acknowledge that dynamic proteomic changes, which may play a role in islet adaptations during pregnancy, may be masked as we do not present data on protein flux.”

We thank the reviewer for the thoughtful comment regarding the adaptive mechanisms of β cells during pregnancy. We agree that the increased translational activity of β cells, rather than neogenesis of α cells, is a plausible explanation for meeting the heightened insulin demands of pregnancy.

As outlined in our revised text below, we discuss that as whole islet insulin levels were changed during pregnancy, the high insulin demands of pregnancy are met through alternate adaptive mechanisms. We propose that GLP-1 paracrine signaling may play a

key role in optimising the utilisation of stored insulin and stimulating the translational machinery necessary to support these increased requirements.

Line 876 – 906: “During pregnancy, the increased demand for insulin to maintain glucose homeostasis presents a significant metabolic challenge. Despite this, total islet insulin levels remain unchanged in proteomic analysis, suggesting that β cells adapt by enhancing their production and secretion rates without accumulating excess insulin stores. Such an enhanced functional output likely relies on regulatory signals. Like Qiao et al., we too observe an increase in GLP-1 abundance in α cells during human pregnancy and propose that this increase in GLP-1 may play a role in supporting β -cell function.

Human islets are anatomically uniquely designed to facilitate paracrine communication, as the different hormone-secreting cells within islets are more intermixed than in mice 20, 21, 22, 23. For example, unlike the organised islet structure in mice where just 28% of β cells have heterotypic contacts (direct interactions/connections between different cell types), in human islets up to 80% of β cells are in direct contact with α cells and may even partially envelop them 22. Additionally, human islets have a greater proportion of α cells, and these α cells secrete significantly more GLP-1 than mouse α cells 22, 23. A recent study using advanced 3D imaging demonstrated that approximately 50% of human insulin-expressing islets lack α cells, although these islets constitute only 16% of the total islet volume 64. This indicates that the vast majority of β cells still maintained either direct or indirect contact with α cells, underscoring the relevance of intra-islet communication. In our study, we observed that 70% of islets contained α cells, all be it compared to the study by Lehrstrand et al. we studied sections only from the pancreas tail, a smaller region of interest, and performed a 2D analysis 64.

It is well described that GLP-1 enhances insulin biosynthesis, secretion, and β -cell responsiveness. In doing so, GLP-1 may also optimise the utilisation of stored insulin and stimulate the translational machinery necessary to support the increased insulin requirements of pregnancy. We propose that GLP-1 facilitates these adaptations through paracrine signalling within the islet microenvironment, enabling β cells to meet the heightened metabolic demands of pregnancy while maintaining steady insulin content. This underscores the potential role of GLP-1 as a key mediator of β -cell adaptation observed during human pregnancy.”

We thank the reviewer for the comment regarding the discussion of α cell neogenesis and α -to- β cell transdifferentiation during pregnancy. Please see our response to the later comment from the reviewer below, which addresses both these concerns.

- Caution should be applied in the interpretation of the image quantification of bihormonal cells. The possibility that the signals of glucagon and insulin overlap accidentally due to superimposed alpha and beta cells is not negligible, even in 4 μm thick sections due to interdigitations of their profiles. The additional labelling of cells for membrane markers and the employment of superresolution microscopy techniques, such as structured illumination microscopy rather than spinning disk confocal microscopy, which relative to conventional confocal microscopy reduces the bleaching of the fluorescence overtime, but does not improve resolution, would have been beneficial.

Thank you for this valuable point. We agree that the interpretation of image quantification for bihormonal cells is limited by the potential overlap of glucagon and insulin signals due to the superimposition of α and β cells using spinning disk confocal microscopy. To address this point, we have included the following statement to acknowledge this limitation and ensure that the data is interpreted in this context.

Line 837 - 840: "However, this interpretation is limited by the resolution of the imaging techniques used. Without the application of super-resolution microscopy, it is challenging to distinguish individual bihormonal cells from closely juxtaposed α and β cells."

- To maximize sensitivity and detect low-abundance targets such as the 5-HT_{2B} receptor the choice of Tyramide Signal Amplification (TSA) would have been a more powerful option than the used Vecta Fluor Excel Amplified IgG.

Thank you for this insightful suggestion. In response to the reviewer's comment, we conducted additional experiments using tyramide signal amplification (TSA) with the Tyramide SuperBoost™ Kit (Thermo Fisher Scientific, cat. no. B40941), as per the manufacturer's instructions, to enhance the detection of the 5-HT_{2B} receptor in human pancreatic sections which we have included in the methods section.

Supplementary Methods: "The 5-HT_{2B} receptor signal was amplified by tyramide signal amplification using the Tyramide SuperBoost™ Kit (Thermo Fisher Scientific, cat. no. B40941), as per the manufacturer's instructions"

The efficacy of the Tyramide SuperBoost™ Kit was validated through IHC-IF analysis of human placental sections, as shown in Supplementary Figure S11c.

Importantly, no signal was detected in β cells despite tyramide signal amplification and is presented in Figure S12. This further validates the reliability of our initial results.

Line 698 – 706: “To further validate the absence of 5-HT2B receptor expression in human β -cells from both pregnant and non-pregnant donors, we utilised tyramide signal amplification to enhance detection sensitivity in human pancreatic sections. The enhanced efficacy of the tyramide signal amplification method was verified through IHC-IF analysis of human placental sections (Supplementary Fig. S12c). Absent 5-HT2B receptor expression in human β -cells was consistently demonstrated, regardless of donor pregnancy status. Colocalisation analysis of whole FFPE sections further validated the lack of overlap between the 5-HT2B receptor and insulin across entire tissue sections even with the use of tyramide signal amplification of the 5-HT2B receptor signal (Supp. Fig. S13).”

- Again regarding the bihormonal cells, were they included in the plots for alpha and beta cells? Have they been considered when investigating the expression of PRLR and 5-HT2B in islets?

Thank you for this valuable point. As noted in our response to a previous comment, considering the limitations mentioned by the reviewer in using spinning disk confocal microscopy to analyse bihormonal cells, bihormonal cells were not included in the plots for α and β cells, nor were they considered when investigating the expression of PRLR and 5-HT2B in islets. This ensures that our analysis reflects a clear distinction between α and β cell populations.

- Placental lactogen, which is produced starting from 24 weeks of gestation, can also bind to the growth hormone receptor (GRH), although with lower affinity than to PRLR. Therefore it would have been appropriate to analyse its expression by immunomicroscopy.

We appreciate the reviewer highlighting the importance of analysing placental lactogen expression by immunomicroscopy, particularly considering its potential role in signaling through both the growth hormone receptor (GHR) and prolactin receptor (PRLR), albeit with differing affinities. While we recognise the relevance of such an analysis, we regret that we could not conduct this experiment due to limited remaining sample availability.

The samples used in our study were rare and were allocated through the nPOD biorepository, which necessitated prioritisation of experiments. We elected to use the remaining pancreatic tissue samples to address the requests for additional experiments by this as well as the other reviewers that focused on validating and strengthening the robustness of the presented data. Given the significance of ensuring the reliability and clarity of the findings we presented; we believed this approach was the most prudent course of action. We acknowledge that a follow up study focusing on immunomicroscopy for placental lactogen could provide additional insights into its interaction dynamics with GHR and PRLR during pregnancy.

- Supplementary Fig. 6. In the western blots show in panel A, the size of the molecular weight markers is missing. In the case of the PRLR HEK293T cells, could the overexpressed receptor be detected also with an antibody directed against the DDK tag? In panel c, the difference in the PRLP signal between WT and PRLP KO cells is not readily visible.

Thank you for this observation. We have addressed these points as follows:

1. Molecular weight markers have been included in the western blots shown in Supplementary Figure 6, panel A.
2. The overexpressed PRLR in HEK293T cells has been additionally validated using a western blot with an antibody directed against the DDK tag, as shown in Supplementary Figure 6, panel A. The anti-DDK tag antibody used is included in Table S2.
3. In panel C, the PRLR signal difference between WT and PRLR KO cells is now more readily appreciable following improved optimisation of the figure.

- The absence of 5-HT2B in pancreatic islet beta cells in situ could have been corroborated single molecule RNA FISH, which benefits from greater potentials for signal amplification.

Thank you for this insightful suggestion. We agree that single-molecule RNA FISH could provide additional corroboration for the absence of 5-HT2B in pancreatic islet β cells. However, while RNA FISH is technically feasible in FFPE tissues, the fixation and embedding process often leads to RNA degradation, which can limit the sensitivity for detecting low-abundance transcripts such as HTR2B. Additionally, the limited availability of pancreatic tissue samples further constrained our ability to perform this analysis. As

our primary focus was on protein expression rather than RNA, we prioritised the use of these scarce samples for protein detection. Given these challenges, we opted to employ tyramide signal amplification, as suggested by the reviewer, which validated the original findings, and we believe robustly supports our conclusion. We appreciate your understanding and value this suggestion for consideration in follow up studies.

- Serotonin can bind to multiple receptors, including 5-HTR3, which was shown to affect insulin secretion of mouse beta cells (Kim et al, 2015) and be upregulated in pregnant mice (Ohara-Imaizumi et al, 2013). Was 5-HTR3 detected in the proteomic analysis? Along this line, the expression of other serotonin receptors, and especially 5-HTR3 would have been appropriate.

Thank you for this valuable observation. 5-HT3 was not detected in the proteomic analysis. We agree that investigating the expression of other serotonin receptors, including 5-HT1D and 5-HT1F, would provide additional insights given their potential roles in insulin secretion and islet adaptations during pregnancy. In our study, we performed validation studies for several antibodies targeting these receptors. However, the high similarity in the amino acid sequences of the immunogen used to raise the commercial antibodies for each respective serotonin receptor subtype resulted in insufficient specificity of the available antibodies, as confirmed by our validation experiments. Given the limited availability of high-quality pancreatic tissue samples, we decided against proceeding with these non-specific antibodies, as this would not yield reliable data. We believe that without reliable reagents, it is not feasible to accurately quantify these receptors at this stage. We appreciate the reviewer's understanding and value this suggestion, which we hope to address in follow up studies when more specific reagents become available.

We have included this as a limitation.

Line 1036 -1041: "We acknowledge the limitation of not exploring additional serotonin receptor isoforms due to challenges with antibody specificity. Despite efforts to validate commercial antibodies for receptors such as the serotonin 1D, serotonin 1F, and serotonin 3 receptors, high sequence similarity among subtypes hindered reliable analysis. Future studies employing alternative approaches may help address this gap."

- The discussion cites evidence that ~80% of human beta cells are in contact with alpha cells (lines 832-833). In view of these data, the suggestion that the greater

alpha cell population size in pregnancy is required to better influence beta cells is rather puzzling/naïve (lines 907-909). This notion, on the other hand, is challenged by recent evidence that ~50% of human insulin-expressing islets are virtually devoid of glucagon-producing α -cells (Lehrstrand et al, 2024).

We thank the reviewer for their insightful comments regarding the role of α -cell expansion in pregnancy and its influence on β cells.

Regarding the interpretation of α -cell expansion during pregnancy, we acknowledge that attributing this solely to "better influence β cells" may oversimplify its role. As noted, approximately 80% of human β cells are already in contact with α cells under non-pregnant conditions, suggesting that increased α -cell mass may not primarily serve to enhance direct interactions with β cells. We now explore how α -cell adaptations may contribute to broader islet functionality, such as the upregulation of GLP-1, which enhances insulin secretion and supports paracrine signaling mechanisms. These adaptations suggest that α cells may play a crucial role in maintaining metabolic homeostasis during the increased metabolic demands of pregnancy.

Lehrstrand et al.'s excellent study indeed highlights that approximately 50% of insulin-expressing islets lack glucagon-producing α cells; however, it is important to note that these islets in fact accounts for just 16% of the total islet volume. Thus, in human islets, the overwhelming majority of β cells (84% of total islet volume) maintain contact with α cells either directly or through intra-islet communication. We have cited this paper in our resubmission.

To address the reviewer's concerns and strengthen our discussion, we have now included additional data from our cohort. We analysed the proportion of islets containing α cells and compared the pregnant to the non-pregnant group. We observed that approximately 70% of islets in the pancreas tail region contained α cells. While our analysis was limited to a smaller region of interest and employed 2D imaging techniques, these findings underscore the role of α -cell interactions in human islets.

Line 587 – 590: "The proportion of islets containing α cells was also analysed and no difference was observed between pregnant and non-pregnant donors ($71.71 \pm 4.97\%$ vs. $68.86 \pm 5.01\%$, $p = 0.5946$) (Fig. 4o)."

Line 908 – 929: "The increased α -cell mass we observe during human pregnancy may enhance the production and secretion of GLP-1 to support β -cell function through paracrine pathways. Importantly, increased glucagon secretion is unlikely to increase

circulating glucagon levels in human pregnancy, as plasma glucagon levels, unlike insulin, rise only intermittently during pregnancy 44, 63 and even mouse studies note low circulating and intra-islet glucagon in late gestation and no change in glucagon content or gene expression. These observations suggest that during pregnancy glucagon secretion from α cells may not play a key role in islet adaptations, however, the contribution of glucagon on intra-islet signaling in human pregnancy cannot be ruled out and requires further study 2, 22, 23.

The physiological adaptations of pregnancy impose significant metabolic challenges, necessitating coordinated islet function. The observed α -cell expansion during pregnancy likely reflects a broader role played by α cells in maintaining maternal glucose homeostasis and adapting to metabolic alterations in pregnancy. Our findings align with evidence from the literature in mice, providing reassurance that the observed adaptations in α cells in mice models are conserved in human pregnancy and may contribute to pregnancy-associated metabolic changes. This concordance supports the need for mechanistic studies to investigate the effects of enhanced α -cell mass and mediators of paracrine communication, such as GLP-1, on islets during human pregnancy. These studies would help clarify the role of α cells in maintaining metabolic balance during this unique physiological state.”

Line 890 – 897: “A recent study using advanced 3D imaging demonstrated that approximately 50% of human insulin-expressing islets lack α cells, although these islets constitute only 16% of the total islet volume. This indicates that the vast majority of β cells still maintained either direct or indirect contact with α cells, underscoring the relevance of intra-islet communication. In our study, we observed that 70% of islets contained α cells, all be it compared to the study by Lehrstrand et al. we studied sections only from the pancreas tail, a smaller region of interest, and performed a 2D analysis.”

- As mentioned above, the suggestion that alpha cells transdifferentiate into beta cells during pregnancy is merely speculative. In such a scenario, shouldn't the number of beta cells increase more and not less than alpha cells, especially in view of the limited evidence of islet cell proliferation by Ki67+ labelling? In this context, the increase of alpha cells in the periductal region of pregnant donors is mentioned (lines 812-814), also in relationship with the number of beta cells in the same region, but no evidence for this is provided (and no duct cell marker is shown). And if so, shouldn't the islet hormone positive clusters, i.e. the islet density being increased rather than remain equivalent in pregnant compared to

non-pregnant donors (Suppl. Fig. S1a)?

We thank the reviewer for their thoughtful comment regarding the mechanisms contributing to β -cell expansion during pregnancy. We acknowledge the speculative nature of α -to- β cell transdifferentiation and agree that further evidence is required to substantiate this mechanism.

In response to the reviewer's query, we have revised the text to clarify our findings and address the limitations of our study. Specifically, we discuss the observed increases in fractional and mean β -cell area during pregnancy and their consistency with prior reports. We highlight the three commonly proposed mechanisms for increasing β -cell numbers—proliferation, neogenesis, and transdifferentiation—and provide a detailed discussion of our findings in this context.

Our analysis did not detect evidence of β -cell proliferation, but as noted in the revised text, this may be due to the gestational age of our samples, which were predominantly from late gestation. Previous studies suggest that proliferation may occur earlier in pregnancy. To explore this further, we propose that additional studies using human pancreas samples from earlier gestational stages are needed to evaluate this possibility.

With regards to islet cell neogenesis, we agree with the reviewer's point that the lack of an increase in islet density in pregnancy finding makes neogenesis an unlikely explanation. We note the discrepancy with *Butler et al.* regarding islet density, attributing this to differences in methodology. *Butler et al.* used a broader definition of islets and analysed randomly selected areas of tissue, whereas we analysed entire tissue sections with stricter criteria to minimise non-specific labelling. This methodological difference may explain why we did not observe a significant increase in islet density. Furthermore, as we do not propose neogenesis as a primary mechanism, we have not included analysis of ductal structures or utilised a ductal marker in our study.

Additionally, while our study observed an increase in bihormonal cells during pregnancy which may suggest transdifferentiation as a possible mechanism of β cell mass expansion. However, we acknowledge the limitations of our imaging techniques in precisely distinguishing individual cell from closely juxtaposed α and β cells and that we did not observe a disproportionately greater increase in β cells compared to α cells, which would be expected if transdifferentiation were a major mechanism.

We have refined the text to focus on the observed changes in islet composition during pregnancy without attributing them to speculative mechanisms. We acknowledge that the data we present does not indicate which specific mechanism is responsible for the

increase in β cell number we observe, and further work is needed to elucidate the mechanism. By doing so, we aim to present a well characterised and evidence-based interpretation of the data.

Line 807 – 849: “The increases in fractional and mean β -cell area observed during pregnancy are consistent with previous reports in human pregnancy 15, 16. Notably, this expansion in β -cell area is driven by increased cell numbers rather than hypertrophy. Three mechanisms for increasing β cell numbers are commonly proposed in the literature: increased proliferation, neogenesis, and transdifferentiation from one islet cell type to β cells.

In our study, we did not detect evidence of β cell proliferation. However, β cells are noted to proliferate in early- to mid-gestation, while our analysis was limited to late gestation, with only a single donor from early pregnancy. Although β cell proliferation remains a controversial topic due to the inherently low proliferative potential of β cells in islet, further studies examining earlier gestational stages in human pancreas samples are necessary before ruling out proliferation as a mechanism for increased cell numbers.

Butler et al., analysed post-mortem pancreas samples from pregnant women, most of whom were in early to mid-gestation, increases due to an increase in islet density the rise in β cell mass occurs as a result of neogenesis¹⁵. However, unlike Butler et al., we did not observe a significant difference in islet density, which does not support the occurrence of islet neogenesis in our study. This discrepancy may stem from differences in methodology. Butler et al. defined a cluster of four or more insulin-positive cells as an islet and quantified islet density within randomly selected areas of the tissue, whereas our study involved analysis of the entire tissue section, employing stringent criteria to exclude smaller islets and minimise non-specific labelling¹⁵.

The potential for transdifferentiation from α cells to β cells has also been described in human islets 58, 59, 60. In our study, we observed bihormonal cells within islets and noted an increase in both their fractional and mean areas during pregnancy, suggesting that α -to- β cell transdifferentiation may contribute to islet plasticity. However, this interpretation is limited by the resolution of the imaging techniques used. Without the application of super-resolution microscopy, it is challenging to distinguish individual bihormonal cells from closely juxtaposed α and β cells. Additionally, we did not observe a disproportionately greater increase in β cells compared to α cells, which would be expected if transdifferentiation were a significant contributing mechanism.

Taken together, our findings are consistent with previous studies showing that β -cell mass increases during pregnancy based on an increase in cell number. However, the underlying mechanisms remain incompletely understood. Further investigation using high-quality human pancreas samples from earlier gestational stages is critical to

elucidate the contributions of proliferation, neogenesis, and transdifferentiation to islet plasticity during pregnancy.”

We hope these revisions address the reviewer’s concerns and enhance the clarity and transparency of the manuscript.

Reviewer response to changes to the manuscript

To the Editor and the Authors

The authors have largely addressed my concerns by acknowledging in the text the limitations related to the number of available samples, reagents, and methodological approaches, which ultimately temper the conclusions that can be drawn from this study. The result is a substantially improved article.

However, regarding my comment about placental lactogen and the expression of GHR, it would be appropriate for the authors to explicitly state among the study’s limitations that PRL may also bind to this receptor, but that its expression could not be investigated due to the restricted availability of tissue samples.

That said, I still maintain that the rarity of samples from pregnant women—contrary to the authors’ implication in their introductory remarks—does not underscore their scientific value, but rather highlights the complexity of drawing sound conclusions about the scientific questions raised. If the limited number and/or significant heterogeneity of samples hinder such conclusions, it would be preferable to refrain from making them, as they risk being merely speculative. Evidence of similar samples being included in prior studies published in high-profile journals does not justify lowering the threshold for rigorous analysis; rather, it raises important questions about our standards for acceptable scientific practice. Along the same lines, being the largest study on islets from pregnant women does not automatically imply that the cohort is adequately powered to yield robust insights. As the authors themselves acknowledge in response to my initial comments, the results remain preliminary. Therefore, in my view, their readiness for publication is still uncertain.

We thank the reviewer for their time and thoughtful comments, which have contributed to improving the manuscript.

We have included in the text that prolactin can also bind the growth hormone receptor but due to sample limitation its' expression could not be evaluated.

Discussion: "Prolactin can also bind to the pancreatic growth hormone receptor (GHR); however, restricted availability of islet tissue prevented assessment of GHR expression, representing a limitation of this study."

Reviewer #2 (Remarks to the Author):

Dear Reviewers,

Thank you for reviewing the manuscript and for your insightful comments.

Please find our responses to your feedback below. The original comments are displayed in bold and responses in normal text. Changes made to the manuscript are in italics and underlined. The relevant line numbers are also shown.

In addition to the untracked manuscript, a manuscript showing the tracked changes and line numbers is also provided.

This study was aimed to interrogate human pancreatic tissue and in doing so we sought the largest biorepository of globally available of pancreatic samples from pregnant women, the Network for Pancreatic Organ Donors with Diabetes (nPOD). This repository is unique in that it takes samples from patients perimortem thus it is important to stipulate that the patients analysed were not selected but instead all the available samples from pregnant women were used for the study. Entry into the biobank is not based on prespecified criteria. Entry into the biobank is based on consent and being close to death. The rarity of these samples underscores their scientific value, and they have been utilised in high-impact publications, including *Cell Metabolism*, *Nature Cell Biology*, *Diabetes*, and *Diabetologia*.

While we acknowledge that a more stringent selection of patient samples would have been ideal, the eight pancreata from pregnant women included in our study constitute the largest study of islets from pregnant women to date. Given how uncommon such specimens are, our study makes full use of all available samples, providing a unique and valuable opportunity to investigate human pancreatic adaptations in pregnancy.

In this manuscript Seedat F et al exploit unique FFPE pancreatic tissue samples from pregnant human donors and matched non-pregnant controls to characterize the adaptive changes occurring in islets during pregnancy, through both proteomics (LC-MS/MS) of laser-captured micro-dissected exocrine and islet tissue samples and morphometric analysis (by immunofluorescence staining and high-resolution imaging) of the whole tissue sections. Proteomic profiling shows a few changes between pregnant donors and matched controls and no significant differences between pregnant donors affected by gestational diabetes mellitus (GDM) and donors with a normal pregnancy. Morphometric analysis of tissue

sections identifies a 1.9-fold increase of islet area and a 4.3-, 1.9- and 3.1-fold increase of alpha, beta and bi-hormonal cell area, respectively, in pregnant donors compared to non-pregnant controls, possibly driven by an increase in islet size and cell number. The evaluation of prolactin receptor (PRLR) and serotonin 2B (5-HT2B) receptor expression by immunofluorescence and high-resolution imaging, as possible mediators of the observed islet changes, results in an increased expression of both PRLR and 5-HT2B receptor in alpha cells only. Finally, a 2.9-fold increase of glucagon-like peptide-1 (GLP-1) positive alpha cell area is also reported. Again, no significant differences are observed between GDM donors and donors with a normal pregnancy.

The work is of potential interest to the diabetes field. The main strengths are the rarity of pancreatic samples from pregnant human donors and the accurate experimental design and analysis. The main limits are the limited number of samples and the overall lack of biological validation. Thus, the manuscript could be implemented via additional context and additional in vitro experiments.

MAJOR

1. It is not clear whether the donors, classified as affected by GDM, have been diagnosed with GDM or if this status has been deducted from clinical details (lines 176-178). The authors should clarify, and eventually specify, which are the clinical details used for this classification and, if possible, the therapy followed by the donors. It is very unlikely that the comparison of 5 normal pregnant and 3 GDM pregnant donors only would give enough statistical power to draw a proper conclusion about the effect (or lack of it) of GDM in the performed evaluation but could be of interest anyway if supported by a proper donor characterization. The authors should add this limit (limited number of donors) also in the relative discussion section (line 886-899).

We thank the reviewer for their comments regarding the classification of donors and their thoughtful observations about the sample size and its potential limitations.

The classification of donors as affected by GDM was based on clinical details provided the nPOD biorepository. Specifically, donors were classified as GDM if they had a reported history of GDM diagnosis during the current pregnancy, based on standard clinical criteria. The Methods section of the manuscript has been revised to explicitly state this. Supplementary Table S1 now included all available details on the therapies received by the GDM donors.

Line 175 – 178: “Pregnant donors were classified by GDM status based on clinical details provided by the nPOD biorepository. Donors were identified as GDM if they had a documented history of GDM diagnosis during the current pregnancy and had received treatment for GDM during the current pregnancy.”

We acknowledge the reviewer’s observation regarding the limited number of donors (5 normal pregnant and 3 GDM pregnant donors), which inherently reduces the statistical power to draw robust conclusions about the effects of GDM. While this limitation does not allow us to definitively establish the role of GDM in our evaluations, we believe the findings remain of interest and provide valuable insights into pregnancy-associated islet adaptations.

To address this, we have added a discussion of the limited sample size for this comparison as a potential limitation of the study in the Discussion and emphasised the need for larger cohorts in future studies.

Line 1024 - 1028: “Furthermore, when comparing GDM to normal pregnancy, it is essential to interpret the data with caution due to the limited sample size. While the findings provide valuable preliminary insights, larger cohorts are needed to validate these observations and draw more definitive conclusions regarding the impact of GDM on islet adaptations during pregnancy.”

2. Proteomic analysis identifies four differentially expressed proteins (DEPs) when pregnant and non-pregnant donors are compared. This, of course, could be due to different reasons since human islets are heterogenic and pregnancy adds additional confounding factors with the variation of beta and alpha cell ratio (as suggested by the morphometric analysis). The additional use of a lower (but still acceptable) statistical threshold, at least for this comparison, might give some additional pieces of information about the specific proteomic changes of human islets in pregnancy, and better clarify the differences with what seen in mouse islets (as in ref 6, mentioned by the authors in the discussion section).

We thank the reviewer for their insightful comment regarding the use of a lower (but still acceptable) statistical threshold on the proteomic analysis. To further investigate potential proteomic differences between islets from pregnant and non-pregnant donors, we conducted an additional exploratory analysis using a more permissive statistical threshold (FDR < 0.15), consistent with the approach used in the proteomic study by analysing islets from pregnant mice conducted by *Horn et al.* This analysis aimed to

identify potential differences in protein abundance between pregnant and non-pregnant islets that may not have been detected under the more stringent FDR cutoff and to facilitate a direct comparison with the mouse study by *Horn et al.*

As described in the Methods, this exploratory analysis identified additional differentially expressed proteins, which are reported in Supplementary Figure S1. However, as noted in the Results, these findings did not influence the primary conclusions of the study, which remain based on the original $FDR < 0.05$ analysis. Furthermore, as discussed in the Discussion, none of the additional differentially expressed proteins identified in the exploratory analysis overlapped with those reported in the mouse study, suggesting potential species-specific differences in islet adaptation to pregnancy.

We believe this additional analysis provides useful context while maintaining the rigor of our original statistical approach. We appreciate the reviewer's suggestion, as it has allowed us to further explore proteomic changes while ensuring that our main conclusions remain robust.

Line 278 - 285: "To further explore proteomic differences between islets from pregnant and non-pregnant donors, an additional exploratory analysis was conducted using a more permissive threshold ($FDR < 0.15$), consistent with the approach used in the proteomic study by analysing islets from pregnant mice conducted by Horn et al. This analysis aimed to identify potential trends in protein expression that may not have been detected under the more stringent FDR cutoff and to allow for a direct comparison with the mouse study. However, the primary conclusions of the study are based solely on the $FDR < 0.05$ analysis."

Line 543 - 545: "An additional exploratory analysis using a more permissive statistical threshold ($FDR < 0.15$) was conducted to further investigate proteomic changes in islets from pregnant women, identifying additional differentially abundant proteins (Supp. Fig. S1)."

Line 792 - 795: "In an exploratory analysis using a more permissive threshold ($FDR < 0.15$), the same FDR threshold used in the study by Horn et al., additional differentially expressed proteins were identified. However, none of these overlapped with those reported in the mouse study."

3. At least one of the identified DEPs should be validated by immunofluorescence on the pancreatic tissue sections and its role in alpha and/or beta cell physiology in pregnancy investigated. Is this protein expressed by beta cells, alpha cells or

both? Does its expression change in pregnant donors and in which cell type? Are there any DEPs that could explain the decisive and unexpected increase of alpha cell area, especially of that positive for GLP-1? Some additional mechanistic studies are needed.

We thank the reviewer for their insightful comments and suggestions, which have helped refine our study. We have addressed the key aspects raised as follows:

To validate the LC-MS/MS findings, we performed immunofluorescence analysis of the pancreatic tissue sections (IHC-IF) for a differentially expressed protein, Cathepsin Z (CTSZ). The results demonstrate that CTSZ expression was significantly upregulated in islets from pregnant donors compared to non-pregnant controls, as shown by an increase in CTSZ-positive area relative to the total islet area and signal intensity. These findings are included in the text and supplied as Supplementary Figure S5, validate the proteomics analysis and confirm the upregulation of CTSZ in pancreatic islets during pregnancy.

Line 611 - 621: “3.5 Quantitative image analysis confirms pregnancy-induced upregulation of CTSZ in islets, validating LC-MS/MS findings

To validate the LC-MS/MS findings, IHC-IF was performed on human islet sections using an anti-CTSZ antibody (Supp. Fig. S6a). Quantitative analysis of the CTSZ-positive area (normalised to whole islet area) and signal intensity (Supp. Fig. S6b and c) in islets from pregnant donors was compared to non-pregnant controls. A significant increase in CTSZ-positive area (0.3865 ± 0.0094 % vs. 0.3305 ± 0.0215 %, $p = 0.038$) and signal intensity (2643 ± 106.1 AU/mm² vs. 2169 ± 154.1 AU/mm²; $p=0.0295$) in pregnant donors was observed, thereby validating the LC-MS/MS-detected upregulation of CTSZ in pancreatic islets during pregnancy.”

Based on the imaging findings, it is evident that CTSZ is predominantly expressed in β rather than α cells. Its upregulation in β cells during pregnancy may reflect an adaptive mechanism to support the increased metabolic demands associated with gestation. The specific functional role of CTSZ in β cells during pregnancy is a compelling area for further study.

While CTSZ is predominantly expressed in β cells and does not directly explain the observed increase in α cell area or GLP-1-positive α cells, this intriguing finding underscores the complexity of islet adaptations during pregnancy. At this stage, none of the DEPs identified in our proteomics analysis directly account for this observation. However, the interplay between β and α cells via paracrine or systemic signaling could be a potential mechanism and remains an area for future exploration.

We recognise the importance of additional mechanistic studies to clarify the role of CTSZ and other DEPs in islet adaptations during pregnancy. However, these investigations require extensive functional assays, cell-specific manipulations, and in vivo models, which are beyond the scope of the current study which focuses on identifying and validating proteomics alterations in human islets during pregnancy. Instead, our work provides an essential foundation for these future studies, which we intend to pursue to further elucidate the mechanistic pathways involved.

We hope that this response adequately addresses the reviewer's comments and appreciate the opportunity to clarify and strengthen our study.

4. Interestingly, the authors find a greater increase of alpha cell area than beta cell area in pregnant donors compared to control. They suggest that this might sustain the increase of insulin secretion seen during pregnancy, possibly through GLP-1 secretion and/or alpha to beta cell transdifferentiation. It is, however, not clear the role of the observed increased expression of PRLR and 5-HT2B in alpha cells in the above-mentioned changes. Some additional mechanistic studies in a mice alpha cell line (such as aTC1.9) or primary human alpha cells would give some hints about the topic. In addition, previous works on mice models should be better discussed and compared to the present results, such as those reported in ref 43 and 56.

We thank the reviewer for this thoughtful comment and for highlighting the potential significance of PRLR and 5-HT2B expression in α cells, as well as the suggestion to conduct additional mechanistic studies. We agree that further investigation into the role of PRLR and 5-HT2B in α cell adaptation, potentially using α cell lines such as aTC1.9 or primary human α cells, could provide valuable insights into the mechanisms underlying the observed changes. However, these studies are beyond the scope of the current manuscript, which focuses on identifying and validating key findings related to human pregnancy-associated islet adaptations. We have acknowledged the mechanistic studies would provide valuable insights into the role of the observed increases in the PRLR and 5-HT2B receptor. We will carefully consider pursuing these mechanistic experiments in a follow up study.

Line 982 - 983: "As both the 5-HT2B receptor and PRLR signaling can influence α cell function, further mechanistic studies are needed to clarify their roles in human α cells during pregnancy."

Additionally, we appreciate the reviewer's suggestion to discuss prior studies using mouse models, including references 43 and 56, and have revised the manuscript to better compare and contrast these findings with our results.

Line 851 – 874: “For the first time, we report an increase in α -cell area during human pregnancy. This finding aligns with studies by Qiao et al. and Quesada-Candela et al., which observed increases in α cell mass during mouse pregnancy. Notably, we observed a more pronounced increase in α -cell area relative to β -cell area during pregnancy, contrasting with mouse models where a proportional increase in α and β cell area are reported⁶⁰. In mouse models, the expansion of α -cell mass during pregnancy was attributed to cell proliferation rather than neogenesis or transdifferentiation. However, we did not observe evidence of α -cell proliferation during pregnancy in our study. This may be due to our focus on pancreas tissue from late-gestation, whereas the proliferation of α cells in mouse studies was predominantly observed during early to mid-gestation.

Qiao et al., demonstrate that GLP-1 from α cells plays an essential role in glucose-homeostasis during mouse pregnancy. They showed that total pancreatic GLP-1 abundance increases significantly during pregnancy in mice. Furthermore, α cell ablation in pregnant mice impaired glucose-stimulated insulin secretion (GSIS), resulting in hypoinsulinaemia and consequent disruptions in maternal glucose metabolism. Remarkably, treatment with GLP-1 restored GSIS in α -cell ablated pregnant mice. When GLP-1 receptor and glucagon receptor antagonist were added to islets from pregnant mice, GSIS was attenuated only in the presence of a GLP-1 receptor antagonist. The addition of a glucagon receptor antagonist, in contrast, had no effect. These findings underscore the critical role of GLP-1, rather than glucagon, in supporting β -cell function and glucose regulation during pregnancy. It has been shown that, similar to glucagon, human α cells also produce and secrete biologically active GLP-1, which acts locally on adjacent β cells to potentiate GSIS 3, 44, 62.”

We hope that these revisions address the reviewer's concerns and enhance the clarity and transparency of the manuscript.

MINOR

1. Line 145: This might be better defined as “ex vivo”, rather than “in vivo”. Please correct.

Thank you for this point. This has been corrected to ex vivo.

Line 141: “ex vivo”

2. Line 475-477: According to this phrase it seems that objects smaller than 10 μm have been excluded during nuclei identification and count. Since the beta cell has an average diameter of 10 μm (doi.org/10.1016/j.semcdb.2020.04.005), It is not clear what the authors intended.

Thank you for highlighting this point. To clarify, only objects larger than 10 μm in the nuclei (DAPI) channel were excluded, ensuring that artefacts or falsely detected objects in the nuclei channel were removed. As islet nuclei typically measure 5–8 μm , we believe this cutoff is appropriate. Cells themselves were not excluded based on this criterion. We have revised the text to correct this error and provide clearer explanation of this in the methods section.

Line 422 – 426: “To identify nuclei within islets, a watershed function was applied to the nuclei (DAPI) channel, followed by a size filter to exclude objects larger than 10 μm , ensuring artefacts or incorrectly detected objects were removed. This filtering process did not affect the identification or quantification of α cells or β cells.”

3. The number of islets isolated and subjected to proteomic analysis for each donor should be indicated in the methods, as well as the number of islets analysed per tissue section in the morphometric part.

Thank you for this valuable comment. The number of islets isolated and subjected to proteomic analysis for each donor, as well as the number of islets analysed per tissue section in the morphometric analysis, are now included in Supplementary Table S1.

4. Fig. 2b: It looks like an Overrepresentation Analysis rather than a Gene Set Enrichment Analysis as reported in figure caption and in the methods section. Please clarify.

The reviewer is correct. An over-representation analysis rather than gene set enrichment analysis was conducted. This has been corrected in the methods and figure caption.

Line 260 – 261: “An over-representation analysis was performed on proteins with a \log_2 fold change (FC) > 1 and FDR < 0.05, representing islet-specific proteins.”

Figure Caption: “b Overrepresentation analysis of enriched pathways from the KEGG and Reactome databases.”

5. Figure S1 is of interest for the reader and might be added to the main figure panels (maybe Fig 4?).

Thank you for this suggestion. The graphs from Figure S1 are not included in Figure 4.

6. Antibody validation (paragraph 3.6) is technically important, but it might better fit the methodology section instead of the results.

Thank you for this thoughtful suggestion. While we agree in principle that antibody validation is a crucial methodological aspect, we have chosen to retain this content within the results section. This decision was made to ensure clarity and accessibility for the reader, as presenting validation data in the methods section might disrupt the logical flow and make it less apparent how the validation of the antibodies support the experimental results. Including antibody validation in the results section allows readers to immediately evaluate the reliability of the data presented and its relevance to the study's findings. We hope the reviewer will understand this approach as an effort to maintain clarity and cohesion in the presentation of our work.

7. Paragraphs 3.7 and 3.8 might be better presented as one paragraph.

These have been merged into a single paragraph.

Line 665 – 677: “3.7 Increase abundance of PRLR and 5-HT2B receptors in α cells during human pregnancy

Immunolabeling of PRLR in both islets and exocrine tissue of pregnant and non-pregnant donors confirmed that PRLR is expressed at detectable levels in adult human pancreatic islets (Fig. 6a, Supp. Fig. S10 and S11a [secondary only control]).

Quantitative analysis of the PRLR signal intensity (as a corollary of PRLR abundance) revealed an increase in PRLR expression in α cells of pregnant donors compared to non-pregnant controls (302.3 ± 10.12 AU/mm² vs. 267.7 ± 4.582 AU/mm²; $p=0.0398$), whereas no difference in PRLR expression was observed in β cells of pregnant donors (277.4 ± 3.002 AU/mm² vs. 270.3 ± 1.704 AU/mm²; $p = 0.0610$) (Fig. 6b and c). No significant differences in PRLR expression was observed between donors with GDM and normal pregnancies (Supp. Fig. S11b and c).

5-HT2B receptor expression was detected in α cells in both pregnant and non-pregnant donor samples. Outside islets, 5-HT2B receptor expression was detected in peri-ductal

regions (Fig. 7a and Supp. Fig. S12a [secondary only control]). Signal intensity quantifications demonstrate an increase in the expression of 5-HT2B receptors in α cells of pregnant donors compared to non-pregnant controls (387.4 ± 10.01 Au/mm² vs. 286.5 ± 13.46 Au/mm²; $p=0.0359$) (Fig. 7b). No significant differences in 5-HT2B receptor expression (Supp. Fig. S12b) was observed between donors with GDM and normal pregnancies.”

8. Figures 7.a and 8.a should also show the staining of at least one non-pregnant donor sample. In addition, the ref to Fig 8.c should be added at line 722.

Thank you for raising this. The staining of a non-pregnant donor sample is included for Figures 7a and 8a.

Figure 8c has been included at line 722.

Line 692 - 693: “Colocalisation analysis confirmed absent colocalisation between 5-HT2B receptor and insulin across whole tissue sections (Fig. 8b and 8c).”

9. Line 890-891: The phrase is not clear; may the authors clarify?

Thank you for highlighting this. This is now clarified.

Line 986 - 990: “Clinical data show no significant differences in first-phase insulin response or insulin secretion throughout pregnancy between women with GDM and those without GDM and who have no risk factors for hyperglycaemia. However, when comparing pregnant women with GDM to non-diabetic pregnant women with risk factors for hyperglycaemia, GDM was associated with reduced insulin secretion.”

10. Please be aware that Kolic J et al (doi:10.1016/j.cmet.2024.06.001) has recently published a manuscript in which almost 8000 islet proteins are identified. The authors may want to correct sentences such as those at line 777-778 and 932-933, accordingly.

Thank you for highlighting this paper. We have cited this important paper in the discussion and amended the sentences at line 777-778 and 932-933.

Line 760 - 765: “A recent study by Kolic et al. analysing fresh-frozen islets from 90 donors by LC-MS/MS, both with and without type 2 diabetes, identified an average of

8,000 proteins⁴⁹. Even with the use of FFPE pancreatic tissue, we successfully detected over 7,000 unique proteins per sample, emphasising the capability of FFPE tissues for deep proteomic studies and identifying the highest number of proteins identified from FFPE human pancreatic islets to date.”

Line 1049 – 1051: “In conclusion, we successfully characterised the proteome of human pancreatic islets during pregnancy, creating the largest dataset of pancreatic islet and exocrine proteins derived from pancreata from pregnant women to date.”

Reviewer response to changes to the manuscript

I thank the authors for their detailed responses and revision work. Overall, the manuscript has been improved through a more in-depth discussion of the results in comparison with the literature, as well as additional analyses that provide a more comprehensive perspective.

Below, I provide my feedback on the proposed modifications.

MAJOR COMMENTS

1. Regarding the first point, the authors have adequately addressed my concern.
2. Regarding the second point, the authors have provided additional context for the proteomic analysis, which is now properly discussed in the Discussion section.
3. Regarding the third point, the authors have correctly performed a technical validation of one of the differentially expressed proteins using immunofluorescence and have also provided an appropriate discussion of the results.

4. The authors have correctly added a discussion of previous studies using mouse models, which highlight a similar increase in alpha cell mass.

Points 3-4: However, it is worth noting that the authors did not perform biological validation or mechanistic studies, stating that these aspects are beyond the scope of the present study. While this is understandable, studies published in prestigious journals, such as Nature Communications, are generally expected to include some form of biological validation to support their findings and speculations.

Additionally, I do not agree with the statement that 'the rarity of these samples underscores their scientific value,' especially since it is followed by references to other journals, which does not seem entirely appropriate. The value and rarity of the samples—while certainly recognized and acknowledged throughout the review process—should not be the primary or sole merit of a study. That said, I understand that proper validation may take time and could, indeed, be the focus

of a future study. Furthermore, the methodological and technical accuracy demonstrated throughout this work somewhat compensates for the lack of biological validation and ensures the study's overall value.

MINOR COMMENTS

1-10. Almost all the minor observations have been addressed, and the Results section is now more accessible to readers.

Final Recommendation

In conclusion, the manuscript has been improved, and most of my concerns have been addressed. I only suggest that the authors correct a typo at line 938 (where there is an extra square bracket). After this minor correction, in my view, the manuscript is ready for publication.

We thank the reviewer for their time and thoughtful comments, which have contributed to improving the manuscript.

We have corrected the typographical error at line 938, removing the additional square bracket.

Reviewer #3 (Remarks to the Author):

Dear Reviewers,

Thank you for reviewing the manuscript and for your insightful comments.

Please find our responses to your feedback below. The original comments are displayed in bold and responses in normal text. Changes made to the manuscript are in italics and underlined. The relevant line numbers are also shown.

In addition to the untracked manuscript, a manuscript showing the tracked changes and line numbers is also provided.

This study was aimed to interrogate human pancreatic tissue and in doing so we sought the largest biorepository of globally available of pancreatic samples from pregnant women, the Network for Pancreatic Organ Donors with Diabetes (nPOD). This repository is unique in that it takes samples from patients perimortem thus it is important to stipulate that the patients analysed were not selected but instead all the available samples from pregnant women were used for the study. Entry into the biobank is not based on prespecified criteria. Entry into the biobank is based on consent and being close to death. The rarity of these samples underscores their scientific value, and they have been utilised in high-impact publications, including *Cell Metabolism*, *Nature Cell Biology*, *Diabetes*, and *Diabetologia*.

While we acknowledge that a more stringent selection of patient samples would have been ideal, the eight pancreata from pregnant women included in our study constitute the largest study of islets from pregnant women to date. Given how uncommon such specimens are, our study makes full use of all available samples, providing a unique and valuable opportunity to investigate human pancreatic adaptations in pregnancy.

This study utilises precious human donor pancreas formalin-fixed paraffin embedded (FFPE) tissue blocks from nPOD from pregnant (n=7) and moderately well matched non-pregnant (n=7) women with the aim of determining the mechanisms of human islet adaptation to pregnancy. From laser captured exocrine and islet tissue, proteomics was performed which was analysed using various bioinformatics processes. Pancreas sections were also immunostained for insulin, glucagon, prolactin receptor (PRLR), serotonin receptor 2B (HTR2B) and GLP-1, and images were systematically analysed using Arivis Vision4D software. The strengths of the work are the high quality nPOP pancreas FFPE blocks, high quality immunohistochemistry analyses of pancreas with extensive validation of

the quality of the antibodies used and the methodology used for analysis using the Arivis software.

The data from the IHC studies showing increased alpha-cell area and increased expression of GLP-1 in alpha cells in pancreases of pregnant women is of particular interest.

A major achievement was successful laser microcapture of endocrine and exocrine tissue and proteomic analyses of the tissue obtained in which endocrine tissue was easily distinguished from exocrine tissue. This methodology, however, showed minimal differences between the pregnant and non-pregnant islet proteomes. This seems surprising considering the marked increases in both alpha and beta cells within the islets, suggesting that the methodology using FFPE tissue material may be limiting (easy to differentiate endocrine from exocrine tissue, but challenging to detect differences within endocrine tissue). Weaknesses include exclusion of small islets from the analyses, a failure of the proteomics to detect the PRLR and HTR2B despite it being detected in exocrine and endocrine IHC analyses. Further, only one protein within serotonin signalling has been focussed on. Another limitation that is unavoidable with the availability of samples is that there was only one first trimester pancreas and no second trimester pancreases available for analysis (islet adaptations which occur earlier in pregnancy may have been missed).

Major comments:

1) The matching of the subjects showed BMI to be higher in the pregnant women (almost significant despite low sample size). Was this a pre-pregnancy or pregnancy BMI?

Thank you for this question. Pre-pregnancy BMI measurements were not available, as the BMI provided is specific to pregnancy. The nPOD biorepository only supplies pregnancy BMI data to investigators, and due to the unique circumstances under which pancreatic tissue is obtained from donors, it is not possible to access pre-pregnancy BMI measurements. This clarification has been added to the Methods, and this has been cited as a limitation in the Discussion.

Line 181 – 182: “For pregnant donors, only the pregnancy BMI is provided.”

Line 1020 – 1024: “For pregnant donors, only the pregnancy BMI is available, as the nPOD biorepository does not provide pre-pregnancy BMI. This limitation further complicates the interpretation of BMI-related effects on islet adaptations during pregnancy.”

2) Table 1 suggests that 2 of the non-pregnant subjects were GAD Ab positive, whereas Suppl Table 1 suggests the GAD Ab subjects were in the pregnant group. Please check the data and correct.

Thank you for highlighting this error. This has been corrected in Table 1 to reflect that the GAD Ab positive donors belonged to the pregnant group.

3) Did the GAD Ab positive pancreases show any evidence of insulinitis, and could this have affected the results? This should be reported in the manuscript.

Thank you for this question. Low-grade insulinitis was detected in the Anti-GAD antibody-positive donors, as noted in the manuscript. These findings are now reported in the manuscript (Table 1) and have been included in the Discussion as limitations which add to the heterogeneity of the cohort.

Line 1018 – 1020: “Variability in ethnicity, GDM status, anti-GAD antibody status, and insulinitis among donors may have contributed to the few observed differences in proteomic profiles and may have masked more subtle biological patterns.”

4) I have concerns about the proteomics findings of pregnant vs non-pregnant islets. Validation of the proteomics of human islets obtained from formalin fixed paraffin embedded sections against fresh human islets (do not need to be pregnant subjects) would be helpful. Are some of the proteins of interest (e.g. PRL receptor, serotonin receptors, tryptophan hydroxylases) better detected in fresh islets, as opposed to FFPE islet tissue?

We appreciate the reviewer’s thoughtful suggestion regarding validation of our proteomic findings in fresh human islets. While fresh islet validation is beyond the scope of our current study due to resource constraints, we have assessed the overlap between our dataset and a recent study by *Kolic et al.*, which performed proteomics on fresh human islets from donors with and without type 2 diabetes patients. **Notably, 88% of the proteins detected in our FFPE-derived islets were also identified in *Kolic et***

al.'s fresh islet dataset, underscoring the robustness and biological relevance of our findings using FFPE islet tissue.

Interestingly, despite the use of fresh islets, *Kolic et al.* also did not detect the prolactin receptor (PRLR), serotonin 2B (5-HT2B) receptors, or tryptophan hydroxylases, suggesting that the absence of these proteins in our dataset may not be solely due to formalin fixation but could reflect intrinsic challenges in detecting these proteins using mass spectrometry or their low abundance in human islets. We have now incorporated this discussion into our manuscript to provide additional context. Future studies employing targeted proteomics approaches may help to further elucidate the expression and regulation of these proteins in human islets during pregnancy.

Line 766 – 776: “A potential concern of using FFPE islets for proteomic analysis is that protein detectability may be impacted. To assess the reliability of our dataset, we compared our dataset with that obtained by Kolic et al. who analysed fresh islets by LC-MS/MS. We found that 88% of the proteins detected in our dataset were also identified in Kolic et al.’s study, demonstrating strong consistency between FFPE-derived and fresh islet proteomes. Notably, Kolic et al. also did not detect the PRLR or 5-HT2B receptor, suggesting that the absence of these proteins in our dataset may not be solely due to FFPE preservation but could also reflect inherent challenges in detecting these proteins using mass spectrometry or their low abundance in human islets. Future studies employing targeted proteomics approaches may help to further elucidate the regulation and functional relevance of these proteins in human islets during pregnancy.”

5) Why has so much focus been placed on HTR2B? It was the tryptophan hydroxylases that had much increased expression in pregnant compared to non-pregnant islets in mice. Also could other serotonin receptor isoforms be more important for pregnancy adaptation in human islets?

Thank you for this insightful comment. As we observed an increase in β cell area associated with an increase in cell number, we chose to study the 5-HT2B receptor based on previous mouse studies, which have identified the 5-HT2B receptor as a key receptor associated with β cell proliferation. This informed our decision to prioritise the 5-HT2B receptor in our investigations.

We do, however, acknowledge the valuable point raised by the reviewer regarding the potential roles of other serotonin receptor isoforms in human islet adaptations during pregnancy. To address this, we performed antibody validation for several commercial targeting additional serotonin receptors, including 5-HT1D and 5-HT1F, which are implicated in insulin secretion and islet function. Unfortunately, despite our efforts, our

validation experiments failed to demonstrate sufficient specificity of the available commercial antibodies. This is likely due to the high similarity in amino acid sequences of the immunogens used to raise antibodies against the 5-HT1D, 5-HT1E, and 5-HT1F receptor subtypes.

Given the limited availability of high-quality pancreatic tissue samples, we decided not to proceed with these non-specific antibodies, as they would not produce reliable data. We agree that further investigation of these receptors would provide valuable insights and hope to address this question in future studies when more specific reagents become available. We appreciate the reviewer's understanding and thank them for this thoughtful suggestion.

We have included this as a limitation.

Line 1036 -1041: "We acknowledge the limitation of not exploring additional serotonin receptor isoforms due to challenges with antibody specificity. Despite efforts to validate commercial antibodies for receptors such as the serotonin 1D, serotonin 1F, and serotonin 3 receptors, high sequence similarity among subtypes hindered reliable analysis. Future studies employing alternative approaches may help address this gap."

6) HTR2B was shown to be present in human islet beta-cells by IHC previously (Diabetologia 2016 59:744–754). Why the discrepancy between this study and that previous study? Please discuss.

Thank you for this important comment. As we already noted in the discussion, 5-HT2B receptor expression in human islets was previously demonstrated using a polyclonal antibody (Diabetologia 2016 59:744–754). Polyclonal antibodies, while valuable, are raised against multiple epitopes and are prone to antigen cross-reactivity, which can limit specificity. In our study, we addressed this by employing a monoclonal antibody clone, which recognises a single epitope, thereby improving specificity. We also performed thorough antibody validation, to ensure reliable detection of the 5-HT2B receptor.

Furthermore, methodological differences such as tissue processing, staining protocols, or experimental conditions may have contributed to the discrepancy between our findings and those of the earlier study. By addressing these potential sources of variability and employing robust validation, we are confident in the reliability of our findings. We have expanded on the original explanation in the Discussion in the manuscript.

Line 954 –967: “5-HT2B receptor expression in human islets has been previously demonstrated using a polyclonal antibody 68. Given that polyclonal antibodies are raised against multiple epitopes, their limitations in antigen specificity are well known 69. To overcome this, we used a monoclonal antibody clone, which offers greater specificity due to its recognition of a single epitope. Additionally, we performed thorough antibody validation, including immunoblotting of recombinant 5-HT2B receptor protein and cell lysates known to express or lack 5-HT2B receptor expression. Validation also included IHC-IF analysis of HAP1 cell pellets and cancerous tissues known to express the 5-HT2B receptor, confirming that the antibody reliably detects the target protein. Differences in antibody specificity and validation methods between studies may account for the observed discrepancy. Furthermore, variations in tissue processing, staining protocols, or experimental conditions could also contribute to differences in reported 5-HT2B receptor expression. Our approach aimed to minimise such variability, providing robust evidence for our findings.”

7) Butler et al (Diabetologia 53, 2167-2176 (2010)), showed that there was a marked increase in small islets (including islet cell clusters) in human pregnancy, and that this could be from neogenesis from pancreatic ductal tissue. Why were small islets excluded from this study, when previously it was considered the small islets were important in islet adaptation to human pregnancy?

We appreciate the reviewer’s comment regarding the potential contribution of small islets and islet cell clusters to β -cell adaptation in pregnancy, as highlighted by *Butler et al.* (2010). In our study, we applied a size threshold of $\geq 1000 \mu\text{m}^2$ to exclude small structures, as this was necessary to ensure the specificity of the segmentation and the quantitative image analysis of well-defined islet structures. While small islets may contribute to β -cell mass expansion, distinguishing them from non-specific staining or isolated endocrine cells in our dataset posed a technical challenge. Thus, for scientific rigor and reproducibility, we maintained this threshold to ensure robust and interpretable data.

To further explore this point, we conducted an exploratory analysis using a lower size threshold ($\geq 50 \mu\text{m}^2$) and assessed islet density, as this metric is most informative for detecting islet neogenesis, particularly the formation of new small islets. This analysis did not reveal a significant difference in islet density between pregnant and non-pregnant donors (pregnant: 5.864 ± 1.145 islets per mm^2 vs. non-pregnant: 4.383 ± 0.6895 islets per mm^2 , $p = 0.2895$). Given this, and to ensure consistency in our approach, we maintain our original analysis while acknowledging that future studies

employing alternative imaging strategies may further clarify the role of small islets in human pregnancy.

8) The methodology section is very detailed which is helpful, but some of the detail could be moved to the supplemental data section.

Thank you for this comment. As suggested by the reviewer, we have moved some of the details to a Supplementary Methods section. In doing so, we ensured that all key points necessary for replicating the work remain in the main Methods section. The techniques moved to the Supplementary Methods are well-established, routine methods commonly used in the field, or additional details that, while not critical, provide further context for interested readers. Their inclusion in the supplementary section allows us to streamline the main text while maintaining clarity and reproducibility.

9) The discussion needs to cover off on the limitations of the study better.

We appreciate the reviewer's feedback regarding the need to expand the discussion of the study's limitations. In response, we have revised the discussion to comprehensively address the key constraints of our study, as outlined below:

Regarding the sample size and cohort heterogeneity, we now explicitly discuss how the limited availability of high-quality pancreatic tissue from pregnant women constrained our sample size and introduced heterogeneity into the cohort. Variability in factors such as ethnicity, GDM status, anti-GAD antibody status, and insulinitis among donors is acknowledged as a potential source of variability in the proteomic profiles, potentially masking more subtle biological patterns.

The limited sample size for the comparison of GDM and normal pregnancy is emphasised as a significant limitation. We caution that the findings, while providing preliminary insights, require validation in larger cohorts to draw more definitive conclusions about the impact of GDM on islet adaptations.

We highlight that only pregnancy BMI was available for this study due to the nature of tissue collection, which precludes the acquisition of pre-pregnancy BMI measurements. This limitation complicates the interpretation of BMI-related effects on islet adaptations during pregnancy and is an area requiring further investigation.

The limitations of islet diameter as a metric due to the irregular shapes of islets are addressed, and we explain why islet area is a more reliable measure in this context.

We also acknowledge that not exploring additional serotonin receptor isoforms is a limitation of this study.

These points have been incorporated into the discussion to provide a thorough and transparent overview of the study's limitations and to suggest directions for future research. We hope these additions adequately address the reviewer's concerns.

We hope that these revisions address the reviewer's concerns and enhance the clarity and transparency of the manuscript.

Line 1016 - 1041: "The limited availability of high-quality pancreatic tissue from pregnant women significantly constrained the sample size in this study and introduced heterogeneity within the cohort. Variability in ethnicity, GDM status, anti-GAD antibody status, and insulinitis among donors may have contributed to the few observed differences in proteomic profiles and may have masked more subtle biological patterns. For pregnant donors, only the pregnancy BMI was available. Due to the circumstances under which donor tissues are obtained, it is not possible to acquire pre-pregnancy BMI measurements, and this limitation complicates the interpretation of BMI-related effects on islet adaptations during pregnancy. Furthermore, when comparing GDM to normal pregnancy, it is essential to interpret the data with caution due to the limited sample size. While the findings provide valuable preliminary insights, larger cohorts are needed to validate these observations and draw more definitive conclusions regarding the impact of GDM on islet adaptations during pregnancy. The heterogeneity of the cohort highlights a broader challenge in the field of pregnancy-related islet adaptations. Greater collaboration between researchers, to increase the availability of pancreatic tissue from pregnant women, in order to study more homogenous cohorts is critical to advancing our understanding of islet adaptations during pregnancy and improving outcomes in GDM research. Future research should focus on studying pancreatic tissue from different stages of gestation; however, high-quality first- and second-trimester pancreas tissue were not available for this study. Measures of islet diameter are limited by the irregular shapes of islets, making islet area a more reliable metric for measurement. We acknowledge the limitation of not exploring additional serotonin receptor isoforms due to challenges with antibody specificity. Despite efforts to validate commercial antibodies for receptors such as the serotonin 1D, serotonin 1F, and serotonin 3 receptors, high sequence similarity among subtypes hindered reliable analysis. Future studies employing alternative approaches may help address this gap."

Overall, it is worthwhile making the point of the importance not to extrapolate too much from small animal studies to humans in determining the mechanisms of islet adaptation to pregnancy. However, the evidence from this study, as it

stands, is not strong enough to exclude important roles for lactogenic hormones and serotonin signaling in the adaptation of human islets to pregnancy.

Thanks for this comment. We agree with the reviewer's valuable point and have incorporated these points in the conclusion.

Line 1057 - 1065: "Our findings suggest that human islet adaptations during pregnancy differ to those observed in mice, however the role of lactogenic hormones and serotonin signaling cannot be excluded. The marked increases in α cell area and GLP-1 abundance suggest that β cells may have a greater reliance on intra-islet paracrine signaling during human pregnancy. This study provides novel data on pancreatic islet alterations during pregnancy and underscores the importance of cautious extrapolation from small animal studies and reinforces the value of using human-based model systems or tissues for pancreatic islet research."

Reviewer response to changes to the manuscript

My comments have been adequately addressed. The limitations of the small sample size and heterogeneity of the pregnancy group, as well as the methodology, is now much better outlined in the manuscript.

We thank the reviewer for their time and thoughtful comments, which have contributed to improving the manuscript.